# Fast Approximate Dynamic Programming for Infinite-Horizon Markov Decision Processes

**M. A. S. Kolarijani**
Delft Center for Systems and Control
Delft University of Technology
The Netherlands
M.A.SharifiKolarijani@tudelft.nl

**G. F. Max**
Delft Center for Systems and Control
Delft University of Technology
The Netherlands
G.F.Max@tudelft.nl

**P. Mohajerin Esfahani**
Delft Center for Systems and Control
Delft University of Technology
The Netherlands
P.MohajerinEsfahani@tudelft.nl

## Abstract

In this study, we consider the infinite-horizon, discounted cost, optimal control of stochastic nonlinear systems with separable cost and constraints in the state and input variables. Using the linear-time Legendre transform, we propose a novel numerical scheme for implementation of the corresponding value iteration (VI) algorithm in the conjugate domain. Detailed analyses of the convergence, time complexity, and error of the proposed algorithm are provided. In particular, with a discretization of size $X$ and $U$ for the state and input spaces, respectively, the proposed approach reduces the time complexity of each iteration in the VI algorithm from $\mathcal{O}(XU)$ to $\mathcal{O}(X + U)$, by replacing the minimization operation in the primal domain with a simple addition in the conjugate domain.

## 1 Introduction

Value iteration (VI) is one of the most basic and wide-spread algorithms employed for tackling problems in reinforcement learning (RL) and optimal control [9, 28] formulated as Markov decision processes (MDPs). The VI algorithm simply involves the consecutive applications of the dynamic programming (DP) operator

$$\mathcal{T}J(x_t) = \min_{u_t} \big\{ C(x_t, u_t) + \gamma \mathbb{E} J(x_{t+1}) \big\},$$

where $C(x_t, u_t)$ is the cost of taking the control action $u_t$ at the state $x_t$. This fixed point iteration is known to converge to the optimal value function for discount factors $\gamma \in (0, 1)$. However, this algorithm suffers from a high computational cost for large-scale finite state spaces. For problems with a continuous state space, the DP operation becomes an infinite-dimensional optimization problem, rendering the exact implementation of VI impossible in most cases. A common solution is to incorporate function approximation techniques and compute the output of the DP operator for a finite sample (i.e., a discretization) of the underlying continuous state space. This approximation again suffers from a high computational cost for fine discretizations of the state space, particularly in high-dimensional problems. We refer the reader to [9, 26] for various approximations of VI.

For some problems, however, it is possible to partially address this issue by using duality theory, i.e., approaching the minimization problem in the conjugate domain. In particular, as we will see

35th Conference on Neural Information Processing Systems (NeurIPS 2021).

in Section 3, the minimization in the primal domain in DP can be transformed to a simple addition in the dual domain, at the expense of three conjugate transforms. However, proper application of this transformation relies on efficient numerical algorithms for conjugation. Fortunately, such an algorithm, known as linear-time Legendre transform (LLT), has been developed in late 90s [23]. Other than the classical application of LLT (and other fast algorithms for conjugate transform) in solving Hamilton-Jacobi equation [1, 13, 14], these algorithms are used in image processing [24], thermodynamics [12], and optimal transport [17].

The application of conjugate duality for the DP problem is not new and actually goes back to Bellman [4]. Further applications of this idea for reducing the computational complexity were later explored in [15, 18]. However, surprisingly, the application of LLT for solving discrete-time optimal control problems, has been limited. In particular, in [11], the authors propose the "fast value iteration" algorithm (without a rigorous analysis of the complexity and error of the proposed algorithm) for a particular class of infinite-horizon optimal control problems with state-independent stage cost $C(x, u) = C(u)$ and deterministic linear dynamics $x_{t+1} = Ax_t + Bu_t$, where $A$ is a non-negative, monotone, invertible matrix. More recently, in [20], we also considered the application of LLT for solving the DP operation in finite-horizon, optimal control of input-affine dynamics $x_{t+1} = f_{\mathrm{s}}(x_t) + Bu_t$ with separable cost $C(x, u) = C_{\mathrm{s}}(x) + C_{\mathrm{i}}(u)$. In particular, we introduced the "discrete conjugate DP" (d-CDP) operator, and provided a detailed analysis of its complexity and error. As we will discuss shortly, the current study is an extension of the corresponding d-CDP algorithm that, among other things, considers infinite horizon, discounted cost problems. We note that the algorithms developed in [16, 24] for distance transform can also potentially tackle the optimal control problems similar to the ones of interest in the current study. In particular, these algorithms require the stage cost to be reformulated as a convex function of the "distance" between the current and next states. While this property might arise naturally, it can generally be restrictive, as it is in the problem class considered in this study. Another line of work that is closely related to ours invloves utilizing max-plus algebra in solving deterministic, continuous-state, continuous-time, optimal control problems; see, e.g., [2, 25]. These works exploit the compatibility of the DP operation with max-plus operations, and approximate the value function as a max-plus linear combination. Recently, in [3, 5], the authors used this idea to propose an approximate VI algorithm for continuous-state, deterministic MDPs. In this regard, we note that the proposed approach in the current study also involves approximating the value function as a max-plus linear combination, namely, the maximum of affine functions. The key difference is however that by choosing a grid-like (factorized) set of slopes for the linear terms (i.e., the basis of the max-plus linear combination), we take advantage of linear time complexity of LLT in computing the constant terms (i.e., the coefficients of the max-plus linear combination).

**Main contribution.** In this study, we focus on an approximate implementation of VI involving discretization of the state and input spaces for solving the optimal control problem of discrete-time systems, with continuous state-input space. Building upon our earlier work [20], we employ conjugate duality to speed-up VI for problems with separable stage cost (in state and input) and input-affine dynamics. We propose the *conjugate* VI (ConjVI) algorithm based on a modified version of the d-CDP operator introduced in [20], and extend the existing results in three directions: We consider *infinite-horizon, discounted cost* problems with *stochastic dynamics*, while incorporating a *numerical scheme for approximation of the conjugate of input cost*. The main contributions of this paper are then as follows: **(i)** we provide sufficient conditions for the convergence of ConjVI (Theorem 3.9); **(ii)** we show that ConjVI can achieve a linear time complexity of $\mathcal{O}(X + U)$ in each iteration (Theorem 3.10), compared to the quadratic time complexity of $\mathcal{O}(XU)$ of the standard VI, where $X$ and $U$ are the cardinalities of the discrete state and input spaces, respectively; **(iii)** we analyze the error of ConjVI (Theorem 3.11), and use that result to provide specific guidelines on the construction of the discrete dual domain (Section 3.4); **(iv)** we provide a MATLAB package for the implementation of the proposed ConjVI algorithm.

**Notations.** The standard inner product in $\mathbb{R}^n$ and the corresponding induced 2-norm are denoted by $\langle \cdot, \cdot \rangle$ and $\|\cdot\|_2$, respectively. $\|\cdot\|_\infty$ denotes the infinity norm. We use the superscript d to denote *finite (discrete)* sets (as in $\mathbb{X}^{\mathrm{d}}$) and *discrete* functions (as in $h^{\mathrm{d}} : \mathbb{X}^{\mathrm{d}} \to \mathbb{R}$). We use the superscript g to denote *grid-like* finite sets (as in $\mathbb{X}^{\mathrm{g}} = \Pi_{i=1}^n \mathbb{X}_i^{\mathrm{g}}$ where $\mathbb{X}_i^{\mathrm{g}} \subset \mathbb{R}$). We also use $\mathbb{X}_{\mathrm{sub}}^{\mathrm{g}}$ to denote the *sub-grid* of $\mathbb{X}^{\mathrm{g}}$ derived by omitting the smallest and the largest elements of $\mathbb{X}^{\mathrm{g}}$ in each dimension. The cardinality of the finite set $\mathbb{X}^{\mathrm{d}}$ or $\mathbb{X}^{\mathrm{g}}$ is denoted by $X$. We use $\widetilde{h^{\mathrm{d}}} : \mathbb{R}^n \to \overline{\mathbb{R}} = \mathbb{R} \cup \{\infty\}$ to denote a *generic extension* of a discrete function $h^{\mathrm{d}}$, and $\overline{h^{\mathrm{d}}} : \mathbb{R}^n \to \overline{\mathbb{R}}$ to denote *multilinear interpolation*

*and extrapolation (LERP)* extension of a discrete function with a *grid-like* domain. Let $\mathbb{X}, \mathbb{Y}$ be two arbitrary sets in $\mathbb{R}^n$. $\text{co}(\mathbb{X})$ is the convex hull of $\mathbb{X}$. We use $\text{d}(\mathbb{X}, \mathbb{Y}) \coloneqq \inf_{x \in \mathbb{X}, y \in \mathbb{Y}} \|x - y\|_2$ to denote the distance between $\mathbb{X}$ and $\mathbb{Y}$. The one-sided Hausdorff distance *from $\mathbb{X}$ to $\mathbb{Y}$* is defined as $\text{d}_{\text{H}}(\mathbb{X}, \mathbb{Y}) \coloneqq \sup_{x \in \mathbb{X}} \inf_{y \in \mathbb{Y}} \|x - y\|_2$. Let $h : \mathbb{R}^n \to \overline{\mathbb{R}}$ be an extended real-valued function with a non-empty effective domain $\text{dom}(h) = \mathbb{X} \coloneqq \{x \in \mathbb{R}^n : h(x) < \infty\}$. The range of $h$ is denoted by $\text{rng}(h) \coloneqq \max_{x \in \mathbb{X}} h(x) - \min_{x \in \mathbb{X}} h(x)$, and the subdifferential of $h$ at a point $x \in \mathbb{X}$ is defined as $\partial h(x) \coloneqq \{y \in \mathbb{R}^n : h(\tilde{x}) \geq h(x) + \langle y, \tilde{x} - x \rangle, \forall \tilde{x} \in \mathbb{X}\}$. We define $\mathbb{L}(h) \coloneqq \Pi_{i=1}^n \left[ \text{L}_i^-(h), \text{L}_i^+(h) \right]$, where $\text{L}_i^+(h)$ (resp. $\text{L}_i^-(h)$) is the maximum (resp. minimum) slope of the function $h$ along the $i$-th dimension. Note that $\partial h(x) \subseteq \mathbb{L}(h)$ for all $x \in \mathbb{X}$. We report the complexities using the standard big-O notations $\mathcal{O}$ and $\widetilde{\mathcal{O}}$, where the latter hides the logarithmic factors. In this study, we are mainly concerned with the dependence of the computational complexities on *the size of the finite sets* involved (discretization of the primal and dual domains). In particular, we ignore the possible dependence of the computational complexities on the dimension of the variables, unless they appear in the power of the size of those discrete sets.

We note that the extended version of this article, including the technical proofs, is available in the supplementary material.

## 2   VI in primal domain

We are concerned with the infinite-horizon, discounted cost, optimal control problems of the form

$$J_\star(x) = \min \ \mathbb{E}_{w_t} \left[ \sum_{t=0}^\infty \gamma^t C(x_t, u_t) \bigg| x_0 = x \right]$$

$$\text{s.t.} \quad x_{t+1} = g(x_t, u_t, w_t), \ x_t \in \mathbb{X}, \ u_t \in \mathbb{U}, \ w_t \sim \mathbb{P}(\mathbb{W}), \quad \forall t \in \{0, 1, \dots\},$$

where $x_t \in \mathbb{R}^n$, $u_t \in \mathbb{R}^m$, and $w_t \in \mathbb{R}^l$ are the state, input and disturbance variables at time $t$, respectively; $\gamma \in (0, 1)$ is the discount factor; $C : \mathbb{X} \times \mathbb{U} \to \mathbb{R}$ is the stage cost; $g : \mathbb{R}^n \times \mathbb{R}^m \times \mathbb{R}^l \to \mathbb{R}^n$ describes the dynamics; $\mathbb{X} \subset \mathbb{R}^n$ and $\mathbb{U} \subset \mathbb{R}^m$ describe the state and input constraints, respectively; and, $\mathbb{P}(\cdot)$ is the distribution of the disturbance over the support $\mathbb{W} \subset \mathbb{R}^l$. Assuming the stage cost $C$ is bounded, the optimal value function solves the equation $J_\star = \mathcal{T} J_\star$, where $\mathcal{T}$ is the DP operator ($C$ and $J$ are extended to infinity outside their effective domains) [7, Prop. 1.2.2]

$$\mathcal{T} J(x) \coloneqq \min_u \left\{ C(x, u) + \gamma \cdot \mathbb{E}_w J\big(g(x, u, w)\big) \right\}, \quad \forall x \in \mathbb{X}. \tag{1}$$

Indeed, $\mathcal{T}$ is $\gamma$-contractive in the infinity-norm [7, Prop. 1.2.4]. This property then gives rise to the VI algorithm $J_{k+1} = \mathcal{T} J_k$, which converges to $J_\star$ as $k \to \infty$, for arbitrary initialization $J_0$. Moreover, assuming that the composition $J \circ g$ (for each $w$) and the cost $C$ are jointly convex in the state and input variables, $\mathcal{T}$ also preserves convexity [8, Prop. 3.3.1].

For numerical implementation of VI, we need to address three issues. First, we need to compute the expectation in (1). In order to simplify the exposition and include the computational cost of this operation explicitly, we consider disturbances with finite support in this study:

**Assumption 2.1** (Disturbance with finite support). *The disturbance $w$ has a finite support $\mathbb{W}^{\text{d}} \subset \mathbb{R}^l$ with a given probability mass function (p.m.f.) $p : \mathbb{W}^{\text{d}} \to [0, 1]$.*

Under the preceding assumption, we have $\mathbb{E}_w J\big(g(x, u, w)\big) = \sum_{w \in \mathbb{W}^{\text{d}}} p(w) \cdot J\big(g(x, u, w)\big)$. The second and more important issue is that the optimization problem (1) is infinite-dimensional for the continuous state space $\mathbb{X}$. This renders the exact implementation of VI impossible, except for a few cases with available closed-form solutions. A common solution to this problem is to deploy a sample-based approach, accompanied by a function approximation scheme. To be precise, for a finite subset $\mathbb{X}^{\text{d}}$ of $\mathbb{X}$, at each iteration $k = 0, 1, \dots$, we take the discrete function $J_k^{\text{d}} : \mathbb{X}^{\text{d}} \to \mathbb{R}$ as the input, and compute the discrete function $J_{k+1}^{\text{d}} = \left[ \mathcal{T} \widetilde{J_k^{\text{d}}} \right]^{\text{d}} : \mathbb{X}^{\text{d}} \to \mathbb{R}$, where $\widetilde{J_k^{\text{d}}} : \mathbb{X} \to \mathbb{R}$ is an extension of $J_k^{\text{d}}$. Finally, for each $x \in \mathbb{X}^{\text{d}}$, we have to solve the minimization problem in (1) over the control input. Since this minimization problem is often a difficult, non-convex problem, a common approximation involves enumeration over a discretization $\mathbb{U}^{\text{d}} \subset \mathbb{U}$. Incorporating these approximations, we end up with the approximate VI algorithm $J_{k+1}^{\text{d}} = \mathcal{T}^{\text{d}} J_k^{\text{d}}$, characterized by the *discrete* DP (d-DP) operator

$$\mathcal{T}^{\text{d}} J^{\text{d}}(x) \coloneqq \min_{u \in \mathbb{U}^{\text{d}}} \left\{ C(x, u) + \gamma \cdot \sum_{w \in \mathbb{W}^{\text{d}}} p(w) \cdot \widetilde{J^{\text{d}}}\big(g(x, u, w)\big) \right\}, \quad \forall x \in \mathbb{X}^{\text{d}}. \tag{2}$$

The convergence of approximate VI described above depends on the properties of the extension operation $\widetilde{[\cdot]}$. In particular, if the extension operation is non-expansive (in the infinity-norm), then $\mathcal{T}^{\mathrm{d}}$ is also $\gamma$-contractive. The error of this approximation also depends on the extension operation and its representative power. We refer the interested reader to [7, 10, 26] for detailed discussions on the convergence and error of different approximation schemes for VI.

The d-DP operator and the corresponding VI algorithm will be our benchmark for evaluating the performance of the alternative algorithm developed in this study. To this end, we finish this section with some remarks on the time complexity of the d-DP operation. Let the time complexity of a single evaluation of the extension operator $\widetilde{[\cdot]}$ in (2) be of $\mathcal{O}(E)$. Then, the time complexity of the d-DP operation (2) is of $\mathcal{O}\left(XUWE\right)$. In this regard, note that the scheme described above essentially involves approximating a continuous-state/action MDP with a finite-state/action MDP, and then applying the VI algorithm. This, in turn, implies the lower bound $\Omega(XU)$ for the time complexity (corresponding to enumeration over $u \in \mathbb{U}^{\mathrm{d}}$ for each $x \in \mathbb{X}^{\mathrm{d}}$). This lower bound is also compatible with the best existing time complexities in the literature for VI for finite MDPs; see, e.g., [3, 27]. However, as we will see in the next section, for a particular class of problems, it is possible to exploit the structure of the underlying continuous system in order to achieve a better time complexity in the corresponding discretized problem.

## 3 Reducing complexity via conjugate duality

In this section, we present the class of problems that allows us to employ conjugate duality and propose an alternative path for solving the corresponding DP operator. We also present the numerical scheme for implementing the proposed alternative path, and analyze its convergence, complexity, and error. We note that the proposed algorithm and its analysis are based on the d-CDP algorithm presented in [20, Sec. 5] for finite-horizon, optimal control of deterministic systems. Here, we extend those results for infinite-horizon, discounted cost, optimal control of stochastic systems. Moreover, unlike [20], our analysis includes the case where the conjugate of input cost is not analytically available and has to be computed numerically; see [20, Assump. 5.1] for more details.

### 3.1 VI in conjugate domain

Throughout this section, we assume that the problem data satisfy the following conditions.

**Assumption 3.1** (Problem class)**.** *The problem data has the following properties: (i) The* dynamics *is of the form* $g(x, u, w) = f(x, u) + w = f_{\mathrm{s}}(x) + Bu + w$, *with additive disturbance, where* $f_{\mathrm{s}} : \mathbb{R}^n \to \mathbb{R}^n$ *is a Lipschitz continuous, possibly nonlinear map, and* $B \in \mathbb{R}^{n \times m}$. *(ii) The* stage cost $C$ *is separable in state and input; that is,* $C(x, u) = C_{\mathrm{s}}(x) + C_{\mathrm{i}}(u)$, *where the state cost* $C_{\mathrm{s}} : \mathbb{X} \to \mathbb{R}$ *and the input cost* $C_{\mathrm{i}} : \mathbb{U} \to \mathbb{R}$ *are Lipschitz continuous. (iii) The* constraint *sets* $\mathbb{X} \subset \mathbb{R}^n$ *and* $\mathbb{U} \subset \mathbb{R}^m$ *are compact. Moreover, for each* $x \in \mathbb{X}$, *the set of admissible inputs* $\mathbb{U}(x) \coloneqq \{u \in \mathbb{U} : g(x, u, w) \in \mathbb{X}, \; \forall w \in \mathbb{W}^{\mathrm{d}}\}$ *is nonempty.*

Some remarks are in order regarding the preceding assumptions. We first note that the setting of Assumption 3.1 goes beyond the classical LQR. In particular, it includes nonlinear dynamics, state and input constraints, and non-quadratic stage costs. Second, the properties laid out in Assumption 3.1 imply that the set of admissible inputs $\mathbb{U}(x)$ is a compact set for each $x \in \mathbb{X}$. This, in turn, implies that the optimal value in (1) is achieved if $J : \mathbb{X} \to \mathbb{R}$ is also assumed to be lower semi-continuous.

For the problem class of Assumption (3.1), we can use duality theory to present an alternative path for computing the output of the DP operator. This path forms the basis for the algorithm proposed in this study. Fix $x \in \mathbb{X}$ and consider the following reformulation of the optimization problem (1)

$$\mathcal{T}J(x) = C_{\mathrm{s}}(x) + \min_{u,z} \left\{ C_{\mathrm{i}}(u) + \gamma \cdot \mathbb{E}_w J(z + w) : z = f(x, u) \right\},$$

where we used additivity of disturbance and separability of stage cost. The corresponding dual problem then reads as

$$\widehat{\mathcal{T}}J(x) \coloneqq C_{\mathrm{s}}(x) + \max_{y} \min_{u,z} \left\{ C_{\mathrm{i}}(u) + \gamma \cdot \mathbb{E}_w J(z + w) + \langle y, f(x, u) - z \rangle \right\}, \tag{3}$$

where $y \in \mathbb{R}^n$ is the dual variable corresponding to the equality constraint. For the deterministic dynamics of Assumption 3.1-(i), we can obtain the following representation for the dual problem.

**Proposition 3.2** (CDP operator). *The dual problem* (3) *equivalently reads as*

$$\epsilon(x) \coloneqq \gamma \cdot \mathbb{E}_w J(x + w), \quad x \in \mathbb{X}, \tag{4a}$$

$$\phi(y) \coloneqq C_i^*(-B^\top y) + \epsilon^*(y), \quad y \in \mathbb{R}^n, \tag{4b}$$

$$\widehat{\mathcal{T}} J(x) = C_s(x) + \phi^*\big(f_s(x)\big), \quad x \in \mathbb{X}, \tag{4c}$$

*where* $[\cdot]^*$ *denotes the conjugate operation.*

Following [20], we call the operator $\widehat{\mathcal{T}}$ in (4) the *conjugate* DP (CDP) operator. We next provide an alternative representation of the CDP operator that captures the essence of this operation.

**Proposition 3.3** (CDP reformulation). *The CDP operator* $\widehat{\mathcal{T}}$ *equivalently reads as*

$$\widehat{\mathcal{T}} J(x) = C_s(x) + \min_u \left\{ C_i^{**}(u) + \gamma \cdot [\mathbb{E}_w J(\cdot + w)]^{**}\big(f(x, u)\big) \right\}, \tag{5}$$

*where* $[\cdot]^{**}$ *denotes the biconjugate operation.*

The preceding result implies that the indirect path through the conjugate domain essentially involves substituting the input cost and (expectation of the) value function by their biconjugates. In particular, it points to a sufficient condition for zero duality gap.

**Corollary 3.4** (Equivalence of $\mathcal{T}$ and $\widehat{\mathcal{T}}$). $\widehat{\mathcal{T}} J = \mathcal{T} J$ *if* $C_i : \mathbb{U} \to \mathbb{R}$ *and* $J : \mathbb{X} \to \mathbb{R}$ *are convex.*

Hence, $\widehat{\mathcal{T}}$ has the same properties as $\mathcal{T}$ if $C_i$ and $J$ are convex. More importantly, if $\mathcal{T}$ and $\widehat{\mathcal{T}}$ preserve convexity, then the *conjugate* VI (ConjVI) algorithm $J_{k+1} = \widehat{\mathcal{T}} J_k$, also converges to the optimal value function $J_\star$, with arbitrary convex initialization $J_0$. For convexity to be preserved, however, we need two more additional assumptions. First, the state cost $C_s : \mathbb{X} \to \mathbb{R}$ needs to be also convex. Then, for $\widehat{\mathcal{T}} J$ to be convex, a sufficient condition is the convexity of $J \circ f$ (jointly in $x$ and $u$), given that $J$ is convex. The following assumption summarizes the sufficient conditions for equivalence of VI and ConjVI algorithms.

**Assumption 3.5** (Convexity). *Consider the following properties for the constraints, costs, and dynamics: (i) The sets* $\mathbb{X} \subset \mathbb{R}^n$ *and* $\mathbb{U} \subset \mathbb{R}^m$ *are convex. (ii) The costs* $C_s : \mathbb{X} \to \mathbb{R}$ *and* $C_i : \mathbb{U} \to \mathbb{R}$ *are convex. (iii) The deterministic dynamics* $f : \mathbb{R}^n \times \mathbb{R}^m \to \mathbb{R}^n$ *is such that given a convex function* $J : \mathbb{X} \to R$, *the composition* $J \circ f$ *is jointly convex in the state and input variables.*

We note that the last condition in the preceding assumption usually does not hold for nonlinear dynamics; however, for $f_s(x) = Ax$ with $A \in \mathbb{R}^{n \times n}$, this is indeed the case for problems satisfying Assumptions 3.1 and 3.5 [6]. Note that, if convexity is not preserved, then the alternative path suffers from duality gap in the sense that in each iteration it uses the *convex envelop* of (the expectation of) the output of the previous iteration.

## 3.2 ConjVI algorithm

The approximate ConjVI algorithm involves consecutive applications of an approximate implementation of the CDP operator (4) until some termination condition is satisfied. Algorithm 1 provides the pseudo-code of this procedure. In particular, we consider solving (4) for a finite set $\mathbb{X}^d \subset \mathbb{X}$, and terminate the iterations when the difference between two consecutive discrete value functions (in the infinity-norm) is less than a given constant $e_t > 0$. Since we are working with a finite subset of the state space, we can restrict the feasibility condition of Assumption 3.1-(iii) to all $x \in \mathbb{X}^d$:

**Assumption 3.6** (Feasibile discretization). *We have* $\mathbb{U}(x) \neq \emptyset$ *for all* $x \in \mathbb{X}^d$.

In what follows, we describe the main steps within the initialization and iterations of Algorithm 1. In particular, the conjugate operations in (4) are handled numerically via the linear-time Legendre transform (LLT) algorithm [23]. LLT is an efficient algorithm for computing the *discrete* conjugate function over a finite *grid-like* dual domain. Precisely, to compute the conjugate of the function $h : \mathbb{X} \to \mathbb{R}$, LLT takes its discretization $h^d : \mathbb{X}^d \to \mathbb{R}$ as an input, and outputs $h^{d*d} : \mathbb{Y}^g \to \mathbb{R}$, for the grid-like dual domain $\mathbb{Y}^g$. We refer the reader to [23] for a detailed description of LLT.

The main steps of the proposed approximate implementation of the CDP operator (4) are as follows: (i) For the expectation operation in (4a), by Assumption 2.1, we again have $\mathbb{E}_w J(\cdot + w) =$

---

**Algorithm 1** ConjVI: Approximate VI in conjugate domain

---

**Input:** dynamics $f_s : \mathbb{R}^n \to \mathbb{R}^n$, $B \in \mathbb{R}^{n \times m}$; finite state space $\mathbb{X}^d \subset \mathbb{X}$; finite input space $\mathbb{U}^d \subset \mathbb{U}$; state cost function $C_s^d : \mathbb{X}^d \to \mathbb{R}$; input cost function $C_i^d : \mathbb{U}^d \to \mathbb{R}$; finite disturbance space $\mathbb{W}^d$ and its p.m.f. $p : \mathbb{W}^d \to [0, 1]$; discount factor $\gamma$; termination bound $e_t$.

**Output:** discrete value function $\widehat{J}^d : \mathbb{X}^d \to \mathbb{R}$.

   *initialization:*
1:  construct the grids $\mathbb{V}^g$, $\mathbb{Z}^g$, and $\mathbb{Y}^g$;
2:  use LLT to compute $C_i^{d*d} : \mathbb{V}^g \to \mathbb{R}$ from $C_i^d : \mathbb{U}^d \to \mathbb{R}$;
3:  $J^d(x) \leftarrow 0$ and $J_+^d(x) \leftarrow C_s^d(x) - \min C_i^d$ for $x \in \mathbb{X}^d$;
   *iteration:*
4:  **while** $\left\| J_+^d - J^d \right\|_\infty \geq e_t$ **do**
5:     $J^d \leftarrow J_+^d$;
6:     $J_+^d \leftarrow \widehat{\mathcal{T}}^d J^d : \mathbb{X}^d \to \mathbb{R}$ according to (6) [$C_i^{d*d}$ in (6c) is already computed in line 2];
7:  **end while**
8:  output $\widehat{J}^d \leftarrow J_+^d$.

---

$\sum_{w \in \mathbb{W}^d} p(w) \cdot J(\cdot + w)$. Hence, we need to pass the value function $J^d : \mathbb{X}^d \to \mathbb{R}$ through the "scaled expection filter" to obtain $\varepsilon^d : \mathbb{X}^d \to \overline{\mathbb{R}}$ in (6a) as an approximation of $\epsilon$ in (4a). Notice that here we are using an extension $\widetilde{J^d} : \mathbb{X} \to \mathbb{R}$ of $J^d$ (recall that we only have access to the discrete value function $J^d$). **(ii)** In order to compute $\phi$ in (4b), we need access to two conjugate functions. First, for $\epsilon^*$, we use the approximation $\varepsilon^{d*d} : \mathbb{Y}^g \to \mathbb{R}$ in (6b), by applying LLT to the data points $\varepsilon^d : \mathbb{X}^d \to \overline{\mathbb{R}}$ for a properly chosen state dual grid $\mathbb{Y}^g \subset \mathbb{R}^n$. We also need the conjugate $C_i^*$ of the input cost. If this function is not analytically available, we approximate it as follows: For a properly chosen input dual grid $\mathbb{V}^g \subset \mathbb{R}^m$, we employ LLT to compute $C_i^{d*d} : \mathbb{V}^g \to \mathbb{R}$ in (6c), using the data points $C_i^d : \mathbb{U}^d \to \mathbb{R}$, where $\mathbb{U}^d$ is a finite subset of $\mathbb{U}$. With these conjugate functions at hand, we can now compute $\varphi^d : \mathbb{Y}^g \to \mathbb{R}$ in (6d), as an approximation of $\phi$ in (4b). In particular, notice that we use the LERP extension $\overline{C_i^{d*d}}$ of $C_i^{d*d}$ to approximate $C_i^{d*}$ at the required point $-B^\top y$ for each $y \in \mathbb{Y}^g$. **(iii)** To be able to compute the output according to (4c), we need to perform another conjugate transform. In particular, we need the value of $\phi^*$ at $f_s(x)$ for $x \in \mathbb{X}^d$. Here, we use the approximation $\varphi^{d*d} : \mathbb{Z}^g \to \mathbb{R}$ in (6e), by applying LLT to the data points $\varphi^d : \mathbb{Y}^g \to \mathbb{R}$ for a properly chosen grid $\mathbb{Z}^g \subset \mathbb{R}^n$. Finally, we use the LERP extension $\overline{\varphi^{d*d}}$ of $\varphi^{d*d}$ to approximate $\varphi^{d*}$ at the required point $f_s(x)$ for each $x \in \mathbb{X}^d$, and compute $\widehat{\mathcal{T}}^d J^d$ in (6f) as an approximation of $\widehat{\mathcal{T}} J$ in (4c). With these approximations, we introduce the *discrete* CDP (d-CDP) operator as follows

$$\varepsilon^d(x) := \gamma \cdot \sum_{w \in \mathbb{W}^d} p(w) \cdot \widetilde{J^d}(x + w), \quad x \in \mathbb{X}^d, \tag{6a}$$

$$\varepsilon^{d*d}(y) = \max_{x \in \mathbb{X}^d} \left\{ \langle x, y \rangle - \varepsilon^d(x) \right\}, \quad y \in \mathbb{Y}^g, \tag{6b}$$

$$C_i^{d*d}(v) = \max_{u \in \mathbb{U}^d} \left\{ \langle u, v \rangle - C_i^d(u) \right\}, \quad v \in \mathbb{V}^g, \tag{6c}$$

$$\varphi^d(y) := \overline{C_i^{d*d}}(-B^\top y) + \varepsilon^{d*d}(y), \quad y \in \mathbb{Y}^g, \tag{6d}$$

$$\varphi^{d*d}(z) = \max_{y \in \mathbb{Y}^g} \left\{ \langle y, z \rangle - \varphi^d(y) \right\}, \quad z \in \mathbb{Z}^g, \tag{6e}$$

$$\widehat{\mathcal{T}}^d J^d(x) := C_s(x) + \overline{\varphi^{d*d}}\big(f_s(x)\big), \quad x \in \mathbb{X}^d. \tag{6f}$$

The proper construction of the grids $\mathbb{Y}^g$, $\mathbb{V}^g$, and $\mathbb{Z}^g$ will be discussed in Section 3.4.

## 3.3   Analysis of ConjVI algorithm

We now provide our main theoretical results concerning the convergence, complexity, and error of the proposed algorithm. Let us begin with presenting the assumptions to be called in this subsection.

**Assumption 3.7** (Grids)**.** *Consider the following properties for the grids in Algorithm 1 (consult the Notations in Section 1): **(i)** The grid $\mathbb{V}^g$ is constructed such that $\mathrm{co}(\mathbb{V}_{\mathrm{sub}}^g) \supseteq \mathbb{L}(C_i^d)$. **(ii)** The grid $\mathbb{Z}^g$ is constructed such that $\mathrm{co}(\mathbb{Z}^g) \supseteq f_s\big(\mathbb{X}^d\big)$. **(iii)** The construction of $\mathbb{Y}^g$, $\mathbb{V}^g$, and $\mathbb{Z}^g$ requires at most*

$\mathcal{O}(X + U)$ operations. The cardinality of the grids $\mathbb{Y}^g$ and $\mathbb{Z}^g$ (resp. $\mathbb{V}^g$) in each dimension is the same as that of $\mathbb{X}^d$ (resp. $\mathbb{U}^d$) in that dimension so that $Y, Z = X$ and $V = U$.

**Assumption 3.8** (Extension operator). *Consider the following properties for the operator $\widetilde{[\cdot]}$ in (6a):*
*(i) The extension operator is non-expansive in the infinity norm; (ii) Given a function $J : \mathbb{X} \to \mathbb{R}$ and its discretization $J^d : \mathbb{X}^d \to \mathbb{R}$, we have $\|J - \widetilde{J^d}\|_\infty \leq e_e$ for some constant $e_e \geq 0$.*

Our first result concerns the contractiveness of the d-CDP operator.

**Theorem 3.9** (Convergence). *Let Assumptions 3.7-(ii) and 3.8-(i) hold. Then, the d-CDP operator (6) is $\gamma$-contractive in the infinity-norm.*

The preceding theorem implies that the approximate ConjVI Algorithm 1 is indeed convergent given that the required conditions are satisfied. In particular, for deterministic dynamics, $\mathrm{co}(\mathbb{Z}^g) \supseteq f_s(\mathbb{X}^d)$ is sufficient for Algorithm 1 to be convergent. We next consider the complexity of our algorithm.

**Theorem 3.10** (Complexity). *Let Assumption 3.7-(iii) hold. Also assume that each evaluation of the extension operator $\widetilde{[\cdot]}$ in (6a) requires $\mathcal{O}(E)$ operations. Then, the time complexities of initialization and each iteration in Algorithm 1 are of $\mathcal{O}(X + U)$ and $\widetilde{\mathcal{O}}(XWE)$, respectively.*

The requirements of Assumption 3.7-(iii) will be discussed in Section 3.4. Recall that each iteration of VI (in primal domain) has a complexity of $\mathcal{O}(XUWE)$, where $E$ denotes the complexity of the extension operation used in (2). This observation points to a basic characteristic of the proposed approach: ConjVI reduces the quadratic complexity of VI to a linear one by replacing the minimization operation in the primal domain with a simple addition in the conjugate domain. Hence, for problem class of Assumption 3.1, ConjVI is expected to lead to a reduction in the computational cost. We note that ConjVI, like VI and other approximation schemes that utilize discretization/abstraction of the continuous state and input spaces, still suffers from the so-called "curse of dimensionality." This is because the sizes $X$ and $U$ of the discretizations increase exponentially with the dimensions $n$ and $m$ of the corresponding spaces. However, for ConjVI, this exponential increase is of rate $\max\{m, n\}$, compared to the rate $m + n$ for VI.

**Theorem 3.11** (Error). *Let Assumptions 3.5, 3.7-(i,ii), and 3.8-(i) hold. Consider the true optimal value function $J_\star = \mathcal{T}J_\star : \mathbb{X} \to \mathbb{R}$ and its discretization $J_\star^d : \mathbb{X}^d \to \mathbb{R}$, and let Assumption 3.8-(ii) hold for $J_\star$. Also, let $\widehat{J}^d : \mathbb{X}^d \to \mathbb{R}$ be the output of Algorithm 1. Then, $\|\widehat{J}^d - J_\star^d\|_\infty \leq \frac{\gamma(e_e + e_t) + e_d}{1 - \gamma}$, where $e_d = e_u + e_v + e_x + e_y + e_z$, and*

$$e_u = c_u \cdot d_H(\mathbb{U}, \mathbb{U}^d), \ e_v = c_v \cdot d_H\left(\mathrm{co}(\mathbb{V}^g), \mathbb{V}^g\right), \ e_x = c_x \cdot d_H\left(\mathbb{X}, \mathbb{X}^d\right),$$
$$e_y = c_y \cdot \max_{x \in \mathbb{X}^d} d\left(\partial(J_\star - C_s)(x), \mathbb{Y}^g\right), \ e_z = c_z \cdot d_H\left(f_s(\mathbb{X}^d), \mathbb{Z}^g\right), \tag{7}$$

*with constants $c_u, c_v, c_x, c_y, c_z > 0$ depending on the problem data.*

Let us first note that Assumption 3.5 implies that the DP and CDP operators preserve convexity, and they both have the true optimal value function $J_\star$ as their fixed point (i.e., the duality gap is zero). Otherwise, the proposed scheme can suffer from large errors due to dualization. Moreover, Assumptions 3.7-(i,ii) on the grids $\mathbb{V}^g$ and $\mathbb{Z}^g$ are required for bounding the error of approximate discrete conjugations using LERP in (6d) and (6f); see the proof of Lemmas A.5 and A.7 of the supplementary material. The remaining sources of error in the proposed approximate implementation of ConjVI are captured by the three main error terms: **(i)** $e_e$ is due to the approximation of the value function using the extension operator $\widetilde{[\cdot]}$; **(ii)** $e_t$ corresponds to the termination of the algorithm after a finite number of iterations; **(iii)** $e_d$ captures the error due to the discretization of the primal and dual state and input domains.

## 3.4 Construction of the grids

In this subsection, we provide specific guidelines for the construction of the grids $\mathbb{Y}^g$, $\mathbb{V}^g$ and $\mathbb{Z}^g$. We note that these discrete sets must be *grid-like* since they form the dual grid for the three conjugate transforms that are handled using LLT. The presented guidelines aim to minimize the error terms in (7) while taking into account the properties laid out in Assumption 3.7. In particular, the schemes described below satisfy the requirements of Assumption 3.7-(iii).

**Construction of $\mathbb{V}^{\mathrm{g}}$.** Assumption 3.7-(i) and the error term $e_{\mathrm{v}}$ in (7) suggest that we find the smallest input dual grid $\mathbb{V}^{\mathrm{g}}$ such that $\mathrm{co}(\mathbb{V}^{\mathrm{g}}_{\mathrm{sub}}) \supseteq \mathbb{L}(C^{\mathrm{d}}_{\mathrm{i}})$. This latter condition essentially means that $\mathbb{V}^{\mathrm{g}}$ must "more than cover the range of slope" of the function $C^{\mathrm{d}}_{\mathrm{i}}$; recall that $\mathbb{L}(C^{\mathrm{d}}_{\mathrm{i}}) = \Pi^m_{j=1} \left[ \mathrm{L}^-_j(C^{\mathrm{d}}_{\mathrm{i}}), \mathrm{L}^-_j(C^{\mathrm{d}}_{\mathrm{i}}) \right]$, where $\mathrm{L}^-_j(C^{\mathrm{d}}_{\mathrm{i}})$ (resp. $\mathrm{L}^+_j(C^{\mathrm{d}}_{\mathrm{i}})$) is the minimum (resp. maximum) slope of $C^{\mathrm{d}}_{\mathrm{i}}$ along the $j$-th dimension. Hence, we need to compute/approximate $\mathrm{L}^\pm_j(C^{\mathrm{d}}_{\mathrm{i}})$ for $j = 1, \ldots, m$. A conservative approximation is $\mathrm{L}^-_j(C_{\mathrm{i}}) = \min \partial C_{\mathrm{i}}/\partial u_j$ and $\mathrm{L}^+_j(C_{\mathrm{i}}) = \max \partial C_{\mathrm{i}}/\partial u_j$, assuming $C_{\mathrm{i}}$ is differentiable. Alternatively, we can directly use the discrete input cost $C^{\mathrm{d}}_{\mathrm{i}}$ for computing $\mathrm{L}^\pm_j(C^{\mathrm{d}}_{\mathrm{i}})$. In particular, if the domain $\mathbb{U}^{\mathrm{d}} = \mathbb{U}^{\mathrm{g}} = \Pi^m_{j=1} \mathbb{U}^{\mathrm{g}}_j$ of $C^{\mathrm{d}}_{\mathrm{i}}$ is grid-like and $C_{\mathrm{i}}$ is convex, we can take $\mathrm{L}^-_j(C^{\mathrm{d}}_{\mathrm{i}})$ (resp. $\mathrm{L}^+_j(C^{\mathrm{d}}_{\mathrm{i}})$) to be the minimum first forward difference (resp. maximum last backward difference) of $C^{\mathrm{d}}_{\mathrm{i}}$ along the $j$-th dimension (this scheme requires $\mathcal{O}(U)$ operations). Having $\mathrm{L}^\pm_j(C^{\mathrm{d}}_{\mathrm{i}})$ at our disposal, we can then construct $\mathbb{V}^{\mathrm{g}}_{\mathrm{sub}} = \Pi^m_{j=1} \mathbb{V}^{\mathrm{g}}_{\mathrm{sub}\,j}$ such that, in each dimension $j$, $\mathbb{V}^{\mathrm{g}}_{\mathrm{sub}\,j}$ is uniform and has the same cardinality as $\mathbb{U}^{\mathrm{g}}_j$, and $\mathrm{co}(\mathbb{V}^{\mathrm{g}}_{\mathrm{sub}\,j}) = \left[ \mathrm{L}^-_j(C^{\mathrm{d}}_{\mathrm{i}}), \mathrm{L}^+_j(C^{\mathrm{d}}_{\mathrm{i}}) \right]$. Finally, we construct $\mathbb{V}^{\mathrm{g}}$ by extending $\mathbb{V}^{\mathrm{g}}_{\mathrm{sub}}$ uniformly in each dimension (by adding a smaller and a larger element to $\mathbb{V}^{\mathrm{g}}_{\mathrm{sub}}$ in each dimension).

**Construction of $\mathbb{Z}^{\mathrm{g}}$.** According to Assumption 3.7-(ii), the grid $\mathbb{Z}^{\mathrm{g}}$ must be constructed such that $\mathrm{co}(\mathbb{Z}^{\mathrm{g}}) \supseteq f_{\mathrm{s}}(\mathbb{X}^{\mathrm{d}})$. This can be simply done by finding the vertices of the smallest box that contains the set $f_{\mathrm{s}}(\mathbb{X}^{\mathrm{d}})$. Those vertices give the diameter of $\mathbb{Z}^{\mathrm{g}}$ in each dimension. We can then, for example, take $\mathbb{Z}^{\mathrm{g}}$ to be the uniform grid with the same cardinality as $\mathbb{Y}^{\mathrm{g}}$ in each dimension (so that $Z = Y$). This way, $\mathrm{d}_{\mathrm{H}}\left( f_{\mathrm{s}}(\mathbb{X}^{\mathrm{d}}), \mathbb{Z}^{\mathrm{g}} \right) \le \mathrm{d}_{\mathrm{H}}\left( \mathrm{co}(\mathbb{Z}^{\mathrm{g}}), \mathbb{Z}^{\mathrm{g}} \right)$, and hence $e_{\mathrm{z}}$ in (7) reduces by using finer grids $\mathbb{Z}^{\mathrm{g}}$. This construction has a time complexity of $\mathcal{O}(X)$.

**Construction of $\mathbb{Y}^{\mathrm{g}}$.** Construction of the state dual grid $\mathbb{Y}^{\mathrm{g}}$ is more involved. According to Theorem 3.11, we need to choose a grid that minimizes $e_{\mathrm{y}}$ in (7). This can be done by choosing $\mathbb{Y}^{\mathrm{g}}$ such that $\mathbb{Y}^{\mathrm{g}} \cap \partial(J_\star - C_{\mathrm{s}}) \ne \emptyset$ for all $x \in \mathbb{X}^{\mathrm{d}}$ so that $e_{\mathrm{y}} = 0$. Even if we had access to the optimal value function $J_\star$, satisfying such a condition could lead to a dual grid $\mathbb{Y}^{\mathrm{g}} \subset \mathbb{R}^n$ of size $\mathcal{O}(X^n)$. Such a large size violates Assumption 3.7-(iii) on the size of $\mathbb{Y}^{\mathrm{g}}$, and essentially renders the proposed algorithm impractical for dimensions $n \ge 2$. A more practical condition is $\mathrm{co}(\mathbb{Y}^{\mathrm{g}}) \cap \partial(J_\star - C_{\mathrm{s}}) \ne \emptyset$ for all $x \in \mathbb{X}^{\mathrm{d}}$ so that $\max_{x \in \mathbb{X}^{\mathrm{d}}} \mathrm{d}\left( \partial(J_\star - C_{\mathrm{s}})(x), \mathbb{Y}^{\mathrm{g}} \right) \le \mathrm{d}_{\mathrm{H}}\left( \mathrm{co}(\mathbb{Y}^{\mathrm{g}}), \mathbb{Y}^{\mathrm{g}} \right)$, and hence $e_{\mathrm{y}}$ reduces by using a finer grid $\mathbb{Y}^{\mathrm{g}}$. The latter condition is satisfied if $\mathrm{co}(\mathbb{Y}^{\mathrm{g}}) \supseteq \mathbb{L}(J_\star - C_{\mathrm{s}})$, i.e., if $\mathbb{Y}^{\mathrm{g}}$ "covers the range of slops" of $(J_\star - C_{\mathrm{s}})$. Hence, we need to approximate the range of slopes of $(J_\star - C_{\mathrm{s}})$. To this end, we first use the fact that $J_\star$ is the fixed point of DP operator (1) to approximate $\mathrm{rng}(J_\star - C_{\mathrm{s}})$ by $R = \frac{\mathrm{rng}(C^{\mathrm{d}}_{\mathrm{i}}) + \gamma \cdot \mathrm{rng}(C^{\mathrm{d}}_{\mathrm{s}})}{1 - \gamma}$. We then construct the gird $\mathbb{Y}^{\mathrm{g}} = \Pi^n_{i=1} \mathbb{Y}^{\mathrm{g}}_i$ such that, for each dimension $i$, we have $\pm \alpha R / \Delta^i_{\mathbb{X}^{\mathrm{d}}} \in \mathrm{co}(\mathbb{Y}^{\mathrm{g}}_i)$, where $\Delta^i_{\mathbb{X}^{\mathrm{d}}}$ denotes the diameter of the projection of $\mathbb{X}^{\mathrm{d}}$ on the $i$-th dimension, and $\alpha > 0$ is a scaling factor mainly depending on the dimension of the state space. This construction has a one-time computational cost of $\mathcal{O}(X + U)$.

**Dynamic construction of $\mathbb{Y}^{\mathrm{g}}$.** Alternatively, we can construct $\mathbb{Y}^{\mathrm{g}}$ *dynamically* at each iteration in order to minimize the corresponding error in each application of the d-CDP operator given by $e_{\mathrm{y}} = c_{\mathrm{y}} \cdot \max_{x \in \mathbb{X}^{\mathrm{d}}} \mathrm{d}\left( \partial(\mathcal{T}J - C_{\mathrm{s}})(x), \mathbb{Y}^{\mathrm{g}} \right)$; see Lemma A.6 and Proposition A.8 of the supplementary material. This means that the construction of $\mathbb{Y}^{\mathrm{g}}$ in Algorithm 1 is moved from line 1 to inside the iterations, after line 5. Similar to the static scheme described above, the aim here is to construct $\mathbb{Y}^{\mathrm{g}}$ such that $\mathrm{co}(\mathbb{Y}^{\mathrm{g}}) \supseteq \mathbb{L}(\mathcal{T}J - C_{\mathrm{s}})$. Since we do not have access to $\mathcal{T}J$ (it is the output of the current iteration), we can again use the definition of the DP operator (1) to approximate $\mathrm{rng}(\mathcal{T}J - C_{\mathrm{s}})$ by $R = \mathrm{rng}(C^{\mathrm{d}}_{\mathrm{i}}) + \gamma \cdot \mathrm{rng}(J^{\mathrm{d}})$ where $J^{\mathrm{d}}$ is the output of the previous iteration. We then construct the gird $\mathbb{Y}^{\mathrm{g}} = \Pi^n_{i=1} \mathbb{Y}^{\mathrm{g}}_i$ such that, for each dimension $i$, we again have $\pm \alpha R / \Delta^i_{\mathbb{X}^{\mathrm{d}}} \in \mathrm{co}(\mathbb{Y}^{\mathrm{g}}_i)$. This construction has a one-time computational cost of $\mathcal{O}(U)$ for computing $\mathrm{rng}(C^{\mathrm{d}}_{\mathrm{i}})$ and a per iteration computational cost of $\mathcal{O}(X)$ for computing $\mathrm{rng}(J^{\mathrm{d}})$. Notice, however, that under this dynamic construction, the error bound of Theorem 3.11 does not hold true. More importantly, with a dynamic grid $\mathbb{Y}^{\mathrm{g}}$, there is no guarantee for ConjVI to converge.

## 4   Numerical simulations

We now showcase the application of the ConjVI algorithm through two numerical examples, and compare its performance with the VI algorithm. The details of these simulations (and another numerical example) are provided in Section 4 of the supplementary material. We also provide the

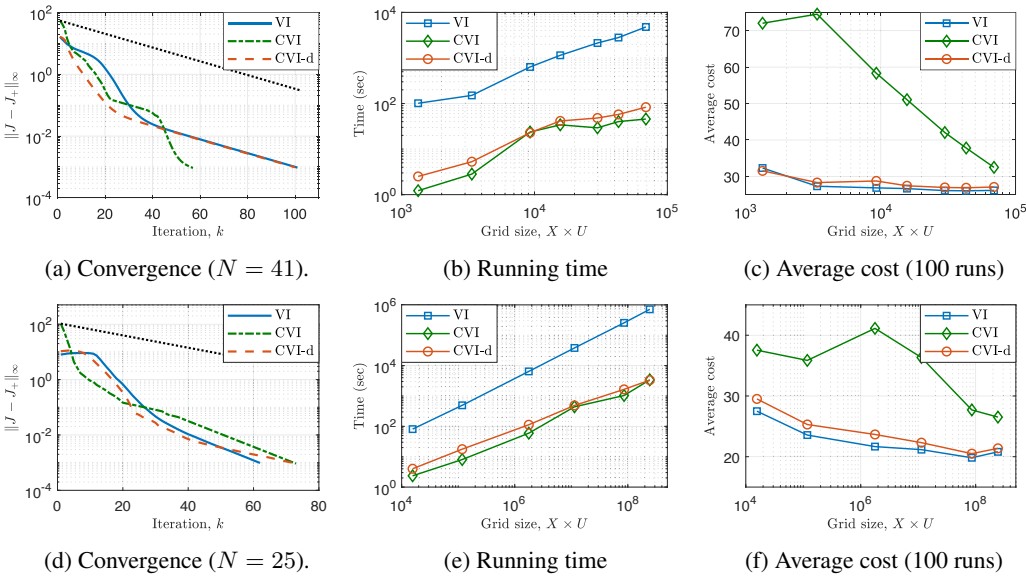

|  |  |  |
|---|---|---|
| (a) Convergence ($N = 41$). | (b) Running time | (c) Average cost (100 runs) |
| (d) Convergence ($N = 25$). | (e) Running time | (f) Average cost (100 runs) |

Figure 1: VI vs. ConjVI (CVI): optimal control of inverted pendulum (top) and batch reactor (bottom). The black dotted line in (a) corresponds to exponential convergence with rate $\gamma = 0.95$. CVI-d corresponds to *dynamic* construction of the dual grid $\mathbb{Y}^g$ in the ConjVI algorithm.

ConjVI MATLAB package [21] for the implementation of the proposed algorithm. The package also includes the numerical simulations of this section. We note that multiple routines in the developed package are borrowed from the d-CDP MATLAB package [22]. Also, for the discrete conjugation (LLT), we used the MATLAB package (in particular, the `LLTd` routine) provided in [23].

**Example 1.** We use the setup of [20, App. C.2.2] for the optimal control of a noisy inverted pendulum with two state variables and one input channel. In particular, we use *nearest neighbor* extension (which is non-expansive) for the extension operators in (2) for VI and in (6a) for ConjVI. We also consider the *dynamic* scheme for the construction of $\mathbb{Y}^g$ in ConjVI (hereafter, referred to as ConjVI-d). Moreover, in each implementation of VI and ConjVI(-d), all of the involved discrete sets are uniform grids with the same size $N$ in each dimension, i.e., $X, Y, Z = N^2$ and $U, V = N$. We are particularly interested in the performance of these algorithms, as the size of the discretizations increases. The results of our numerical simulations are shown in the top panels of Figure 1. As shown in Figures 1a, both VI and ConjVI algorithms are convergent with a rate $\leq \gamma = 0.95$. In particular, ConjVI terminates in 57 iterations, compared to 101 iterations required for VI to terminate. As expected, this faster convergence, combined with the lower time complexity of ConjVI in each iteration, leads to a significant reduction in the running time of this algorithm compared to VI; see Figure 1b. Since we do not have access to the true optimal value function, in order to evaluate the performance of the outputs of the these algorithms, we consider the performance of the greedy policy with respect to the discrete value function $J^d$ computed using these algorithms. Figure 1c reports the average cost of 100 random instances over $T = 100$ steps. As shown, the reduction in the running time in ConjVI comes with an increase in the cost of the controlled trajectories, particularly for coarse discretizations of the state-action space. On the other hand, using the *dynamic* scheme for construction of $\mathbb{Y}^g$, we see that ConjVI-d, with a small increase in the running time compared to ConjVI, achieves almost the same performance as VI concerning the quality of the greedy actions.

**Example 2.** We now consider the optimal control of an unstable *deterministic* batch reactor with four states and two inputs. The setup is borrowed from [19, Sec. 6]. Once again, all of the involved discrete sets are uniform grids with the same size $N$ in each dimension, i.e., $X, Y, Z = N^4$ and $U, V = N^2$. We note that we use *multi-linear interpolation and extrapolation* for the extension operator in (2) for VI. Due to the extrapolation, the extension operator is no longer non-expansive, and hence the convergence of VI is not guaranteed. On the other hand, since the dynamics is deterministic, there is no need for extension in ConjVI (the scaled expectation in (6a) in ConjVI reduces to the simple scaling $\varepsilon^d = \gamma \cdot J^d$ for deterministic dynamics), and hence the convergence of ConjVI only requires

$\mathrm{co}(\mathbb{Z}^g) \supseteq f_s(\mathbb{X}^g)$. The results of simulations are shown in the bottom panels of Figure 1. Once again, we see the trade-off between the time complexity and the greedy control performance in VI and ConjVI. On the other hand, ConjVI-d has the same control performance as VI with an insignificant increase in running time compared to ConjVI. Moreover, in Figure 1d, we observe the non-monotone behavior of ConjVI-d. In this regard, recall that when the grid $\mathbb{Y}^g$ is constructed dynamically and varies at each iteration, the d-CDP operator is not necessarily contractive, which is here the case for the first six iterations. The VI algorithm is also showing a non-monotone behavior, where for the first nine iterations the d-DP operation is actually expansive. As we noted earlier, this is because the multi-linear extrapolation operation used for extension is expansive.

# 5 Final remarks

In this paper, we proposed the ConjVI algorithm which reduces the time complexity of the VI algorithm from $\mathcal{O}(XU)$ to $\mathcal{O}(X+U)$. This better time complexity however comes at the expense of restricting the class of problem. In particular, there are two main conditions that must be satisfied in order to be able to apply the ConjVI algorithm: The dynamics must be of the form $x^+ = f_s(x) + Bu + w$, and the stage cost $C(x,u) = C_s(x) + C_i(u)$ must be separable. Moreover, since ConjVI essentially solves the dual problem, for non-convex problems, it suffers from a non-zero duality gap. Based on our simulation results, we also notice a trade-off between computational complexity and control action quality: While ConjVI has a lower computational cost, VI generates better control actions. However, the dynamic scheme for the construction of state dual grid $\mathbb{Y}^g$ allows us to achieve almost the same performance as VI when it comes to the quality of control actions, with a small extra computational burden. In what follows, we provide our final remarks on the limitations of the proposed ConjVI algorithm and its relation to existing approximate VI algorithms.

**Relation to existing approximate VI algorithms.** The basic idea for complexity reduction introduced in this study can be potentially combined with and further improve the existing sample-based VI algorithms. These sample-based algorithms solely focus on transforming the infinite-dimensional optimization in DP problems into computationally tractable ones, and in general, they have a time complexity of $\mathcal{O}(XU)$, depending on the product of the cardinalities of the discrete state and action spaces. The proposed ConjVI algorithm, on the other hand, focuses on reducing this time complexity to $\mathcal{O}(X+U)$, by avoiding the minimization over input in each iteration. Take, for example, the aggregation technique in [26, Sec. 8.1] that leads to a piece-wise constant approximation of the value function. It is straightforward to combine ConjVI with this type of state space aggregation. Indeed, the first numerical example of Section 4 (optimal control of inverted pendulum) essentially uses such an aggregation by approximating the value function via nearest neighbor extension.

**Cost functions with a large Lipschitz constant.** Recall that for the proposed ConjVI algorithm to be computationally efficient, the size $Y$ of the state dual grid $\mathbb{Y}^g$ must be controlled by the size $X$ of the discrete state space $\mathbb{X}^d$ (Assumption 3.7-(iii)). Then, as the range of slope of the value function $J_\star$ increases, the corresponding error $e_y$ in (7) due to discretization of the dual state space increases. The proposed dynamic approach for construction of $\mathbb{Y}^g$ partially addresses this issue by focusing on the range of slope of $J_k^d$ in each iteration in order to minimize the discretization error of the same iteration $k$. However, when the cost function has a large Lipschitz constant, even this latter approach can fail to provide a good approximation of the value function. (See the supplementary material for a numerical example).

**Gradient-based algorithms for solving the minimization over input.** Let us first note that the minimization over $u$ in sample-based VI algorithms usually involves solving a difficult non-convex problem. This is particularly due to that fact that the extension operation employed in these algorithms for approximating the value function using the sample points does not lead to a convex function in $u$ (e.g., take kernel-based approximations or neural networks). This is why in MDP and RL literature, it is actually quite common to consider a finite action space in the first place [10, 26]. Moreover, the minimization over $u$ again must be solved for each sample point in each iteration, while application of ConjVI avoids solving this minimization in each iteration. In this regard, let us note that ConjVI actually uses a convex approximation of the value function, which allows for application of a gradient-based algorithm for minimization over $u$ within the ConjVI algorithm. However, such an algorithm has a per iteration complexity of $\mathcal{O}(XY) = \mathcal{O}(X^2)$, which is practically inefficient.

## Acknowledgments

This research is part of a project that has received funding from the European Research Council (ERC) under the grant TRUST-949796. The authors are also grateful to anonymous reviewers for their comments concerning the three remarks in Section 5.

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
