**Paper organization.** The problem statement and its standard solution via the VI algorithm (in primal domain) are presented in Section 2. In Section 3, we present our main results: We begin with presenting the class of problems that are of interest, and then introduce the alternative approach for

VI in conjugate domain and its numerical implementation. The theoretical results on the convergence, complexity, and error of the proposed algorithm along with the guidelines on the construction of dual grids are also provided in this section. In Section 4, we compare the performance of the ConjVI with that of VI algorithm through three numerical examples. Section 5 concludes the paper with some final remarks. All the technical proofs are provided in Appendix A.

**Notations.** We use $\mathbb{R}$ and $\overline{\mathbb{R}} = \mathbb{R} \cup \{\infty\}$ to denote the real line and the extended reals, respectively, and $\mathbb{E}_w[\cdot]$ to denote expectation with respect (w.r.t.) to the random variable $w$. The standard inner product in $\mathbb{R}^n$ and the corresponding induced 2-norm are denoted by $\langle \cdot, \cdot \rangle$ and $\|\cdot\|_2$, respectively. We also use $\|\cdot\|_2$ to denote the operator norm (w.r.t. the 2-norm) of a matrix; i.e., for $A \in \mathbb{R}^{m \times n}$, we denote $\|A\|_2 = \sup\{\|Ax\|_2 : \|x\|_2 = 1\}$. The infinity-norm is denoted by $\|\cdot\|_\infty$.

Arbitrary sets (finite/infinite, countable/uncountable) are denoted as $\mathbb{X}, \mathbb{Y}, \ldots$. For *finite* (discrete) sets, we use the superscript d as in $\mathbb{X}^d, \mathbb{Y}^d, \ldots$ to differentiate them from infinite sets. Moreover, we use the superscript g to differentiate *grid-like* finite sets. Precisely, a grid $\mathbb{X}^g \subset \mathbb{R}^n$ is the Cartesian product $\mathbb{X}^g = \Pi_{i=1}^n \mathbb{X}_i^g = \mathbb{X}_1^g \times \ldots \times \mathbb{X}_n^g$, where $\mathbb{X}_i^g$ is a finite subset of $\mathbb{R}$. We also use $\mathbb{X}_{\text{sub}}^g$ to denote the *sub-grid* of $\mathbb{X}^g$ derived by omitting the smallest and the largest elements of $\mathbb{X}^g$ in each dimension. The cardinality of a finite set $\mathbb{X}^d$ or $\mathbb{X}^g$ is denoted by $X$. Let $\mathbb{X}, \mathbb{Y}$ be two arbitrary sets in $\mathbb{R}^n$. The convex hull of $\mathbb{X}$ is denoted by $\text{co}(\mathbb{X})$. The diameter of $\mathbb{X}$ is defined as $\Delta_{\mathbb{X}} := \sup_{x, \tilde{x} \in \mathbb{X}} \|x - \tilde{x}\|_2$. We use $\text{d}(\mathbb{X}, \mathbb{Y}) := \inf_{x \in \mathbb{X}, y \in \mathbb{Y}} \|x - y\|_2$ to denote the distance between $\mathbb{X}$ and $\mathbb{Y}$. The one-sided Hausdorff distance *from* $\mathbb{X}$ *to* $\mathbb{Y}$ is defined as $\text{d}_{\text{H}}(\mathbb{X}, \mathbb{Y}) := \sup_{x \in \mathbb{X}} \inf_{y \in \mathbb{Y}} \|x - y\|_2$.

Let $h : \mathbb{R}^n \to \overline{\mathbb{R}}$ be an extended real-valued function with a non-empty effective domain $\text{dom}(h) = \mathbb{X} := \{x \in \mathbb{R}^n : h(x) < \infty\}$, and range $\text{rng}(h) = \max_{x \in \mathbb{X}} h(x) - \min_{x \in \mathbb{X}} h(x)$. We use $h^d : \mathbb{X}^d \to \overline{\mathbb{R}}$ to denote the discretization of $h$, where $\mathbb{X}^d$ is a finite subset of $\mathbb{R}^n$. Whether a function is discrete is usually also clarified by providing its domain explicitly. We particularly use this notation in combination with a second operation to emphasize that the second operation is applied on the discretized version of the operand. E.g., we use $\widetilde{h^d} : \mathbb{R}^n \to \overline{\mathbb{R}}$ to denote a *generic extension* of $h^d$. If the domain $\mathbb{X}^d = \mathbb{X}^g$ of $h^d$ is grid-like, we then use $\overline{h^d}$ (as opposed to $\widetilde{h^d}$) for the extension using *multi-linear interpolation and extrapolation (LERP)*. The Lipschtiz constant of $h$ over a set $\mathbb{Y} \subset \text{dom}(h)$ is denoted by $\text{L}(h; \mathbb{Y}) := \sup_{x, y \in \mathbb{Y}} |h(x) - h(y)| / \|x - y\|_2$. We also denote $\text{L}(h) := \text{L}(h; \text{dom}(h))$ and $\mathbb{L}(h) := \Pi_{i=1}^n [\text{L}_i^-(h), \text{L}_i^+(h)]$, where $\text{L}_i^+(h)$ (resp. $\text{L}_i^-(h)$) is the maximum (resp. minimum) slope of the function $h$ along the $i$-th dimension, The subdifferential of $h$ at a point $x \in \mathbb{X}$ is defined as $\partial h(x) := \{y \in \mathbb{R}^n : h(\tilde{x}) \geq h(x) + \langle y, \tilde{x} - x \rangle, \forall \tilde{x} \in \mathbb{X}\}$. Note that $\partial h(x) \subseteq \mathbb{L}(h)$ for all $x \in \mathbb{X}$; in particular, $\mathbb{L}(h) = \cup_{x \in \mathbb{X}} \partial h(x)$ if $h$ is convex. The Legendre-Fenchel transform (convex conjugate) of $h$ is the function $h^* : \mathbb{R}^n \to \overline{\mathbb{R}}$, defined by $h^*(y) = \sup_x \{\langle y, x \rangle - h(x)\}$. We note that the conjugate function $h^*$ is convex by construction. We again use the notation $h^{d*}$ to emphasize the fact that the domain of the underlying function is *finite*, that is, $h^{d*}(y) = \sup_{x \in \mathbb{X}^d} \{\langle y, x \rangle - h(x)\}$. The biconjugate and discrete biconjugate operators are defined accordingly and denoted by $[\cdot]^{**} = [[\cdot]^*]^*$ and $[\cdot]^{d*d*} = [[\cdot]^{d*}]^{d*}$, respectively.

We report the complexities using the standard big-O notations $\mathcal{O}$ and $\widetilde{\mathcal{O}}$, where the latter hides the logarithmic factors. In this study, we are mainly concerned with the dependence of the computational complexities on *the size of the finite sets* involved (discretization of the primal and dual domains). In particular, we ignore the possible dependence of the computational complexities on the dimension of the variables, unless they appear in the power of the size of those discrete sets; e.g., the complexity of a single evaluation of an analytically available function is taken to be of $\mathcal{O}(1)$, regardless of the dimension of its input and output arguments.

## 2 VI in primal domain

We are concerned with the infinite-horizon, discounted cost, optimal control problems of the form

$$J_\star(x) = \min \ \mathbb{E}_{w_t} \left[ \sum_{t=0}^\infty \gamma^t C(x_t, u_t) \Big| x_0 = x \right]$$

$$\text{s.t.} \quad x_{t+1} = g(x_t, u_t, w_t), \ x_t \in \mathbb{X}, \ u_t \in \mathbb{U}, \ w_t \sim \mathbb{P}(\mathbb{W}), \quad \forall t \in \{0, 1, \ldots\},$$

where $x_t \in \mathbb{R}^n$, $u_t \in \mathbb{R}^m$, and $w_t \in \mathbb{R}^l$ are the state, input and disturbance variables at time $t$, respectively; $\gamma \in (0, 1)$ is the discount factor; $C : \mathbb{X} \times \mathbb{U} \to \mathbb{R}$ is the stage cost; $g : \mathbb{R}^n \times \mathbb{R}^m \times$

$\mathbb{R}^l \to \mathbb{R}^n$ describes the dynamics; $\mathbb{X} \subset \mathbb{R}^n$ and $\mathbb{U} \subset \mathbb{R}^m$ describe the state and input constraints, respectively; and, $\mathbb{P}(\cdot)$ is the distribution of the disturbance over the support $\mathbb{W} \subset \mathbb{R}^l$. Assuming the stage cost $C$ is bounded, the optimal value function solves the Bellman equation $J_\star = \mathcal{T} J_\star$, where $\mathcal{T}$ is the DP operator ($C$ and $J$ are extended to infinity outside their effective domains) [8, Prop. 1.2.2]

$$\mathcal{T} J(x) := \min_u \left\{ C(x, u) + \gamma \cdot \mathbb{E}_w J\big(g(x, u, w)\big) \right\}, \quad \forall x \in \mathbb{X}. \tag{1}$$

Indeed, $\mathcal{T}$ is $\gamma$-contractive in the infinity-norm, i.e., $\|\mathcal{T} J_1 - \mathcal{T} J_2\|_\infty \leq \gamma \|J_1 - J_2\|_\infty$ [8, Prop. 1.2.4]. This property then gives rise to the VI algorithm $J_{k+1} = \mathcal{T} J_k$ which converges to $J_\star$ as $k \to \infty$, for arbitrary initialization $J_0$. Moreover, assuming that the composition $J \circ g$ (for each $w$) and the cost $C$ are jointly convex in the state and input variables, $\mathcal{T}$ also preserves convexity [9, Prop. 3.3.1].

For numerical implementation of VI, we need to address three issues. First, we need to compute the expectation in (1). In order to simplify the exposition and include the computational cost of this operation explicitly, we consider disturbances with finite support in this study:

**Assumption 2.1** (Disturbance with finite support)**.** *The disturbance $w$ has a finite support $\mathbb{W}^d \subset \mathbb{R}^l$ with a given probability mass function (p.m.f.) $p : \mathbb{W}^d \to [0, 1]$.*

Under the preceding assumption, we have $\mathbb{E}_w J\big(g(x, u, w)\big) = \sum_{w \in \mathbb{W}^d} p(w) \cdot J\big(g(x, u, w)\big)$.[1] The second and more important issue is that the optimization problem (1) is infinite-dimensional for the continuous state space $\mathbb{X}$. This renders the exact implementation of VI impossible, except for a few cases with available closed-form solutions. A common solution to this problem is to deploy a sample-based approach, accompanied by a function approximation scheme. To be precise, for a finite subset $\mathbb{X}^d$ of $\mathbb{X}$, at each iteration $k = 0, 1, \ldots$, we take the discrete function $J_k^d : \mathbb{X}^d \to \mathbb{R}$ as the input, and compute the discrete function $J_{k+1}^d = \left[ \mathcal{T} \widetilde{J_k^d} \right]^d : \mathbb{X}^d \to \mathbb{R}$, where $\widetilde{J_k^d} : \mathbb{X} \to \mathbb{R}$ is an extension of $J_k^d$.[2] Finally, for each $x \in \mathbb{X}^d$, we have to solve the minimization problem in (1) over the control input. Since this minimization problem is often a difficult, non-convex problem, a common approximation again involves enumeration over a discretization $\mathbb{U}^d$ of the input space $\mathbb{U}$.

Incorporating these approximations, we end up with the approximate VI algorithm $J_{k+1}^d = \mathcal{T}^d J_k^d$, characterized by the *discrete* DP (d-DP) operator

$$\mathcal{T}^d J^d(x) := \min_{u \in \mathbb{U}^d} \left\{ C(x, u) + \gamma \cdot \sum_{w \in \mathbb{W}^d} p(w) \cdot \widetilde{J^d}\big(g(x, u, w)\big) \right\}, \quad \forall x \in \mathbb{X}^d. \tag{2}$$

The convergence of approximate VI described above depends on the properties of the extension operation $\widetilde{[\cdot]}$. In particular, if $\widetilde{[\cdot]}$ is non-expansive (in the infinity-norm), then $\mathcal{T}^d$ is also $\gamma$-contractive. For example, for a grid-like discretization of the state space $\mathbb{X}^d = \mathbb{X}^g$, the extension using *interpolative* LERP is non-expansive; see Lemma A.2. The error of this approximation ($\lim \left\| J_k^d - J_\star^d \right\|_\infty$) also depends on the extension operation $\widetilde{[\cdot]}$ and its representative power. We refer the interested reader to [8, 11, 27] for detailed discussions on the convergence and error of different approximations of VI.

The d-DP operator and the corresponding approximate VI algorithm will be our benchmark for evaluating the performance of the alternative algorithm developed in this study. To this end, we finish this section with some remarks on the time complexity of the d-DP operation. Let the time complexity of a single evaluation of the extension operator $\widetilde{[\cdot]}$ in (2) be of $\mathcal{O}(E)$.[3] Then, the time complexity of the d-DP operation (2) is of $\mathcal{O}\big(XUWE\big)$. In this regard, note that the scheme described above essentially involves approximating a continuous-state/action MDP with a finite-state/action MDP, and then applying VI. This, in turn, implies the lower bound $\Omega(XU)$ for the time complexity

---

[1] Indeed, $\mathbb{W}^d$ can be considered as a finite approximation of the true support $\mathbb{W}$ of the disturbance. Moreover, one can consider other approximation schemes, such as Monte Carlo simulation, for this expectation operation.

[2] The extension can be considered as a generic parametric approximation $\widehat{J}_{\theta_k} : \mathbb{X} \to \mathbb{R}$, where the parameters $\theta_k$ are computed using regression, i.e., by fitting $\widehat{J}_{\theta_k}$ to the data points $J_k^d : \mathbb{X}^d \to \mathbb{R}$.

[3] For example, for the linear approximation $\widetilde{J^d}(x) = \sum_{i=1}^B \alpha_i \cdot b_i(x)$, we have $E = B$ (the size of the basis), while for the kernel-based approximation $\widetilde{J^d}(x) = \sum_{\bar{x} \in \mathbb{X}^d} \alpha_{\bar{x}} \cdot r(x, \bar{x})$, we generally have $E \leq X$. In particular, if $\mathbb{X}^d = \mathbb{X}^g$ is grid-like, and $\widetilde{J^d} = \overline{J^d}$ is approximated using LERP, then $E = \log X$ [21, Rem. 2.2].

(corresponding to enumeration over $u \in \mathbb{U}^\mathrm{d}$ for each $x \in \mathbb{X}^\mathrm{d}$). This lower bound is also compatible with the best existing time complexities in the literature for VI for finite MDPs; see, e.g., [3, 28]. However, as we will see in the next section, for a particular class of problems, it is possible to exploit the structure of the underlying continuous system in order to achieve a better time complexity in the corresponding discretized problem.

## 3 Reducing complexity via conjugate duality

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

**Assumption 3.6** (Feasibile discretization). *The set $\mathbb{U}(x)$ is nonempty for all $x \in \mathbb{X}^{\mathrm{d}}$.*

In what follows, we describe the main steps within the initialization and iterations of Algorithm 1. In particular, the conjugate operations in (4) are handled numerically via the linear-time Legendre transform (LLT) algorithm [24]. LLT is an efficient algorithm for computing the *discrete* conjugate function over a finite *grid-like* dual domain. Precisely, to compute the conjugate of the function $h : \mathbb{X} \to \mathbb{R}$, LLT takes its discretization $h^{\mathrm{d}} : \mathbb{X}^{\mathrm{d}} \to \mathbb{R}$ as an input, and outputs $h^{\mathrm{d}*\mathrm{d}} : \mathbb{Y}^{\mathrm{g}} \to \mathbb{R}$, for the grid-like dual domain $\mathbb{Y}^{\mathrm{g}}$. We refer the reader to [24] for a detailed description of LLT. The main steps of the proposed approximate implementation of the CDP operator (4) are as follows:

(i) For the expectation operation in (4a), by Assumption 2.1, we again have

$$\mathbb{E}_w J(\cdot + w) = \sum_{w \in \mathbb{W}^{\mathrm{d}}} p(w) \cdot J(\cdot + w).$$

---
**Algorithm 1** ConjVI: Approximate VI in conjugate domain
---
**Input:** dynamics $f_s : \mathbb{R}^n \to \mathbb{R}^n$, $B \in \mathbb{R}^{n \times m}$; finite state space $\mathbb{X}^d \subset \mathbb{X}$; finite input space $\mathbb{U}^d \subset \mathbb{U}$; state cost function $C_s^d : \mathbb{X}^d \to \mathbb{R}$; input cost function $C_i^d : \mathbb{U}^d \to \mathbb{R}$; finite disturbance space $\mathbb{W}^d$ and its p.m.f. $p : \mathbb{W}^d \to [0, 1]$; discount factor $\gamma$; termination bound $e_t$.

**Output:** discrete value function $\widehat{J}^d : \mathbb{X}^d \to \mathbb{R}$.
  *initialization:*
1: construct the grid $\mathbb{V}^g$;
2: use LLT to compute $C_i^{d*d} : \mathbb{V}^g \to \mathbb{R}$ from $C_i^d : \mathbb{U}^d \to \mathbb{R}$;
3: construct the grid $\mathbb{Z}^g$;
4: construct the grid $\mathbb{Y}^g$;
5: $J^d(x) \leftarrow 0$ for $x \in \mathbb{X}^d$;
6: $J_+^d(x) \leftarrow C_s^d(x) - \min C_i^d$ for $x \in \mathbb{X}^d$;
  *iteration:*
7: **while** $\left\| J_+^d - J^d \right\|_\infty \geq e_t$ **do**
8:  $J^d \leftarrow J_+^d$;
   *d-CDP operation:*
9:  $\varepsilon^d(x) \leftarrow \gamma \cdot \sum_{w \in \mathbb{W}^d} p(w) \cdot \widetilde{J^d}(x + w)$ for $x \in \mathbb{X}^d$;
10:  use LLT to compute $\varepsilon^{d*d} : \mathbb{Y}^g \to \overline{\mathbb{R}}$ from $\varepsilon^d : \mathbb{X}^d \to \mathbb{R}$;
11:  **for** each $y \in \mathbb{Y}^g$ **do**
12:   use LERP to compute $\overline{C_i^{d*d}}(-B^\top y)$ from $C_i^{d*d} : \mathbb{V}^g \to \mathbb{R}$;
13:   $\varphi^d(y) \leftarrow \overline{C_i^{d*d}}(-B^\top y) + \varepsilon^{d*d}(y)$;
14:  **end for**
15:  use LLT to compute $\varphi^{d*d} : \mathbb{Z}^g \to \mathbb{R}$ from $\varphi^d : \mathbb{Y}^g \to \mathbb{R}$;
16:  **for** each $x \in \mathbb{X}^d$ **do**
17:   use LERP to compute $\overline{\varphi^{d*d}}(f_s(x))$ from $\varphi^{d*d} : \mathbb{Z}^g \to \mathbb{R}$;
18:   $J_+^d(x) \leftarrow C_s(x) + \overline{\varphi^{d*d}}(f_s(x))$;
19:  **end for**
20: **end while**
21: output $\widehat{J}^d \leftarrow J_+^d$.
---

Hence, we need to pass the value function $J^d : \mathbb{X}^d \to \mathbb{R}$ through the "scaled expection filter" to obtain $\varepsilon^d : \mathbb{X}^d \to \overline{\mathbb{R}}$ in (6a) as an approximation of $\epsilon$ in (4a). Notice that here we are using an extension $\widetilde{J^d} : \mathbb{X} \to \mathbb{R}$ of $J^d$ (recall that we only have access to the discrete value function $J^d$).

(ii) In order to compute $\phi$ in (4b), we need access to two conjugate functions:

   (a) For $\epsilon^*$, we use the approximation $\varepsilon^{d*d} : \mathbb{Y}^g \to \mathbb{R}$ in (6b), by applying LLT to the data points $\varepsilon^d : \mathbb{X}^d \to \overline{\mathbb{R}}$ for a properly chosen state dual grid $\mathbb{Y}^g \subset \mathbb{R}^n$.

   (b) If the conjugate $C_i^*$ of the input cost is not analytically available, we approximate it as follows: For a properly chosen input dual grid $\mathbb{V}^g \subset \mathbb{R}^m$, we employ LLT to compute $C_i^{d*d} : \mathbb{V}^g \to \mathbb{R}$ in (6c), using the data points $C_i^d : \mathbb{U}^d \to \mathbb{R}$, where $\mathbb{U}^d$ is a finite subset of $\mathbb{U}$.

   With these conjugate functions at hand, we can now compute $\varphi^d : \mathbb{Y}^g \to \mathbb{R}$ in (6d), as an approximation of $\phi$ in (4b). In particular, notice that we use the LERP extension $\overline{C_i^{d*d}}$ of $C_i^{d*d}$ to approximate $C_i^{d*}$ at the required point $-B^\top y$ for each $y \in \mathbb{Y}^g$.

(iii) To be able to compute the output according to (4c), we need to perform another conjugate transform. In particular, we need the value of $\phi^*$ at $f_s(x)$ for $x \in \mathbb{X}^d$. Here, we use the approximation $\varphi^{d*d} : \mathbb{Z}^g \to \mathbb{R}$ in (6e), by applying LLT to the data points $\varphi^d : \mathbb{Y}^g \to \mathbb{R}$ for a properly chosen grid $\mathbb{Z}^g \subset \mathbb{R}^n$. Finally, we use the LERP extension $\overline{\varphi^{d*d}}$ of $\varphi^{d*d}$ to approximate $\varphi^{d*}$ at the required point $f_s(x)$ for each $x \in \mathbb{X}^d$, and compute $\widehat{\mathcal{T}}^d J^d$ in (6f) as an approximation of $\widehat{\mathcal{T}} J$ in (4c).

With these approximations, we can introduce the *discrete* CDP (d-CDP) operator as follows

$$\varepsilon^{\mathrm{d}}(x) := \gamma \cdot \sum_{w \in \mathbb{W}^{\mathrm{d}}} p(w) \cdot \widetilde{J^{\mathrm{d}}}(x+w), \quad x \in \mathbb{X}^{\mathrm{d}}, \tag{6a}$$

$$\varepsilon^{\mathrm{d}*\mathrm{d}}(y) = \max_{x \in \mathbb{X}^{\mathrm{d}}} \left\{ \langle x, y \rangle - \varepsilon^{\mathrm{d}}(x) \right\}, \quad y \in \mathbb{Y}^{\mathrm{g}}, \tag{6b}$$

$$C_{\mathrm{i}}^{\mathrm{d}*\mathrm{d}}(v) = \max_{u \in \mathbb{U}^{\mathrm{d}}} \left\{ \langle u, v \rangle - C_{\mathrm{i}}^{\mathrm{d}}(u) \right\}, \quad v \in \mathbb{V}^{\mathrm{g}}, \tag{6c}$$

$$\varphi^{\mathrm{d}}(y) := \overline{C_{\mathrm{i}}^{\mathrm{d}*\mathrm{d}}}(-B^{\top}y) + \varepsilon^{\mathrm{d}*\mathrm{d}}(y), \quad y \in \mathbb{Y}^{\mathrm{g}}, \tag{6d}$$

$$\varphi^{\mathrm{d}*\mathrm{d}}(z) = \max_{y \in \mathbb{Y}^{\mathrm{g}}} \left\{ \langle y, z \rangle - \varphi^{\mathrm{d}}(y) \right\}, \quad z \in \mathbb{Z}^{\mathrm{g}}, \tag{6e}$$

$$\widehat{\mathcal{T}}^{\mathrm{d}} J^{\mathrm{d}}(x) := C_{\mathrm{s}}(x) + \overline{\varphi^{\mathrm{d}*\mathrm{d}}}\big(f_{\mathrm{s}}(x)\big), \quad x \in \mathbb{X}^{\mathrm{d}}. \tag{6f}$$

The proper construction of the grids $\mathbb{Y}^{\mathrm{g}}$, $\mathbb{V}^{\mathrm{g}}$, and $\mathbb{Z}^{\mathrm{g}}$ will be discussed in Section 3.4. We finish this subsection with the two following two remarks.

**Remark 3.7** (Deterministic systems)**.** *For deterministic systems, i.e., $g(x, u, w) = f(x, u)$, we do not need to compute any expectation. Then, the operation in (6a) becomes the simple scaling $\varepsilon^{\mathrm{d}} = \gamma \cdot J^{\mathrm{d}}$.*

**Remark 3.8** (Analytically available $C_{\mathrm{i}}^{*}$)**.** *If the conjugate $C_{\mathrm{i}}^{*}$ of the input cost is analytically available, we can use it directly in (6d) instead of $\overline{C_{\mathrm{i}}^{\mathrm{d}*\mathrm{d}}}$ and avoid the corresponding approximation; i.e., there is no need for construction of $\mathbb{V}^{\mathrm{g}}$ and the computation of $C_{\mathrm{i}}^{\mathrm{d}*\mathrm{d}}$ in (6c).*

### 3.3 Analysis of ConjVI algorithm

We now provide our main theoretical results concerning the convergence, complexity, and error of the proposed algorithm. Let us begin with presenting the assumptions to be called in this subsection.

**Assumption 3.9** (Grids)**.** *Consider the following properties for the grids in Algorithm 1 (consult the Notations in Section 1):*

*(i) The grid $\mathbb{V}^{\mathrm{g}}$ is constructed such that $\mathrm{co}(\mathbb{V}_{\mathrm{sub}}^{\mathrm{g}}) \supseteq \mathbb{L}(C_{\mathrm{i}}^{\mathrm{d}})$.*

*(ii) The grid $\mathbb{Z}^{\mathrm{g}}$ is constructed such that $\mathrm{co}(\mathbb{Z}^{\mathrm{g}}) \supseteq f_{\mathrm{s}}\big(\mathbb{X}^{\mathrm{d}}\big)$.*

*(iii) The construction of $\mathbb{Y}^{\mathrm{g}}$, $\mathbb{V}^{\mathrm{g}}$, and $\mathbb{Z}^{\mathrm{g}}$ requires at most $\mathcal{O}(X + U)$ operations. The cardinality of the grids $\mathbb{Y}^{\mathrm{g}}$ and $\mathbb{Z}^{\mathrm{g}}$ (resp. $\mathbb{V}^{\mathrm{g}}$) in each dimension is the same as that of $\mathbb{X}^{\mathrm{d}}$ (resp. $\mathbb{U}^{\mathrm{d}}$) in that dimension so that $Y, Z = X$ and $V = U$.*

**Assumption 3.10** (Extension operator)**.** *Consider the following properties for the extension operator $\widetilde{[\cdot]}$ in (6a):*

*(i) The extension operator is non-expansive w.r.t. the infinity norm; that is, for two discrete functions $J_{i}^{\mathrm{d}} : \mathbb{X}^{\mathrm{d}} \to \mathbb{R}$ $(i = 1, 2)$ and their extensions $\widetilde{J_{i}^{\mathrm{d}}} : \mathbb{X} \to \mathbb{R}$, we have $\|\widetilde{J_{1}^{\mathrm{d}}} - \widetilde{J_{2}^{\mathrm{d}}}\|_{\infty} \leq \|J_{1}^{\mathrm{d}} - J_{2}^{\mathrm{d}}\|_{\infty}$.*

*(ii) Given a function $J : \mathbb{X} \to \mathbb{R}$ and its discretization $J^{\mathrm{d}} : \mathbb{X}^{\mathrm{d}} \to \mathbb{R}$, the error of the extension operator is uniformly bounded, that is, $\|J - \widetilde{J^{\mathrm{d}}}\|_{\infty} \leq e_{\mathrm{e}}$ for some constant $e_{\mathrm{e}} \geq 0$.*

Our first result concerns the contractiveness of the d-CDP operator.

**Theorem 3.11** (Convergence)**.** *Let Assumptions 3.9-(ii) and 3.10-(i) hold. Then, the d-CDP operator (6) is $\gamma$-contractive w.r.t. the infinity-norm.*

The preceding theorem implies that the approximate ConjVI Algorithm 1 is indeed convergent given that the required conditions are satisfied. In particular, for deterministic dynamics, $\mathrm{co}(\mathbb{Z}^{\mathrm{g}}) \supseteq f_{\mathrm{s}}\big(\mathbb{X}^{\mathrm{d}}\big)$ is sufficient for Algorithm 1 to be convergent. We next consider the complexity of our algorithm.

**Theorem 3.12** (Complexity)**.** *Let Assumption 3.9-(iii) hold. Also assume that each evaluation of the extension operator $\widetilde{[\cdot]}$ in (6a) requires $\mathcal{O}(E)$ operations. Then, the time complexities of initialization and each iteration in Algorithm 1 are of $\mathcal{O}(X + U)$ and $\widetilde{\mathcal{O}}(XWE)$, respectively.*

The requirements of Assumption 3.9-(iii) will be discussed in Section 3.4. Recall that each iteration of VI (in primal domain) has a complexity of $\mathcal{O}(XUWE)$, where $E$ denotes the complexity of the extension operation used in (2). This observation points to a basic characteristic of the proposed approach: ConjVI reduces the quadratic complexity of VI to a linear one by replacing the minimization operation in the primal domain with a simple addition in the conjugate domain. Hence, for problem class of Assumption 3.1, ConjVI is expected to lead to a reduction in the computational cost. We note that ConjVI, like VI and other approximation schemes that utilize discretization/abstraction of the continuous state and input spaces, still suffers from the so-called "curse of dimensionality." This is because the sizes $X$ and $U$ of the discretizations increase exponentially with the dimensions $n$ and $m$ of the corresponding spaces. However, for ConjVI, this exponential increase is of rate $\max\{m, n\}$, compared to the rate $m + n$ for VI.

Let us also note that the most crucial step that allows the speedup discussed above is the *interpolative discrete conjugation* in (6f) that approximates $\varphi^{\mathrm{d}*\mathrm{d}}$ at the point $f_{\mathrm{s}}(x)$. In this regard, notice that we can alternatively compute $\varphi^{\mathrm{d}*\mathrm{d}}\big(f_{\mathrm{s}}(x)\big) = \max_{y \in \mathbb{Y}^{\mathrm{g}}} \big\{ \langle y, f_{\mathrm{s}}(x) \rangle - \varphi^{\mathrm{d}}(y) \big\}$ exactly via enumeration over $y \in \mathbb{Y}^{\mathrm{g}}$ for each $x \in \mathbb{X}^{\mathrm{d}}$ (then, the computation of $\varphi^{\mathrm{d}*\mathrm{d}} : \mathbb{Z}^{\mathrm{g}} \to \mathbb{R}$ in (6e) is not needed anymore). However, this approach requires $\mathcal{O}(XY) = \mathcal{O}(X^2)$ operations in the last step, hence rendering the proposed approach computationally impractical. Of course, the application of interpolative discrete conjugation has its cost: The LERP extension in (6f) can lead to non-convex outputs (even if Assumption 3.5 holds true). This, in turn, can introduce a dualization error. We finish with the following result on the error of the proposed ConjVI algorithm.

**Theorem 3.13** (Error). *Let Assumptions 3.5, 3.9-(i)&(ii), and 3.10-(i) hold. Consider the true optimal value function $J_\star = \mathcal{T} J_\star : \mathbb{X} \to \mathbb{R}$ and its discretization $J_\star^{\mathrm{d}} : \mathbb{X}^{\mathrm{d}} \to \mathbb{R}$, and let Assumption 3.10-(ii) hold for $J_\star$. Also, let $\widehat{J}^{\mathrm{d}} : \mathbb{X}^{\mathrm{d}} \to \mathbb{R}$ be the output of Algorithm 1. Then,*

$$\|\widehat{J}^{\mathrm{d}} - J_\star^{\mathrm{d}}\|_\infty \leq \frac{\gamma(e_{\mathrm{e}} + e_{\mathrm{t}}) + e_{\mathrm{d}}}{1 - \gamma}, \tag{7}$$

*where $e_{\mathrm{d}} = e_{\mathrm{u}} + e_{\mathrm{v}} + e_{\mathrm{x}} + e_{\mathrm{y}} + e_{\mathrm{z}}$, and*

$$e_{\mathrm{u}} = c_{\mathrm{u}} \cdot \mathrm{d}_{\mathrm{H}}(\mathbb{U}, \mathbb{U}^{\mathrm{d}}), \tag{8a}$$

$$e_{\mathrm{v}} = c_{\mathrm{v}} \cdot \mathrm{d}_{\mathrm{H}}\big(\mathrm{co}(\mathbb{V}^{\mathrm{g}}), \mathbb{V}^{\mathrm{g}}\big), \tag{8b}$$

$$e_{\mathrm{x}} = c_{\mathrm{x}} \cdot \mathrm{d}_{\mathrm{H}}\big(\mathbb{X}, \mathbb{X}^{\mathrm{d}}\big), \tag{8c}$$

$$e_{\mathrm{y}} = c_{\mathrm{y}} \cdot \max_{x \in \mathbb{X}^{\mathrm{d}}} \mathrm{d}\big(\partial(J_\star - C_{\mathrm{s}})(x), \mathbb{Y}^{\mathrm{g}}\big), \tag{8d}$$

$$e_{\mathrm{z}} = c_{\mathrm{z}} \cdot \mathrm{d}_{\mathrm{H}}\big(f_{\mathrm{s}}(\mathbb{X}^{\mathrm{d}}), \mathbb{Z}^{\mathrm{g}}\big), \tag{8e}$$

*with constants $c_{\mathrm{u}}, c_{\mathrm{v}}, c_{\mathrm{x}}, c_{\mathrm{y}}, c_{\mathrm{z}} > 0$ depending on the problem data.*

Let us first note that Assumption 3.5 implies that the DP and CDP operators preserve convexity, and they both have the true optimal value function $J_\star$ as their fixed point (i.e., the duality gap is zero). Otherwise, the proposed scheme can suffer from large errors due to dualization. Moreover, Assumptions 3.9-(i)&(ii) on the grids $\mathbb{V}^{\mathrm{g}}$ and $\mathbb{Z}^{\mathrm{g}}$ are required for bounding the error of approximate discrete conjugations using LERP in (6d) and (6f); see the proof of Lemmas A.5 and A.7. The remaining sources of error in the proposed approximate implementation of ConjVI are captured by the three error terms in (7):

(i) $e_{\mathrm{e}}$ is due to the approximation of the value function using the extension operator $\widetilde{[\cdot]}$;

(ii) $e_{\mathrm{t}}$ corresponds to the termination of the algorithm after a finite number of iterations;

(iii) $e_{\mathrm{d}}$ captures the error due to the discretization of the primal and dual state and input domains.

We again finish with the following remarks on the modification of the proposed algorithm for deterministic systems and analytically available $C_{\mathrm{i}}^*$.

**Remark 3.14** (Deterministic systems). *If the dynamics is deterministic, then the complexity of each iteration of Algorithm 1 reduces to $\widetilde{\mathcal{O}}(X)$. Moreover, in this case, the error term $e_{\mathrm{e}}$ disappears.*

**Remark 3.15** (Analytically available $C_{\mathrm{i}}^*$). *If the conjugate $C_{\mathrm{i}}^*$ of the input cost is analytically available and used in (6d) instead of the LERP extension $\overline{C_{\mathrm{i}}^{\mathrm{d}*\mathrm{d}}}$, the error term due to discretization modifies to $e_{\mathrm{d}} = e_{\mathrm{x}} + e_{\mathrm{y}} + e_{\mathrm{z}}$. That is, the error terms $e_{\mathrm{u}}$ and $e_{\mathrm{v}}$ corresponding to the discretization of the primal and dual input spaces disappear.*

## 3.4 Construction of the grids

In this subsection, we provide specific guidelines for the construction of the grids $\mathbb{Y}^g$, $\mathbb{V}^g$ and $\mathbb{Z}^g$. We note that these discrete sets must be *grid-like* since they form the dual grid for the three conjugate transforms that are handled using LLT. The presented guidelines aim to minimize the error terms in (8) while taking into account the properties laid out in Assumption 3.9. In particular, the schemes described below satisfy the requirements of Assumption 3.9-(iii).

**Construction of $\mathbb{V}^g$.** Assumption 3.9-(i) and the error term $e_v$ in (8b) suggest that we find the smallest input dual grid $\mathbb{V}^g$ such that $\mathrm{co}(\mathbb{V}^g_{\mathrm{sub}}) \supseteq \mathbb{L}(C_i^d)$. This latter condition essentially means that $\mathbb{V}^g$ must "more than cover the range of slope" of the function $C_i^d$; recall that $\mathbb{L}(C_i^d) = \Pi_{j=1}^m \left[ L_j^-(C_i^d), L_j^-(C_i^d) \right]$, where $L_j^-(C_i^d)$ (resp. $L_j^+(C_i^d)$) is the minimum (resp. maximum) slope of $C_i^d$ along the $j$-th dimension. Hence, we need to compute/approximate $L_j^\pm(C_i^d)$ for $j = 1, \ldots, m$. A conservative approximation is $L_j^-(C_i) = \min \partial C_i / \partial u_j$ and $L_j^+(C_i) = \max \partial C_i / \partial u_j$, assuming $C_i$ is differentiable. Alternatively, we can directly use the discrete input cost $C_i^d$ for computing $L_j^\pm(C_i^d)$. In particular, if the domain $\mathbb{U}^d = \mathbb{U}^g = \Pi_{j=1}^m \mathbb{U}_j^g$ of $C_i^d$ is grid-like and $C_i$ is convex, we can take $L_j^-(C_i^d)$ (resp. $L_j^+(C_i^d)$) to be the minimum first forward difference (resp. maximum last backward difference) of $C_i^d$ along the $j$-th dimension (this scheme requires $\mathcal{O}(U)$ operations). Having $L_j^\pm(C_i^d)$ at our disposal, we can then construct $\mathbb{V}^g_{\mathrm{sub}} = \Pi_{j=1}^m \mathbb{V}^g_{\mathrm{sub}\,j}$ such that, in each dimension $j$, $\mathbb{V}^g_{\mathrm{sub}\,j}$ is uniform and has the same cardinality as $\mathbb{U}_j^g$, and $\mathrm{co}(\mathbb{V}^g_{\mathrm{sub}\,j}) = \left[ L_j^-(C_i^d), L_j^+(C_i^d) \right]$. Finally, we construct $\mathbb{V}^g$ by extending $\mathbb{V}^g_{\mathrm{sub}}$ uniformly in each dimension (by adding a smaller and a larger element to $\mathbb{V}^g_{\mathrm{sub}}$ in each dimension while preserving the resolution in that dimension).

**Construction of $\mathbb{Z}^g$.** According to Assumption 3.9-(ii), the grid $\mathbb{Z}^g$ must be constructed such that $\mathrm{co}(\mathbb{Z}^g) \supseteq f_s(\mathbb{X}^d)$. This can be simply done by finding the vertices of the smallest box that contains the set $f_s(\mathbb{X}^d)$. Those vertices give the diameter of $\mathbb{Z}^g$ in each dimension. We can then, for example, take $\mathbb{Z}^g$ to be the uniform grid with the same cardinality as $\mathbb{Y}^g$ in each dimension (so that $Z = Y$). This way,

$$d_H\left( f_s(\mathbb{X}^d), \mathbb{Z}^g \right) \leq d_H\left( \mathrm{co}(\mathbb{Z}^g), \mathbb{Z}^g \right),$$

and hence $e_z$ in (8e) reduces by using finer grids $\mathbb{Z}^g$. This construction has a complexity of $\mathcal{O}(X)$.

**Construction of $\mathbb{Y}^g$.** Construction of the state dual grid $\mathbb{Y}^g$ is more involved. According to Theorem 3.13, we need to choose a grid that minimizes $e_y$ in (8d). This can be done by choosing $\mathbb{Y}^g$ such that $\mathbb{Y}^g \cap \partial(J_\star - C_s) \neq \emptyset$ for all $x \in \mathbb{X}^d$ so that $e_y = 0$. Even if we had access to the optimal value function $J_\star$, satisfying such a condition could lead to a dual grid $\mathbb{Y}^g \subset \mathbb{R}^n$ of size $\mathcal{O}(X^n)$. Such a large size violates Assumption 3.9-(iii) on the size of $\mathbb{Y}^g$, and essentially renders the proposed algorithm impractical for dimensions $n \geq 2$. A more practical condition is $\mathrm{co}(\mathbb{Y}^g) \cap \partial(J_\star - C_s) \neq \emptyset$ for all $x \in \mathbb{X}^d$ so that

$$\max_{x \in \mathbb{X}^d} d\left( \partial(J_\star - C_s)(x), \mathbb{Y}^g \right) \leq d_H\left( \mathrm{co}(\mathbb{Y}^g), \mathbb{Y}^g \right),$$

and hence $e_y$ reduces by using a finer grid $\mathbb{Y}^g$. The latter condition is satisfied if $\mathrm{co}(\mathbb{Y}^g) \supseteq \mathbb{L}(J_\star - C_s)$, i.e., if $\mathbb{Y}^g$ "covers the range of slope" of $(J_\star - C_s)$. Hence, we need to approximate the range of slope of $(J_\star - C_s)$. To this end, we first use the fact that $J_\star$ is the fixed point of DP operator (1) to approximate $\mathrm{rng}(J_\star - C_s)$ by

$$R = \frac{\mathrm{rng}(C_i^d) + \gamma \cdot \mathrm{rng}(C_s^d)}{1 - \gamma}.$$

We then construct the gird $\mathbb{Y}^g = \Pi_{i=1}^n \mathbb{Y}_i^g$ such that, for each dimension $i$, we have

$$\pm \frac{\alpha R}{\Delta_{\mathbb{X}^d}^i} \in \mathrm{co}(\mathbb{Y}_i^g) \tag{9}$$

where $\Delta_{\mathbb{X}^d}^i$ denotes the diameter of the projection of $\mathbb{X}^d$ on the $i$-th dimension. Here, the coefficient $\alpha > 0$ is a scaling factor mainly depending on the dimension of the state space. In particular, by setting $\alpha = 1$, the value $R/\Delta_{\mathbb{X}^d}^i$ is the slope of a linear function with range $R$ over the domain $\Delta_{\mathbb{X}^d}^i$. This construction has a one-time cost of $\mathcal{O}(X + U)$ for computing $\mathrm{rng}(C_i^d)$ and $\mathrm{rng}(C_s^d)$.

**Dynamic construction of $\mathbb{Y}^{\mathrm{g}}$.** Alternatively, we can construct $\mathbb{Y}^{\mathrm{g}}$ *dynamically* at each iteration in order to minimize the corresponding error in each application of the d-CDP operator given by (see Lemma A.6 and Proposition A.8)

$$e_{\mathrm{y}} = c_{\mathrm{y}} \cdot \max_{x \in \mathbb{X}^{\mathrm{d}}} \mathrm{d}\left(\partial(\mathcal{T}J - C_{\mathrm{s}})(x), \mathbb{Y}^{\mathrm{g}}\right).$$

This means that line 4 in Algorithm 1 is moved inside the iterations, after line 8. Similar to the static scheme described above, the aim here is to construct $\mathbb{Y}^{\mathrm{g}}$ such that $\mathrm{co}(\mathbb{Y}^{\mathrm{g}}) \supseteq \mathbb{L}(\mathcal{T}J - C_{\mathrm{s}})$. Since we do not have access to $\mathcal{T}J$ (it is the output of the current iteration), we can again use the definition of the DP operator (1) to approximate $\mathrm{rng}(\mathcal{T}J - C_{\mathrm{s}})$ by

$$R = \mathrm{rng}(C_{\mathrm{i}}^{\mathrm{d}}) + \gamma \cdot \mathrm{rng}(J^{\mathrm{d}}),$$

where $J^{\mathrm{d}}$ is the output of the previous iteration. We then construct the gird $\mathbb{Y}^{\mathrm{g}} = \Pi_{i=1}^{n} \mathbb{Y}_i^{\mathrm{g}}$ such that, for each dimension $i$, the condition (9) holds. This construction has a one-time computational cost of $\mathcal{O}(U)$ for computing $\mathrm{rng}(C_{\mathrm{i}}^{\mathrm{d}})$ and a per iteration computational cost of $\mathcal{O}(X)$ for computing $\mathrm{rng}(J^{\mathrm{d}})$. Notice, however, that under this dynamic construction, the error bound of Theorem 3.13 does not hold true. More importantly, with a dynamic grid $\mathbb{Y}^{\mathrm{g}}$ that varies in each iteration, there is no guarantee for ConjVI to converge.

## 4 Numerical simulations

In this section, we compare the performance of the proposed ConjVI algorithm with the benchmark VI algorithm (in primal domain) through three numerical examples. For the first example, we focus on a synthetic system satisfying the conditions of assumptions considered in this study in order to examine our theoretical results. We then showcase the application of ConjVI in solving the optimal control problem of an inverted pendulum and a batch reactor. The simulations were implemented via MATLAB version R2017b, on a PC with Intel Xeon 3.60 GHz processor and 16 GB RAM. We also provide the ConjVI MATLAB package [22] for the implementation of the proposed algorithm. The package also includes the numerical simulations of this section. W note that multiple routines in the developed package are borrowed from the d-CDP MATLAB package [23]. Also, for the discrete conjugation (LLT), we used the MATLAB package (in particular, the `LLTd` routine) provided in [24].

### 4.1 Example 1 – Synthetic

We consider the linear system $x^+ = Ax + Bu + w$ with $A = [2 \; 1; \; 1 \; 3]$, $B = [1 \; 1; \; 1 \; 2]$. The problem of interest is the infinite-horizon, optimal control of this system with cost functions $C_{\mathrm{s}}(x) = 10 \left\| x \right\|_2^2$ and $C_{\mathrm{i}}(u) = e^{|u_1|} + e^{|u_2|} - 2$, and discount factor $\gamma = 0.95$. We consider state and input constraint sets $\mathbb{X} = [-1, 1]^2$ and $\mathbb{U} = [-2, 2]^2$, respectively. The disturbance is assumed to have a uniform distribution over the finite support $\mathbb{W}^{\mathrm{d}} = \{0, \pm 0.05\} \times \{0\}$ of size $W = 3$. Notice how the stage cost is a combination of a quadratic term (in state) and an exponential term (in input). Particularly, the control problem at hand does not have a closed-form solution. We use uniform, grid-like discretizations $\mathbb{X}^{\mathrm{g}}$ and $\mathbb{U}^{\mathrm{g}}$ for the state and input spaces such that $\mathrm{co}(\mathbb{X}^{\mathrm{g}}) = \mathbb{X}$ and $\mathrm{co}(\mathbb{U}^{\mathrm{g}}) = \mathbb{U}$. This choice allows us to deploy *multilinear interpolation*, which is non-expansive, as the extension operator $\widetilde{[\cdot]}$ in the d-DP operation (2) in VI, and in the d-CDP operation (6a) in ConjVI. The grids $\mathbb{V}^{\mathrm{g}}, \mathbb{Z}^{\mathrm{g}} \subset \mathbb{R}^2$ are also constructed uniformly, following the guidelines provided in Section 3.2. For the construction of $\mathbb{Y}^{\mathrm{g}} \subset \mathbb{R}^2$, we also follow the guidelines of Section 3.2 with $\alpha = 1$. In particular, we also consider the *dynamic* scheme for the construction of $\mathbb{Y}^{\mathrm{g}}$ in ConjVI (hereafter, referred to as ConjVI-d). Moreover, in each implementation of VI and ConjVI(-d), all of the involved grids ($\mathbb{X}^{\mathrm{g}}, \mathbb{U}^{\mathrm{g}}, \mathbb{Y}^{\mathrm{g}}, \mathbb{V}^{\mathrm{g}}, \mathbb{Z}^{\mathrm{g}}$) are chosen to be of the same size $N^2$ (with $N$ points in each dimension). We are particularly interested in the performance of these algorithms, as $N$ increases. We note that the described setup satisfies all of the assumptions in this study.

The results of our numerical simulations are shown in Figure 1. As shown in Figures 1a, both VI and ConjVI are indeed convergent with a rate less than or equal to the discount factor $\gamma = 0.95$; see Theorem 3.11. In particular, ConjVI terminates in $k_{\mathrm{t}} = 55$ iterations, compared to $k_{\mathrm{t}} = 102$ iterations required for VI to reach the termination bound $e_{\mathrm{t}} = 0.001$. Not surprisingly, this faster convergence, combined with the lower time complexity of ConjVI in each iteration, leads to a significant reduction in the running time of this algorithm compared to VI. This effect can be clearly

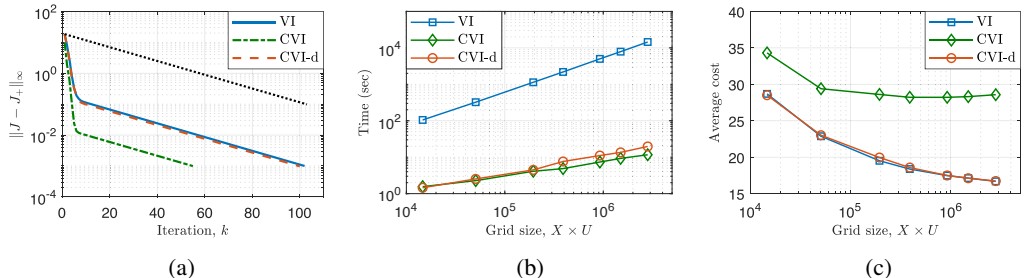

Figure 1: VI vs. ConjVI (CVI) – synthetic example with *stochastic* dynamics $x^+ = Ax + Bu + w$: (a) Convergence rate for $N = 41$; (b) Running time; (c) Average cost of one hundred instances of the control problem with random initial conditions over $T = 100$ time steps. The black dashed-dotted line in (a) corresponds to exponential convergence with coefficient $\gamma = 0.95$. CVI-d corresponds to *dynamic* construction of the dual grid $\mathbb{Y}^g$ in the ConjVI algorithm.

seen in Figure 1b, where the run-time of ConjVI for $N = 41$ is an order of magnitude less than that of VI for $N = 11$. In this regard, we note that the setting of this numerical example leads to $\mathcal{O}(k_t N^4 W)$ and $\mathcal{O}(k_t N^2 W)$ time complexities for VI and ConjVI, respectively; see Theorem 3.12 and the discussion after that. Indeed, the running times in Figure 1b match these complexities.

Since we do not have access to the true optimal value function, in order to evaluate the performance of the outputs of the VI and ConjVI, we consider the performance of the greedy policy

$$\mu(x) \in \operatorname*{argmin}_{u \in \mathbb{U}(x) \cap \mathbb{U}^g} \big\{ C(x, u) + \gamma \cdot \mathbb{E}_w \overline{J^d}\big(g(x, u, w)\big) \big\},$$

w.r.t. the discrete value function $J^d$ computed using these algorithms (we note that, for finding the greedy action, we used the same discretization $\mathbb{U}^g$ of the input space and the same extension $\overline{J^d}$ of the value function as the one used in VI and ConjVI, however, this need not to be the case in general). Figure 1c reports the average cost of one hundred instances of the optimal control problem with greedy control actions. As shown, the reduction in the run-time in ConjVI comes with an increase in the cost of the controlled trajectories.

Let us now consider the effect of *dynamic* construction of the state dual grid $\mathbb{Y}^g$. As can be seen in Figure 1a, using a dynamic $\mathbb{Y}^g$ leads to a slower convergence (ConjVI-d terminates in $k_t = 100$ iterations). We note that the relative behaviour of the convergence rates in Figures 1a was also seen for other grid sizes in the discretization scheme. However, we see a small increase in the running time of ConjVI-d compared to ConjVI since the per iteration complexity for ConjVI-d is again of $\mathcal{O}(k_t N^2 W)$; see Figure 1b. More importantly, as depicted in Figure 1c, ConjVI-d shows almost the same performance as VI when it comes to the quality of the greedy actions. This is because the dynamic construction of $\mathbb{Y}^g$ in ConjVI-d uses the available computational power (related to size of the discretization) smartly by finding the smallest grid $\mathbb{Y}^g$ in each iteration, in order to minimize the error of that same iteration.

We note that our simulations show that for the *deterministic* system, ConjVI-d has a similar converge rate as ConjVI. This effect can be seen in Figure 2, where ConjVI-d terminates in 10 iterations. Interestingly, in this particular example, ConjVI actually converges to the fixed point after 7 iterations ($J_8^d = \widehat{\mathcal{T}}^d J_7^d$) for the deterministic system. Let us finally note that the conjugate $C_i^*$ of the input cost in the provided example is indeed analytically available. One can use this analytic representation in order to exactly compute $C_i^*$ in (6f) and avoid the corresponding numerical approximation. With such a modification, the computational cost reduces, however, our numerical experiments show that for the provided example, the ConjVI outputs effectively the same value function within the same number of iterations (results are not shown here).

## 4.2 Example 2 – Inverted pendulum

We use the setup (model and stage cost) of [21, App. C.2.2] with discount factor $\gamma = 0.95$. In particular, the state and input costs are both quadratic ($\|\cdot\|_2^2$), and the discrete-time, nonlinear

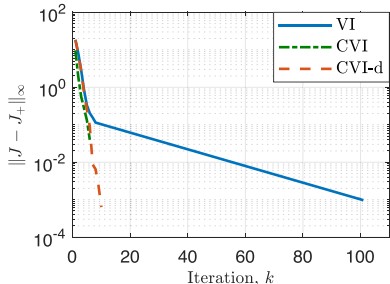

Figure 2: Convergence of VI and ConjVI with *deterministic* dynamics $x^+ = Ax + Bu$; cf. Figure 1a.

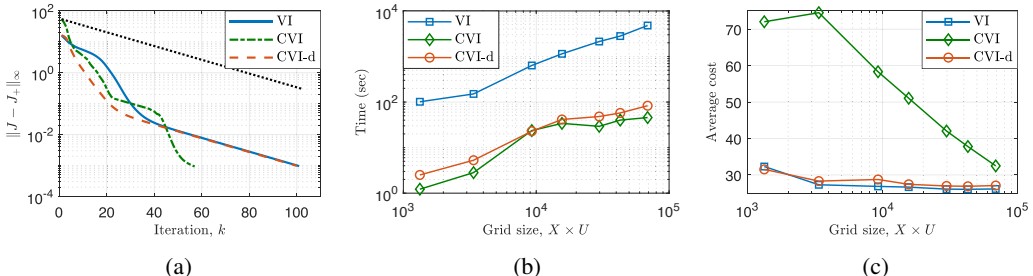

(a)            (b)            (c)

Figure 3: VI vs. ConjVI (CVI) – optimal control of noisy inverted pendulum: (a) Convergence rate for $N = 41$; (b) Running time; (c) Average cost of one hundred instances of the control problem with random initial conditions over $T = 100$ time steps. The black dashed-dotted line in (a) corresponds to exponential convergence with coefficient $\gamma = 0.95$. CVI-d corresponds to *dynamic* construction of the dual grid $\mathbb{Y}^g$ in the ConjVI algorithm.

dynamics is of the form $x^+ = f_s(x) + Bu + w$, where

$$f_s(x_1, x_2) = \begin{bmatrix} x_1 + \alpha_{12}x_2 \\ \alpha_{21}\sin x_1 + \alpha_{22}x_2 \end{bmatrix}, \ B = \begin{bmatrix} 0 \\ \beta \end{bmatrix}, \quad (\alpha_{12}, \alpha_{21}, \alpha_{22}, \beta \in \mathbb{R}).$$

State and input constraints are described by $\mathbb{X} = [-\frac{\pi}{3}, \frac{\pi}{3}] \times [-\pi, \pi] \subset \mathbb{R}^2$ and $\mathbb{U} = [-3, 3] \subset \mathbb{R}$. The disturbance has a uniform distribution over the finite support $\mathbb{W}^g = \{0, \pm 0.025\frac{\pi}{3}, \pm 0.05\frac{\pi}{3}\} \times \{0, \pm 0.025\pi, \pm 0.05\pi\} \subset \mathbb{R}^2$ of size $W = 5^2$. We use uniform, grid-like discretizations $\mathbb{X}^g$ and $\mathbb{U}^g$ for the state and input spaces such that $co(\mathbb{X}^g) = [-\frac{\pi}{4}, \frac{\pi}{4}] \times [-\pi, \pi] \subset \mathbb{X}$ and $co(\mathbb{U}^g) = \mathbb{U}$. This choice of discrete state space $\mathbb{X}^g$ particularly satisfies the feasibility condition of Assumption 3.6. (Note however that the continuous state space $\mathbb{X}$ does not satisfy the feasibility condition of Assumption 3.1-(iii)). Also, we use *nearest neighbor* extension (which is non-expansive) for the extension operators in (2) for VI and in (6a) for ConjVI. The grids $\mathbb{V}^g \subset \mathbb{R}$ and $\mathbb{Z}^g, \mathbb{Y}^g \subset \mathbb{R}^2$ are also constructed uniformly, following the guidelines of Section 3.4 (with $\alpha = 1$). We again also consider the *dynamic* scheme for the construction of $\mathbb{Y}^g$. Moreover, in each implementation of VI and ConjVI(-d) the termination bound is $e_t = 0.001$, and all of the involved grids are chosen to be of the same size $N$ in each dimension, i.e., $X = Y = Z = N^2$ and $U = V = N$.

The results of simulations are shown in Figures 3 and 4. As reported, we essentially observe the same behaviors as before. In particular, application of ConjVI(-d), especially for deterministic dynamics, leads to a faster convergence and a significant reduction in the running time; see Figures 3a, 3b and 4. Note that Figure 4 also shows the non-monotone behavior of ConjVI-d for scaling factor $\alpha = 3$. In this regard, recall that when the grid $\mathbb{Y}^g$ is constructed dynamically and varies at each iteration, the d-CDP operator is not necessarily contractive. Moreover, as shown in Figures 3b and 3c, this dynamic scheme leads to a huge improvement in the performance of the corresponding greedy policy at the expense of a small increase in the computational cost.

### 4.3 Example 3 – Batch Reactor

Our last numerical example concerns the optimal control of a system with four states and two input channels, namely, an unstable batch reactor. The setup (dynamics, cost, and constraints) are borrowed

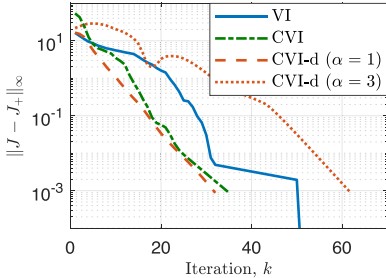

Figure 4: Convergence of VI and ConjVI with *deterministic* dynamics $x^+ = f_s(x) + Bu$; cf. Figure 3a.

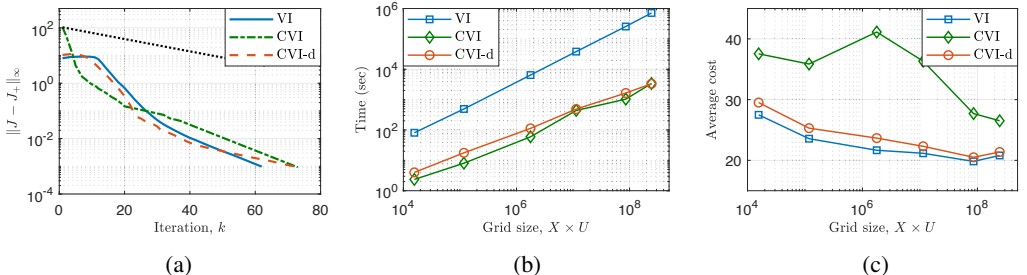

(a)                      (b)                      (c)

Figure 5: VI vs. ConjVI (CVI) – optimal control of batch reactor: (a) Convergence rate for $N = 25$; (b) Running time; (c) Average cost of one hundred instances of the control problem with random initial conditions over $T = 100$ time steps. The black dashed-dotted line in (a) corresponds to exponential convergence with coefficient $\gamma = 0.95$. CVI-d corresponds to *dynamic* construction of the dual grid $\mathbb{Y}^g$ in the ConjVI algorithm.

from [20, Sec. 6]. In particular, we consider a *deterministic* linear dynamics $x^+ = Ax + Bu$, with costs $C_s(x) = 2\|x\|_2^2$ and $C_i(u) = \|u\|_2^2$, discount factor $\gamma = 0.95$, and constraints $x \in \mathbb{X} = [-2,2]^4 \subset \mathbb{R}^4$ and $u \in \mathbb{U} = [-2,2]^2 \subset \mathbb{R}^2$. Once again, we use uniform, grid-like discretizations $\mathbb{X}^g$ and $\mathbb{U}^g$ for the state and input spaces such that $\text{co}(\mathbb{X}^g) = [-1,1]^4 \subset \mathbb{X}$ and $\text{co}(\mathbb{U}^g) = \mathbb{U}$. The grids $\mathbb{V}^g \subset \mathbb{R}^2$ and $\mathbb{Z}^g, \mathbb{Y}^g \subset \mathbb{R}^4$ are also constructed uniformly, following the guidelines of Section 3.4 (with $\alpha = 1$). Moreover, in each implementation of VI and ConjVI, the termination bound is $e_t = 0.001$ and all of the involved grids are chosen to be of the same size $N$ in each dimension, i.e., $X = Y = Z = N^4$ and $U = V = N^2$. Finally, we note that we use *multi-linear interpolation and extrapolation* for the extension operator in (2) for VI. Due to the extrapolation, the extension operator is no longer non-expansive and hence the convergence of VI is not guaranteed. On the other hand, since the dynamics is deterministic, there is no need for extension in ConjVI (recall that the scaled expectation in (6a) in ConjVI reduces to the simple scaling $\varepsilon^d = \gamma \cdot J^d$ for deterministic dynamics), and hence the convergence of ConjVI only requires $\text{co}(\mathbb{Z}^g) \supseteq f_s(\mathbb{X}^g)$.

The results of our numerical simulations are shown in Figure 5. Once again, we see the trade-off between the time complexity and the greedy control performance in VI and ConjVI. On the other hand, ConjVI-d has the same control performance as VI with an insignificant increase in running time compared to ConjVI. In Figure 5a, we again observe the non-monotone behavior of ConjVI-d (the d-CDP operator is expansive in the first six iterations). The VI algorithm is also showing a non-monotone behavior, where for the first nine iterations the d-DP operation is actually expansive. As we noted earlier, this is because the extension via multi-linear extrapolation is expansive.

## 5 Final remarks

In this paper, we proposed the ConjVI algorithm which reduces the time complexity of the VI algorithm from $\mathcal{O}(XU)$ to $\mathcal{O}(X + U)$. This better time complexity however comes at the expense of restricting the class of problem. In particular, there are two main conditions that must be satisfied in order to be able to apply the ConjVI algorithm:

Table 1: VI vs. ConjVI - optimal control the batch reactor with stage cost (10) and $\eta = 0.01$.

| Algrithm | Run-time (sec) | Average cost (100 runs) |
|---|---|---|
| VI | 7669 | 33.9 |
| ConjVI | 55 | 73.5 |
| ConjVI-d | 90 | 74.0 |

   (i)  the dynamics must be of the form $x^+ = f_{\mathrm{s}}(x) + Bu + w$; and,

  (ii)  the stage cost $C(x, u) = C_{\mathrm{s}}(x) + C_{\mathrm{i}}(u)$ must be separable.

Moreover, since ConjVI essentially solves the dual problem, for non-convex problems, it suffers from a non-zero duality gap. Based on our simulation results, we also notice a trade-off between computational complexity and control action quality: While ConjVI has a lower computational cost, VI generates better control actions. However, the dynamic scheme for the construction of state dual grid $\mathbb{Y}^{\mathrm{g}}$ allows us to achieve almost the same performance as VI when it comes to the quality of control actions, with a small extra computational burden. In what follows, we provide our final remarks on the limitations of the proposed ConjVI algorithm and its relation to existing approximate VI algorithms.

**Relation to existing approximate VI algorithms.** The basic idea for complexity reduction introduced in this study can be potentially combined with and further improve the existing sample-based VI algorithms. These sample-based algorithms solely focus on transforming the infinite-dimensional optimization in DP problems into computationally tractable ones, and in general, they have a time complexity of $\mathcal{O}(XU)$, depending on the product of the cardinalities of the discrete state and action spaces. The proposed ConjVI algorithm, on the other hand, focuses on reducing this time complexity to $\mathcal{O}(X + U)$, by avoiding the minimization over input in each iteration. Take, for example, the aggregation technique in [27, Sec. 8.1] that leads to a piece-wise constant approximation of the value function. It is straightforward to combine ConjVI with this type of state space aggregation. Indeed, the numerical example of of Section 4.2 essentially uses such an aggregation by approximating the value function via nearest neighbor extension.

**Cost functions with a large Lipschitz constant.** Recall that for the proposed ConjVI algorithm to be computationally efficient, the size $Y$ of the state dual grid $\mathbb{Y}^{\mathrm{g}}$ must be controlled by the size $X$ of the discrete state space $\mathbb{X}^{\mathrm{d}}$ (Assumption 3.9-(iii)). Then, as the range of slope of the value function $J_\star$ increases, the corresponding error $e_{\mathrm{y}}$ in (8d) due to discretization of the dual state space increases. The proposed dynamic approach for construction of $\mathbb{Y}^{\mathrm{g}}$ partially addresses this issue by focusing on the range of slope of $J_k^{\mathrm{d}}$ in each iteration in order to minimize the discretization error of the same iteration $k$. However, when the cost function has a large Lipschitz constant, even this latter approach can fail to provide a good approximation of the value function. Table 1 reports the result of the numerical simulation of the unstable batch reactor with the stage cost

$$C(x, u) = -\frac{4}{1+\eta} + \sum_{i=1}^{4} \frac{1}{1+\eta-|x_i|} - \frac{2}{2+\eta} + \sum_{j=1}^{2} \frac{1}{2+\eta-|u_j|}, \quad \|x\|_\infty \le 1, \ \|u\|_\infty \le 2.$$
(10)

Clearly, as $\eta \to 0$, we increase the range of slope of the cost function. As can be seen, the quality of the greedy action generated by ConjVI-d also deteriorates in this case.

**Gradient-based algorithms for solving the minimization over input.** Let us first note that the minimization over $u$ in sample-based VI algorithms usually involves solving a difficult non-convex problem. This is particularly due to that fact that the extension operation employed in these algorithms for approximating the value function using the sample points does not lead to a convex function in $u$ (e.g., take kernel-based approximations or neural networks). This is why in MDP and RL literature, it is actually quite common to consider a finite action space in the first place [11, 27]. Moreover, the minimization over $u$ again must be solved for each sample point in each iteration, while application of ConjVI avoids solving this minimization in each iteration. In this regard, let us note that ConjVI actually uses a convex approximation of the value function, which allows for application of a gradient-based algorithm for minimization over $u$ within the ConjVI algorithm. Indeed, in each

iteration $k = 0, 1, \ldots$, ConjVI solves (for deterministic dynamics)

$$J_{k+1}^{\mathrm{d}}(x) = C_{\mathrm{s}}(x) + \min_u \left\{ C_{\mathrm{i}}(u) + \gamma \cdot \max_{y \in \mathbb{Y}^{\mathrm{g}}} \left[ \langle y, f_{\mathrm{s}}(x) + Bu \rangle - J_k^{\mathrm{d} * \mathrm{d}}(y) \right] \right\}, \quad x \in \mathbb{X}^{\mathrm{d}},$$

where

$$J_k^{\mathrm{d} * \mathrm{d}}(y) = \max_{x \in \mathbb{X}^{\mathrm{d}}} \left\{ \langle x, y \rangle - J_k^{\mathrm{d}}(x) \right\}, \quad y \in \mathbb{Y}^{\mathrm{g}},$$

is the discrete conjugate of the output of the previous iteration (computed using the LLT algorithm). Then, it is not hard to see that a subgradient of the objective of the minimization can be computed using $\mathcal{O}(Y)$ operations: for a given $u$, assuming we have access to the subdifferential $\partial C_{\mathrm{i}}(u)$, the subdifferential of the objective function is $\partial C_{\mathrm{i}}(u) + \gamma \cdot B^{\top} y_u$, where

$$y_u \in \operatorname*{argmax}_{y \in \mathbb{Y}^{\mathrm{g}}} \left\{ \langle y, f_{\mathrm{s}}(x) + Bu \rangle - J_k^{\mathrm{d} * \mathrm{d}}(y) \right\}.$$

This, leads to a per iteration complexity of $\mathcal{O}(XY) = \mathcal{O}(X^2)$, which is again practically inefficient.

# A    Technical proofs

## A.1    Proof of Proposition 3.2

This result is an extension of [21, Lem. 4.2] that accounts for the separable cost, the discount factor, and additive disturbance. Inserting the dynamics of Assumption 3.1-(i) into (3), we can use the definition of conjugate transform to obtain (all the functions are extended to infinity outside their effective domains)

$$
\begin{aligned}
\widehat{\mathcal{T}} J(x) &- C_{\mathrm{s}}(x) \\
&= \max_y \min_{u,z} \left\{ C_{\mathrm{i}}(u) + \gamma \cdot \mathbb{E}_w J(z + w) + \langle y, f_{\mathrm{s}}(x) + Bu - z \rangle \right\} \\
&= \max_y \left\{ \langle y, f_{\mathrm{s}}(x) \rangle - \max_u \left[ \langle -B^{\top} y, u \rangle - C_{\mathrm{i}}(u) \right] - \max_z \left[ \langle y, z \rangle - \gamma \cdot \mathbb{E}_w J(z + w) \right] \right\} \\
&= \max_y \left\{ \langle y, f_{\mathrm{s}}(x) \rangle - C_{\mathrm{i}}^*(-B^{\top} y) - [\gamma \cdot \mathbb{E}_w J(\cdot + w)]^*(y) \right\} \\
&= \max_y \left\{ \langle y, f_{\mathrm{s}}(x) \rangle - C_{\mathrm{i}}^*(-B^{\top} y) - \epsilon^*(y) \right\} \\
&= \max_y \left\{ \langle y, f_{\mathrm{s}}(x) \rangle - \phi(y) \right\} \\
&= \phi^* \big( f_{\mathrm{s}}(x) \big),
\end{aligned}
$$

where we used the definition of *epsilon* and $\phi$ in (4a) and (4b), respectively.

## A.2    Proof of Proposition 3.3

We can use the representation (4) and the definition of conjugate operation to obtain

$$
\begin{aligned}
\widehat{\mathcal{T}} J(x) - C_{\mathrm{s}}(x) &= \max_y \{ \langle f_{\mathrm{s}}(x), y \rangle - \phi(y) \} \\
&= \max_y \left\{ \langle f_{\mathrm{s}}(x), y \rangle - C_{\mathrm{i}}^*(-B^{\top} y) - \epsilon^*(y) \right\} \\
&= \max_y \left\{ \langle f_{\mathrm{s}}(x), y \rangle - [C_{\mathrm{i}}^*]^{**}(-B^{\top} y) - \epsilon^*(y) \right\} \\
&= \max_y \left\{ \langle f_{\mathrm{s}}(x), y \rangle - \max_{u \in \mathrm{co}(\mathbb{U})} \left[ \langle -B^{\top} y, u \rangle - C_{\mathrm{i}}^{**}(u) \right] - \epsilon^*(y) \right\} \\
&= \max_y \min_{u \in \mathrm{co}(\mathbb{U})} \left\{ C_{\mathrm{i}}^{**}(u) + \langle y, f_{\mathrm{s}}(x) + Bu \rangle - \epsilon^*(y) \right\},
\end{aligned}
$$

where we used the fact that $C_{\mathrm{i}}^* : \mathbb{R}^m \to \mathbb{R}$ is proper, closed, and convex, and hence $[C_{\mathrm{i}}^*]^{**} = C_{\mathrm{i}}^*$. This follows from the fact that $\mathrm{dom}(C_{\mathrm{i}}) = \mathbb{U}$ is assumed to be compact (Assumption 3.1-(iii)). Hence, the objective function of this maximin problem is convex in $u$, with $\mathrm{co}(\mathbb{U})$ being compact,

which follows from convexity of $C_i^{**} : \mathrm{co}(\mathbb{U}) \to \mathbb{R}$. Also, the objective function is concave in $y$, which follows from the convexity of $\epsilon^*$. Then, by Sion's Minimax Theorem (see, e.g., [29, Thm. 3]), we have minimax-maximin equality, i.e.,

$$
\begin{aligned}
\widehat{\mathcal{T}} J(x) - C_{\mathrm{s}}(x) &= \min_u \max_y \left\{ C_i^{**}(u) + \langle y, f(x,u) \rangle - \epsilon^*(y) \right\} \\
&= \min_u \left\{ C_i^{**}(u) + \max_y \left[ \langle y, f(x,u) \rangle - \epsilon^*(y) \right] \right\} \\
&= \min_u \left\{ C_i^{**}(u) + \epsilon^{**}\big(f(x,u)\big) \right\} \\
&= \min_u \left\{ C_i^{**}(u) + \gamma \cdot [\mathbb{E}_w J(\cdot + w)]^{**}\big(f(x,u)\big) \right\},
\end{aligned}
$$

where the last equality, we used the fact that $[\gamma h]^{**} = \gamma \cdot h^{**}$; see [4, Prop. 13.23–(i)&(iv)].

## A.3  Proof of Corollary 3.4

By Proposition 3.3, we need to show $C_i^{**} = C_i$ and $[\mathbb{E}_w J(\cdot + w)]^{**} = \mathbb{E}_w J(\cdot + w)$ so that

$$
\begin{aligned}
C_i^{**}(u) + \gamma \cdot [\mathbb{E}_w J(\cdot + w)]^{**}\big(f(x,u)\big) &= C_i(u) + \gamma \cdot [\mathbb{E}_w J(\cdot + w)]\big(f(x,u)\big) \\
&= C_i(u) + \gamma \cdot \mathbb{E}_w J\big(f(x,u) + w\big) \\
&= C_i(u) + \gamma \cdot \mathbb{E}_w J\big(g(x,u,w)\big).
\end{aligned}
$$

This holds if $C_i$ and $\mathbb{E}_w J(\cdot + w)$ are proper, closed and convex. This is indeed the case since $\mathbb{X}$ and $\mathbb{U}$ are compact, and $C_i : \mathbb{U} \to \mathbb{R}$ and $J : \mathbb{X} \to \mathbb{R}$ are assumed to be convex.

## A.4  Proof of Theorem 3.11

We begin with two preliminary lemmas on the non-expansiveness of conjugate and multilinear interpolation operations within the d-CDP operation (6).

**Lemma A.1** (Non-expansiveness of conjugate operator). *Consider two functions $h_i$ ($i = 1, 2$), with the same nonempty effective domain $\mathbb{X}$. For any $y \in \mathrm{dom}(h_1^*) \cap \mathrm{dom}(h_2^*)$, we have*

$$
|h_1^*(y) - h_2^*(y)| \leq \|h_1 - h_2\|_\infty .
$$

*Proof.* For any $y \in \mathrm{dom}(h_1^*) \cap \mathrm{dom}(h_2^*)$, we have

$$
h_1^*(y) = \max_{x \in \mathbb{X}} \langle x, y \rangle - h_1(x) = \max_{x \in \mathbb{X}} \langle x, y \rangle - h_2(x) + h_2(x) - h_1(x).
$$

Hence,

$$
h_2^*(y) - \|h_1 - h_2\|_\infty \leq h_1^*(y) \leq h_2^*(y) + \|h_1 - h_2\|_\infty ,
$$

that is,

$$
|h_1^*(y) - h_2^*(y)| \leq \|h_1 - h_2\|_\infty .
$$

$\square$

**Lemma A.2** (Non-expansiveness of interpolative LERP operator). *Consider two discrete functions $h_i^{\mathrm{d}}$ ($i = 1, 2$) with the same grid-like domain $\mathbb{X}^{\mathrm{g}} \subset \mathbb{R}^n$, and their* interpolative *LERP extensions $\overline{h_i^{\mathrm{d}}} : \mathrm{co}(\mathbb{X}^{\mathrm{g}}) \to \mathbb{R}$. We have*

$$
\left\| \overline{h_1^{\mathrm{d}}} - \overline{h_2^{\mathrm{d}}} \right\|_\infty \leq \left\| h_1^{\mathrm{d}} - h_2^{\mathrm{d}} \right\|_\infty .
$$

*Proof.* For any $x \in \mathrm{co}(\mathbb{X}^{\mathrm{g}})$, we have ($i = 1, 2$)

$$
\overline{h_i^{\mathrm{d}}}(x) = \sum_{j=1}^{2^n} \alpha^j \, h_i^{\mathrm{d}}(x^j),
$$

where $x^j$, $j = 1, \ldots, 2^n$, are the vertices of the hyper-rectangular cell that contains $x$, and $\alpha^j$, $j = 1, \ldots, 2^n$, are convex coefficients (i.e., $\alpha^j \in [0, 1]$ and $\sum_j \alpha^j = 1$). Then

$$|\overline{h_1^{\mathrm{d}}}(x) - \overline{h_2^{\mathrm{d}}}(x)| \leq \sum_{j=1}^{2^n} \alpha^j \, |h_1^{\mathrm{d}}(x^j) - h_2^{\mathrm{d}}(x^j)| \leq \left\| h_1^{\mathrm{d}} - h_2^{\mathrm{d}} \right\|_\infty.$$

$\square$

With these preliminary results at hand, we can now show that $\widehat{\mathcal{T}}^{\mathrm{d}}$ is $\gamma$-contractive. Consider two discrete functions $J_i^{\mathrm{d}} : \mathbb{X}^{\mathrm{d}} \to \mathbb{R}$ ($i = 1, 2$). For any $x \in \mathbb{X}^{\mathrm{d}} \subset \mathbb{R}^n$, we have

$$\left| \widehat{\mathcal{T}}^{\mathrm{d}} J_1^{\mathrm{d}}(x) - \widehat{\mathcal{T}}^{\mathrm{d}} J_2^{\mathrm{d}}(x) \right| \overset{(6f)}{=} \left| \overline{\varphi_1^{\mathrm{d}*\mathrm{d}}}\big(f_{\mathrm{s}}(x)\big) - \overline{\varphi_2^{\mathrm{d}*\mathrm{d}}}\big(f_{\mathrm{s}}(x)\big) \right| \overset{\text{Lem. } A.2}{\leq} \left\| \varphi_1^{\mathrm{d}*\mathrm{d}} - \varphi_2^{\mathrm{d}*\mathrm{d}} \right\|_\infty$$

$$\overset{\text{Def.}}{\leq} \left\| \varphi_1^{\mathrm{d}*} - \varphi_2^{\mathrm{d}*} \right\|_\infty \overset{\text{Lem. } A.1}{\leq} \left\| \varphi_1^{\mathrm{d}} - \varphi_2^{\mathrm{d}} \right\|_\infty \overset{(6d)}{\leq} \left\| \varepsilon_1^{\mathrm{d}*\mathrm{d}} - \varepsilon_2^{\mathrm{d}*\mathrm{d}} \right\|_\infty$$

$$\overset{\text{Def.}}{\leq} \left\| \varepsilon_1^{\mathrm{d}*} - \varepsilon_2^{\mathrm{d}*} \right\|_\infty \overset{\text{Lem. } A.1}{\leq} \left\| \varepsilon_1^{\mathrm{d}} - \varepsilon_2^{\mathrm{d}} \right\|_\infty$$

$$\overset{(6a)}{=} \gamma \cdot \left\| \sum_{w \in \mathbb{W}^{\mathrm{d}}} p(w) \cdot \left( \widetilde{J_1^{\mathrm{d}}}(x + w) - \widetilde{J_2^{\mathrm{d}}}(x + w) \right) \right\|_\infty$$

$$\leq \gamma \cdot \left\| \widetilde{J_1^{\mathrm{d}}} - \widetilde{J_2^{\mathrm{d}}} \right\|_\infty \leq \gamma \cdot \left\| J_1^{\mathrm{d}} - J_2^{\mathrm{d}} \right\|_\infty.$$

We note that we are using: (i) Assumption 3.9-(ii) in the application of Lemma A.2, (ii) the fact that $\mathrm{dom}(\varphi_i^{\mathrm{d}*}) = \mathrm{dom}(\varepsilon_i^{\mathrm{d}*}) = \mathbb{R}^n$ for $i = 1, 2$ in the two applications of Lemma A.1, and (iii) Assumption 3.10-(i) in the last inequality.

### A.5 Proof of Theorem 3.12

In what follows, we provide the time complexity of each line of Algorithm 1. In particular, we use the fact that $Y, Z = X$ and $V = U$ by Assumption 3.9-(iii). The complexity of construction of $\mathbb{V}^{\mathrm{g}}$ in line 1 is of $\mathcal{O}(X + U)$ by Assumption 3.9-(iii). The LLT of line 2 requires $\mathcal{O}(U + V) = \mathcal{O}(U)$ operations [24, Cor. 5]. The complexity of lines 3 and 4 is of $\mathcal{O}(X + U)$ by Assumption 3.9-(iii) on the complexity of construction of $\mathbb{Z}^{\mathrm{g}}$ and $\mathbb{Y}^{\mathrm{g}}$. The operation of line 5 also has a complexity of $\mathcal{O}(X)$, and line 6 requires $\mathcal{O}(X + U)$ operations. This leads to the reported $\mathcal{O}(X + U)$ time complexity for initialization.

In each iteration, lines 8 requires $\mathcal{O}(X)$ operations. The complexity of line 9 is of $\mathcal{O}(XWE)$ by the assumption on the complexity of the extension operator $\widetilde{[\cdot]}$. The LLT of line 10 requires $\mathcal{O}(X + Y) = \mathcal{O}(X)$ operations [24, Cor. 5]. The application of LERP in line 12 has a complexity of $\mathcal{O}(\log V)$ [21, Rem. 2.2]. Hence, the `for loop` over $y \in \mathbb{Y}^{\mathrm{g}}$ requires $\mathcal{O}(Y \log V) = \mathcal{O}(X \log U) = \widetilde{\mathcal{O}}(X)$ operations. The LLT of line 15 requires $\mathcal{O}(Z + Y) = \mathcal{O}(X)$ operations [24, Cor. 5]. The application of LERP in line 17 has a complexity of $\mathcal{O}(\log Z)$ [21, Rem. 2.2]. Hence, the `for loop` over $x \in \mathbb{X}^{\mathrm{d}}$ requires $\mathcal{O}(X \log Z) = \mathcal{O}(X \log X) = \widetilde{\mathcal{O}}(X)$ operations. The time complexity of each iteration is then of $\widetilde{\mathcal{O}}(XWE)$.

### A.6 Proof of Theorem 3.13

Note that the ConjVI Algorithm 1 involves consecutive applications of the d-CDP operator $\widehat{\mathcal{T}}^{\mathrm{d}}$ (6), and terminates after a finite number of iterations corresponding to the bound $e_{\mathrm{t}}$. We begin with bounding the difference between the DP and d-CDP operators.

**Error of d-CDP operation.** In what follows *we assume that $J : \mathbb{X} \to \mathbb{R}$ is a Lipschitz continuous, convex function that satisfies the condition of Assumption 3.10-(ii).* By Corollary 3.4, this assumption implies that the DP and CDP operators are equivalent, i.e., $\mathcal{T}J = \widehat{\mathcal{T}}J$. Hence, it suffices to bound the error of the d-CDP operator $\widehat{\mathcal{T}}^{\mathrm{d}}$ w.r.t. the CDP operator $\widehat{\mathcal{T}}$. We begin with the following preliminary lemma.

**Lemma A.3.** *The scaled expectation $\epsilon$ in (4a) is Lipschitz continuous and convex with a nonempty, compact effective domain. Moreover, $\mathrm{L}(\epsilon) \leq \gamma \cdot \mathrm{L}(J)$.*

*Proof.* The convexity follows from the fact that expectation preserves convexity and $\gamma > 0$. The effective domain of $\epsilon$ is nonempty by the feasibility condition of Assumption 3.1-(iii), and is compact since $\mathbb{X}$ is assumed to be compact. Finally, the bound on the Lipschitz constant of $\epsilon$ immediately follows from (4a). □

We now provide our step-by-step proof. Consider the function $\epsilon$ in (4a) and its discretization $\epsilon^{\mathrm{d}} : \mathbb{X}^{\mathrm{d}} \to \overline{\mathbb{R}}$. Also, consider the discrete function $\varepsilon^{\mathrm{d}} : \mathbb{X}^{\mathrm{d}} \to \overline{\mathbb{R}}$ in (6a).

**Lemma A.4.** *We have* $\mathrm{dom}(\epsilon^{\mathrm{d}}) = \mathrm{dom}(\varepsilon^{\mathrm{d}}) \neq \emptyset$. *Moreover,* $\left\| \epsilon^{\mathrm{d}} - \varepsilon^{\mathrm{d}} \right\|_{\infty} \leq \gamma \cdot e_{\mathrm{e}}$.

*Proof.* The first statement follows from the feasibility condition of Assumption 3.6. For the second statement, note that for every $x \in \mathrm{dom}(\epsilon^{\mathrm{d}}) = \mathrm{dom}(\varepsilon^{\mathrm{d}})$, we can use (4a) and (6a) to write

$$
\begin{aligned}
\left| \epsilon^{\mathrm{d}}(x) - \varepsilon^{\mathrm{d}}(x) \right| &= \gamma \cdot \left| \sum_{w \in \mathbb{W}^{\mathrm{d}}} p(w) \cdot \left( J(x+w) - \widetilde{J^{\mathrm{d}}}(x+w) \right) \right| \\
&\leq \gamma \cdot \sum_{w \in \mathbb{W}^{\mathrm{d}}} p(w) \cdot \left| J(x+w) - \widetilde{J^{\mathrm{d}}}(x+w) \right| \\
&\leq \gamma \cdot \left\| J - \widetilde{J^{\mathrm{d}}} \right\|_{\infty}.
\end{aligned}
$$

The result then follows from Assumption 3.10-(ii) on $J$. □

Now, consider the function $\phi : \mathbb{R}^{n} \to \mathbb{R}$ in (4b) and its discretization $\phi^{\mathrm{d}} : \mathbb{Y}^{\mathrm{g}} \to \mathbb{R}$. Also, consider the discrete function $\varphi^{\mathrm{d}} : \mathbb{Y}^{\mathrm{g}} \to \mathbb{R}$ in (6d).

**Lemma A.5.** *We have* $\left\| \phi^{\mathrm{d}} - \varphi^{\mathrm{d}} \right\|_{\infty} \leq \gamma \cdot e_{\mathrm{e}} + e_{\mathrm{u}} + e_{\mathrm{v}} + e_{\mathrm{x}}$, *where*

$$
\begin{aligned}
e_{\mathrm{u}} &= \left[ \|B\|_{2} \cdot \Delta_{\mathbb{Y}^{\mathrm{g}}} + \mathrm{L}(C_{\mathrm{i}}) \right] \cdot \mathrm{d}_{\mathrm{H}}(\mathbb{U}, \mathbb{U}^{\mathrm{d}}), \\
e_{\mathrm{v}} &= \Delta_{\mathbb{U}^{\mathrm{d}}} \cdot \mathrm{d}_{\mathrm{H}}\left( \mathrm{co}(\mathbb{V}^{\mathrm{g}}), \mathbb{V}^{\mathrm{g}} \right), \\
e_{\mathrm{x}} &= \left[ \Delta_{\mathbb{Y}^{\mathrm{g}}} + \gamma \cdot \mathrm{L}(J) \right] \cdot \mathrm{d}_{\mathrm{H}}(\mathbb{X}, \mathbb{X}^{\mathrm{d}}).
\end{aligned}
$$

*Proof.* Let $y \in \mathbb{Y}^{\mathrm{g}}$. According to (4b) and (6d), we have (note that $\varepsilon^{\mathrm{d}*\mathrm{d}}(y) = \varepsilon^{\mathrm{d}*}(y)$)

$$
\phi^{\mathrm{d}}(y) - \varphi^{\mathrm{d}}(y) = \phi(y) - \varphi(y) = C_{\mathrm{i}}^{*}(-B^{\top}y) - \overline{C_{\mathrm{i}}^{\mathrm{d}*\mathrm{d}}}(-B^{\top}y) + \epsilon^{*}(y) - \varepsilon^{\mathrm{d}*}(y). \tag{11}
$$

First, let us use [21, Lem. 2.5] to write

$$
\begin{aligned}
0 \leq C_{\mathrm{i}}^{*}(-B^{\top}y) - C_{\mathrm{i}}^{\mathrm{d}*}(-B^{\top}y) &\leq \left[ \| - B^{\top}y \|_{2} + \mathrm{L}(C_{\mathrm{i}}) \right] \cdot \mathrm{d}_{\mathrm{H}}(\mathbb{U}, \mathbb{U}^{\mathrm{d}}) \\
&\leq \left[ \|B\|_{2} \cdot \Delta_{\mathbb{Y}^{\mathrm{g}}} + \mathrm{L}(C_{\mathrm{i}}) \right] \cdot \mathrm{d}_{\mathrm{H}}(\mathbb{U}, \mathbb{U}^{\mathrm{d}}) = e_{\mathrm{u}}. \tag{12}
\end{aligned}
$$

Also, Assumption 3.9-(i) allows to use [21, Cor. 2.7] and write

$$
0 \leq \overline{C_{\mathrm{i}}^{\mathrm{d}*\mathrm{d}}}(-B^{\top}y) - C_{\mathrm{i}}^{\mathrm{d}*}(-B^{\top}y) \leq \Delta_{\mathbb{U}^{\mathrm{d}}} \cdot \mathrm{d}_{\mathrm{H}}\left( \mathrm{co}(\mathbb{V}^{\mathrm{g}}), \mathbb{V}^{\mathrm{g}} \right) = e_{\mathrm{v}}. \tag{13}
$$

Now, by Lemma A.1 (non-expansiveness of conjugation) and Lemma A.4, we have

$$
\left| \epsilon^{\mathrm{d}*}(y) - \varepsilon^{\mathrm{d}*}(y) \right| \leq \left\| \epsilon^{\mathrm{d}} - \varepsilon^{\mathrm{d}} \right\|_{\infty} \leq \gamma \cdot e_{\mathrm{e}}. \tag{14}
$$

Moreover, we can use [21, Lem. 2.5] and Lemma A.3 to obtain

$$
\begin{aligned}
0 \leq \epsilon^{*}(y) - \epsilon^{\mathrm{d}*}(y) &\leq \left[ \|y\|_{2} + \mathrm{L}(\epsilon) \right] \cdot \mathrm{d}_{\mathrm{H}}(\mathbb{X}, \mathbb{X}^{\mathrm{d}}) \\
&\leq \left[ \Delta_{\mathbb{Y}^{\mathrm{g}}} + \gamma \cdot \mathrm{L}(J) \right] \cdot \mathrm{d}_{\mathrm{H}}(\mathbb{X}, \mathbb{X}^{\mathrm{d}}) = e_{\mathrm{x}}. \tag{15}
\end{aligned}
$$

Combining (11)-(15), we then have

$$
\begin{aligned}
\left| \phi^{\mathrm{d}}(y) - \varphi^{\mathrm{d}}(y) \right| &= \left| C_{\mathrm{i}}^{*}(-B^{\top}y) - \overline{C_{\mathrm{i}}^{\mathrm{d}*\mathrm{d}}}(-B^{\top}y) + \epsilon^{*}(y) - \varepsilon^{\mathrm{d}*}(y) \right| \\
&\leq \left| C_{\mathrm{i}}^{*}(-B^{\top}y) - C_{\mathrm{i}}^{\mathrm{d}*}(-B^{\top}y) \right| + \left| C_{\mathrm{i}}^{\mathrm{d}*}(-B^{\top}y) - \overline{C_{\mathrm{i}}^{\mathrm{d}*\mathrm{d}}}(-B^{\top}y) \right| \\
&\quad + \left| \epsilon^{*}(y) - \epsilon^{\mathrm{d}*}(y) \right| + \left| \epsilon^{\mathrm{d}*}(y) - \varepsilon^{\mathrm{d}*}(y) \right| \\
&\leq e_{\mathrm{u}} + e_{\mathrm{v}} + \gamma \cdot e_{\mathrm{e}} + e_{\mathrm{x}}.
\end{aligned}
$$

□

Next, consider the discrete composite functions $[\phi^* \circ f_\mathrm{s}]^\mathrm{d} : \mathbb{X}^\mathrm{d} \to \mathbb{R}$ and $[\varphi^{\mathrm{d}*} \circ f_\mathrm{s}]^\mathrm{d} : \mathbb{X}^\mathrm{d} \to \mathbb{R}$. In particular, notice that $\phi^* \circ f_\mathrm{s}$ appears in (4c).

**Lemma A.6.** *We have* $\left\| [\phi^* \circ f_\mathrm{s}]^\mathrm{d} - [\varphi^{\mathrm{d}*} \circ f_\mathrm{s}]^\mathrm{d} \right\|_\infty \leq \gamma \cdot e_\mathrm{e} + e_\mathrm{u} + e_\mathrm{v} + e_\mathrm{x} + e_\mathrm{y}$, *where*

$$e_\mathrm{y} = \left[ \Delta_{f_\mathrm{s}(\mathbb{X}^\mathrm{d})} + \Delta_\mathbb{X} + \|B\|_2 \cdot \Delta_\mathbb{U} \right] \cdot \max_{x \in \mathbb{X}^\mathrm{d}} \mathrm{d}\left( \partial(\mathcal{T}J - C_\mathrm{s})(x), \mathbb{Y}^\mathrm{g} \right).$$

*Proof.* Let $x \in \mathbb{X}^\mathrm{d}$. Also let $\phi^\mathrm{d} : \mathbb{Y}^\mathrm{g} \to \mathbb{R}$ be the discretization of $\phi : \mathbb{R}^n \to \mathbb{R}$. Since $\phi$ is convex by construction, we can use [21, Lem. 2.5] to obtain (recall that $\mathrm{L}(h; \mathbb{X})$ denotes the Lipschtiz constant of $h$ restricted to the set $\mathbb{X} \subset \mathrm{dom}(h)$)

$$0 \leq \phi^*\big(f_\mathrm{s}(x)\big) - \phi^{\mathrm{d}*}\big(f_\mathrm{s}(x)\big) \leq \min_{y \in \partial\phi^*(f_\mathrm{s}(x))} \left\{ \left[ \|f_\mathrm{s}(x)\|_2 + \mathrm{L}\left( \phi; \{y\} \cup \mathbb{Y}^\mathrm{g} \right) \right] \cdot \mathrm{d}(y, \mathbb{Y}^\mathrm{g}) \right\} \quad (16)$$

By using (4c) and the equivalence of DP and CDP operators we have $\phi^* \circ f_\mathrm{s} = \widehat{\mathcal{T}}J - C_\mathrm{s} = \mathcal{T}J - C_\mathrm{s}$. Also, the definition (4b) implies that

$$\begin{aligned} \mathrm{L}(\phi) &\leq \mathrm{L}\left( C_\mathrm{i}^* \circ -B^\top \right) + \mathrm{L}(\epsilon^*) \leq \|B\|_2 \cdot \mathrm{L}(C_\mathrm{i}^*) + \mathrm{L}(\epsilon^*) \\ &\leq \|B\|_2 \cdot \Delta_{\mathrm{dom}(C_\mathrm{i})} + \Delta_{\mathrm{dom}(\epsilon)} \leq \|B\|_2 \cdot \Delta_\mathbb{U} + \Delta_\mathbb{X}, \end{aligned}$$

where for the last inequality we used the fact that $\mathrm{dom}(\epsilon) \subseteq \mathrm{dom}(J) = \mathbb{X}$. Using this results in (16), we have

$$\begin{aligned} 0 \leq \phi^*\big(f_\mathrm{s}(x)\big) - \phi^{\mathrm{d}*}\big(f_\mathrm{s}(x)\big) &\leq \min_{y \in \partial(\mathcal{T}J - C_\mathrm{s})(x)} \left\{ \left[ \|f_\mathrm{s}(x)\|_2 + \Delta_\mathbb{X} + \|B\|_2\, \Delta_\mathbb{U} \right] \cdot \mathrm{d}(y, \mathbb{Y}^\mathrm{g}) \right\} \\ &\leq \left[ \Delta_{f_\mathrm{s}(\mathbb{X}^\mathrm{d})} + \Delta_\mathbb{X} + \|B\|_2 \cdot \Delta_\mathbb{U} \right] \cdot \max_{x' \in \mathbb{X}^\mathrm{d}} \mathrm{d}\left( \partial(\mathcal{T}J - C_\mathrm{s})(x'), \mathbb{Y}^\mathrm{g} \right) \\ &= e_\mathrm{y}. \end{aligned} \quad (17)$$

Second, by Lemmas A.1 and A.5, we have

$$\left| \phi^{\mathrm{d}*}(z) - \varphi^{\mathrm{d}*}(z) \right| \leq \left\| \phi^\mathrm{d} - \varphi^\mathrm{d} \right\|_\infty \leq \gamma \cdot e_\mathrm{e} + e_\mathrm{u} + e_\mathrm{v} + e_\mathrm{x}, \quad (18)$$

for all $z \in \mathbb{R}^n$, including $z = f_\mathrm{s}(x)$. Here, we are using the fact that $\mathrm{dom}(\phi^\mathrm{d}) = \mathrm{dom}(\varphi^\mathrm{d}) = \mathbb{Y}^\mathrm{g}$ and $\mathrm{dom}(\phi^{\mathrm{d}*}) = \mathrm{dom}(\varphi^{\mathrm{d}*}) = \mathbb{R}^n$. Combining inequalities (17) and (18), we obtain

$$\begin{aligned} \left| \phi^*\big(f_\mathrm{s}(x)\big) - \varphi^{\mathrm{d}*}\big(f_\mathrm{s}(x)\big) \right| &\leq \left| \phi^*\big(f_\mathrm{s}(x)\big) - \phi^{\mathrm{d}*}\big(f_\mathrm{s}(x)\big) \right| + \left| \phi^{\mathrm{d}*}\big(f_\mathrm{s}(x)\big) - \varphi^{\mathrm{d}*}\big(f_\mathrm{s}(x)\big) \right| \\ &\leq e_\mathrm{y} + \gamma \cdot e_\mathrm{e} + e_\mathrm{u} + e_\mathrm{v} + e_\mathrm{x}. \end{aligned}$$

This completes the proof. $\qquad\qquad\qquad\qquad\qquad\qquad\qquad\qquad\qquad\qquad\qquad\qquad\square$

We are now left with the final step. Consider the output of the d-CDP operator $\widehat{\mathcal{T}}^\mathrm{d}J^\mathrm{d} : \mathbb{X}^\mathrm{d} \to \mathbb{R}$. Also, consider the output of the CDP operator $\widehat{\mathcal{T}}J : \mathbb{X} \to \mathbb{R}$ and its discretization $[\widehat{\mathcal{T}}J]^\mathrm{d} : \mathbb{X}^\mathrm{d} \to \mathbb{R}$.

**Lemma A.7.** *We have*

$$\left\| \widehat{\mathcal{T}}^\mathrm{d}J^\mathrm{d} - [\widehat{\mathcal{T}}J]^\mathrm{d} \right\|_\infty \leq \gamma \cdot e_\mathrm{e} + e_\mathrm{u} + e_\mathrm{v} + e_\mathrm{x} + e_\mathrm{y} + e_\mathrm{z},$$

*where*

$$e_\mathrm{z} = \Delta_{\mathbb{Y}^\mathrm{g}} \cdot \mathrm{d}_\mathrm{H}\left( f_\mathrm{s}(\mathbb{X}^\mathrm{d}), \mathbb{Z}^\mathrm{g} \right).$$

*Proof.* Let $x \in \mathbb{X}^\mathrm{d}$. According to (4c) and (6f), we have

$$\widehat{\mathcal{T}}^\mathrm{d}J^\mathrm{d}(x) - [\widehat{\mathcal{T}}J]^\mathrm{d}(x) = \widehat{\mathcal{T}}^\mathrm{d}J^\mathrm{d}(x) - \widehat{\mathcal{T}}J(x) = \overline{\varphi^{\mathrm{d}*\mathrm{d}}}\big(f_\mathrm{s}(x)\big) - \phi^*\big(f_\mathrm{s}(x)\big) \quad (19)$$

Now, by Lemma A.6, we have

$$\left| \phi^*\big(f_\mathrm{s}(x)\big) - \varphi^{\mathrm{d}*}\big(f_\mathrm{s}(x)\big) \right| \leq \gamma \cdot e_\mathrm{e} + e_\mathrm{u} + e_\mathrm{v} + e_\mathrm{x} + e_\mathrm{y}. \quad (20)$$

Moreover, Assumption 3.9-(ii) allows us to use [21, Cor. 2.7] and obtain

$$0 \leq \overline{\varphi^{\mathrm{d}*\mathrm{d}}}\big(f_\mathrm{s}(x)\big) - \varphi^{\mathrm{d}*}\big(f_\mathrm{s}(x)\big) \leq \Delta_{\mathbb{Y}^\mathrm{g}} \cdot \mathrm{d}_\mathrm{H}\left( f_\mathrm{s}(\mathbb{X}^\mathrm{d}), \mathbb{Z}^\mathrm{g} \right) = e_\mathrm{z}. \quad (21)$$

Combining (19), (20), and (21), we then have

$$\left|\widehat{\mathcal{T}}^{\mathrm{d}} J^{\mathrm{d}}(x) - [\widehat{\mathcal{T}} J]^{\mathrm{d}}(x)\right| = \left|\overline{\varphi^{\mathrm{d}*\mathrm{d}}}\big(f_{\mathrm{s}}(x)\big) - \phi^*\big(f_{\mathrm{s}}(x)\big)\right|$$

$$\leq \left|\overline{\varphi^{\mathrm{d}*\mathrm{d}}}\big(f_{\mathrm{s}}(x)\big) - \varphi^{\mathrm{d}*}\big(f_{\mathrm{s}}(x)\big)\right| + \left|\varphi^{\mathrm{d}*}\big(f_{\mathrm{s}}(x)\big) - \phi^*\big(f_{\mathrm{s}}(x)\big)\right|$$

$$\leq \gamma \cdot e_{\mathrm{e}} + e_{\mathrm{u}} + e_{\mathrm{v}} + e_{\mathrm{x}} + e_{\mathrm{y}} + e_{\mathrm{z}}.$$

$\square$

The following proposition summarizes the result of the preceding arguments. We note that this result extends [21, Thm. 5.3] by considering the error of extension operation for computing the expectation w.r.t. to the additive disturbance in (6a) and the approximate discrete conjugation of the input cost in (6d).

**Proposition A.8** (Error of d-CDP operation). *Let $J : \mathbb{X} \to \mathbb{R}$ be a Lipschitz continuous, convex function that satisfies the condition of Assumption 3.10-(ii). Also, let Assumptions 3.9-(i)&(ii) hold. Consider the output of the d-CDP operator $\widehat{\mathcal{T}}^{\mathrm{d}} J^{\mathrm{d}} : \mathbb{X}^{\mathrm{d}} \to \mathbb{R}$ and the discretization of the output of the DP operator $[\mathcal{T} J]^{\mathrm{d}} : \mathbb{X}^{\mathrm{d}} \to \mathbb{R}$. We have*

$$\left\|\widehat{\mathcal{T}}^{\mathrm{d}} J^{\mathrm{d}} - [\mathcal{T} J]^{\mathrm{d}}\right\|_{\infty} \leq \gamma \cdot e_{\mathrm{e}} + e_{\mathrm{u}} + e_{\mathrm{v}} + e_{\mathrm{x}} + e_{\mathrm{y}} + e_{\mathrm{z}} = \gamma \cdot e_{\mathrm{e}} + e_{\mathrm{d}},$$

With the preceding result at hand, we can now provide a bound for the difference between the fixed points of the d-CDP and DP operators. To this end, let $\widehat{J}_{\star}^{\mathrm{d}} = \widehat{\mathcal{T}}^{\mathrm{d}} \widehat{J}_{\star}^{\mathrm{d}} : \mathbb{X}^{\mathrm{d}} \to \mathbb{R}$ be the fixed point of the d-CDP operator. Recall that $J_{\star} = \mathcal{T} J_{\star} : \mathbb{X} \to \mathbb{R}$ and $J_{\star}^{\mathrm{d}} : \mathbb{X}^{\mathrm{d}} \to \mathbb{R}$ are the true optimal value function and its discretization.

**Lemma A.9** (Error of fixed point of d-CDP operator). *We have*

$$\left\|\widehat{J}_{\star}^{\mathrm{d}} - J_{\star}^{\mathrm{d}}\right\|_{\infty} \leq \frac{\gamma \cdot e_{\mathrm{e}} + e_{\mathrm{d}}}{1 - \gamma}.$$

*Proof.* By Assumptions 3.9-(ii) and 3.10-(i), the operator $\widehat{\mathcal{T}}^{\mathrm{d}}$ is $\gamma$-contractive (Theorem 3.11) and hence

$$\left\|\widehat{\mathcal{T}}^{\mathrm{d}} \widehat{J}_{\star}^{\mathrm{d}} - \widehat{\mathcal{T}}^{\mathrm{d}} J_{\star}^{\mathrm{d}}\right\|_{\infty} \leq \gamma \cdot \left\|\widehat{J}_{\star}^{\mathrm{d}} - J_{\star}^{\mathrm{d}}\right\|_{\infty}.$$

Also, notice that the composition $J \circ f$ is assumed to be jointly convex in the state and input variables for a convex function $J : \mathbb{X} \to \mathbb{R}$. Then, Assumption 3.1 implies that $J_{\star}$ is indeed Lipschitz continuous and convex. Moreover, Assumption 3.9-(ii) holds, and $J_{\star}$ is assumed to satisfy the condition of Assumption 3.10-(ii). Hence, by Proposition A.8, we have

$$\left\|\widehat{\mathcal{T}}^{\mathrm{d}} J_{\star}^{\mathrm{d}} - [\mathcal{T} J_{\star}]^{\mathrm{d}}\right\|_{\infty} \leq \gamma \cdot e_{\mathrm{e}} + e_{\mathrm{d}}.$$

Using these two inequalities, we can then write

$$\left\|\widehat{J}_{\star}^{\mathrm{d}} - J_{\star}^{\mathrm{d}}\right\|_{\infty} = \left\|\widehat{J}_{\star}^{\mathrm{d}} - \widehat{\mathcal{T}}^{\mathrm{d}} J_{\star}^{\mathrm{d}} + \widehat{\mathcal{T}}^{\mathrm{d}} J_{\star}^{\mathrm{d}} - J_{\star}^{\mathrm{d}}\right\|_{\infty}$$

$$\leq \left\|\widehat{J}_{\star}^{\mathrm{d}} - \widehat{\mathcal{T}}^{\mathrm{d}} J_{\star}^{\mathrm{d}}\right\|_{\infty} + \left\|\widehat{\mathcal{T}}^{\mathrm{d}} J_{\star}^{\mathrm{d}} - J_{\star}^{\mathrm{d}}\right\|_{\infty}$$

$$= \left\|\widehat{\mathcal{T}}^{\mathrm{d}} \widehat{J}_{\star}^{\mathrm{d}} - \widehat{\mathcal{T}}^{\mathrm{d}} J_{\star}^{\mathrm{d}}\right\|_{\infty} + \left\|\widehat{\mathcal{T}}^{\mathrm{d}} J_{\star}^{\mathrm{d}} - [\mathcal{T} J_{\star}]^{\mathrm{d}}\right\|_{\infty}.$$

$$\leq \gamma \cdot \left\|\widehat{J}_{\star}^{\mathrm{d}} - J_{\star}^{\mathrm{d}}\right\|_{\infty} + \gamma \cdot e_{\mathrm{e}} + e_{\mathrm{d}}.$$

This completes the proof. $\square$

Finally, we can use the fact that $\widehat{\mathcal{T}}^{\mathrm{d}}$ is $\gamma$-cantractive in order to provide the following bound on the error due to finite termination of the algorithm. Recall that $\widehat{J}^{\mathrm{d}} : \mathbb{X}^{\mathrm{d}} \to \mathbb{R}$ is the output of Algorithm 1.

**Lemma A.10** (Error of finite termination). *We have*

$$\left\|\widehat{J}^{\mathrm{d}} - \widehat{J}_{\star}^{\mathrm{d}}\right\|_{\infty} \leq \frac{\gamma \cdot e_{\mathrm{t}}}{1 - \gamma}.$$

*Proof.* By Assumptions 3.9-(ii) and 3.10-(i), the operator $\widehat{\mathcal{T}}^{\mathrm{d}}$ is $\gamma$-contractive (Theorem 3.11). Let us assume that Algorithm 1 terminates after $k \geq 0$ iterations so that $\widehat{J}^{\mathrm{d}} = J_{k+1}^{\mathrm{d}}$ and $\left\| J_{k+1}^{\mathrm{d}} - J_k^{\mathrm{d}} \right\|_\infty \leq e_{\mathrm{t}}$. Then,

$$
\begin{aligned}
\left\| \widehat{J}^{\mathrm{d}} - \widehat{J}_\star^{\mathrm{d}} \right\|_\infty &= \left\| J_{k+1}^{\mathrm{d}} - \widehat{\mathcal{T}}^{\mathrm{d}} J_{k+1}^{\mathrm{d}} + \widehat{\mathcal{T}}^{\mathrm{d}} J_{k+1}^{\mathrm{d}} - \widehat{J}_\star^{\mathrm{d}} \right\|_\infty \\
&\leq \left\| J_{k+1}^{\mathrm{d}} - \widehat{\mathcal{T}}^{\mathrm{d}} J_{k+1}^{\mathrm{d}} \right\|_\infty + \left\| \widehat{\mathcal{T}}^{\mathrm{d}} J_{k+1}^{\mathrm{d}} - \widehat{J}_\star^{\mathrm{d}} \right\|_\infty \\
&= \left\| \widehat{\mathcal{T}}^{\mathrm{d}} J_k^{\mathrm{d}} - \widehat{\mathcal{T}}^{\mathrm{d}} J_{k+1}^{\mathrm{d}} \right\|_\infty + \left\| \widehat{\mathcal{T}}^{\mathrm{d}} J_{k+1}^{\mathrm{d}} - \widehat{\mathcal{T}}^{\mathrm{d}} \widehat{J}_\star^{\mathrm{d}} \right\|_\infty \\
&\leq \gamma \cdot \left\| J_k^{\mathrm{d}} - J_{k+1}^{\mathrm{d}} \right\|_\infty + \gamma \cdot \left\| J_{k+1}^{\mathrm{d}} - \widehat{J}_\star^{\mathrm{d}} \right\|_\infty \\
&\leq \gamma \cdot e_{\mathrm{t}} + \gamma \left\| \widehat{J}^{\mathrm{d}} - \widehat{J}_\star^{\mathrm{d}} \right\|_\infty,
\end{aligned}
$$

where for the second inequality we used the fact that $\widehat{\mathcal{T}}^{\mathrm{d}}$ is a contraction. $\square$

The inequality (7) is then derived by combining the results of Lemmas A.9 and A.10.

## Acknowledgments

This research is part of a project that has received funding from the European Research Council (ERC) under the grant TRUST-949796. The authors are also grateful to anonymous reviewers for their comments concerning the three remarks in Section 5.