# OpenReview forum: "Fast Approximate Dynamic Programming for Infinite-Horizon Markov Decision Processes"
_NeurIPS.cc/2021/Conference — NeurIPS 2021 Poster_

### Official Review · Reviewer_53cU · 2021-07-16

**Rating:** 7
**Confidence:** 3

**Summary:**

The paper develops an approximate value iteration method for an infinite-horizon, discounted-cost Markov Decision Processes (MDPs) that satisfy a given set of regulatory assumptions. The idea is to work on a dual space that replaces the DP value function by a reformulation written in terms of biconjugate operations. The authors present an algorithm that applies the conjugate operators iteratively, and present convergence and other structural results associated with the resulting errors. Numerical results compare the proposed approach against a traditional value iteration and a variant that generated dynamic discretization grids.

**Limitations And Societal Impact:**

The paper indeed lacks a discussion of the limitations of their methodology. In particular, authors could be clearer with respect to trade-offs of the conjugate formulation and the traditional VI algorithm.

**Main Review:**

**Originality**

The paper is an extension the previous work by Kolarijani and Esfahani (2020), which lays out the fundamental ideas underlying conjugate dual operators in MDPs and has similar error-type analysis. The key contribution of this paper is that it extends those ideas in different ways, incorporating stochastic dynamics (more briefly discussed in that paper) and involves a more general class of MDPs (e.g., infinite-horizon and resulting numerical approximations).

While such extensions are no doubt a challenging and admirable endeavor, their contribution is much more technical and builds upon the novelty of that previous work. Assuming the previous work is a separate publication and the current one does not encompass that one, my concern is that the originality of this paper is more restricted to an MDP audience, given the (usual) wide array of assumptions that must be in place for this approach to be applicable.

**Quality** and **Clarity**

The paper is excellently well written and rigorous. It is, however, quite dense to read: some concepts I was only able to grasp after reading the "extended manuscript" in the supplemental material, which includes more examples (I appreciate the authors for that). Nonetheless, it is overall a very pleasant and fluid read. There are also six pages of technical proofs, some involved but most using standard contraction and similar operations - I did my best to verify them and they seem correct.


**Significance**

As mentioned before, my major concern is that the paper seems to be more suitable to a more specialized audience of NeurIPS due to its strong technical aspects. The numerical simulations are very limited, and in my view are not very effective in demonstrating the practical relevance of their new approach.

However, the paper reads very well and the conjugate reformulation is a relevant (and impactful) contribution. It does reduce the VI complexity significantly and can be of potential use in different dynamic programming models.

**Questions and Minor Points**

- How restrictive is assumption 3.1-(ii)?
- In Assumption 3.5: "feasibile" --> "feasible"

**Time Spent Reviewing:**

6

---

> ### Author Response · Authors · 2021-08-09
> **Response to Reviewer 53cU**
>
> We appreciate the reviewer’s constructive criticism.
> We also thank the reviewer for bringing some typos to our attention.
> Here is our response to his/her comments.
>
> ## 1. Scope of the study
> We acknowledge the reviewers concern.
> Indeed, the paper can come across as technically and notationally demanding.
> We tried our best to adopt informative notational conventions to help the reader.
> Regarding the audience of NeurIPS, we believe that the basic idea for complexity reduction introduced in this study is unconventional and novel, and it can be also of interest in Reinforcement Learning community.
>
> ## 2. Limited numerical simulations
> Following reviewer's suggestion, we implemented and plan to include the numerical results for a system of a higher dimension (namely, unstable batch reactor, with $n=4$ states and $m=2$ inputs, borrowed from "Computer aided control system design" by Rosenbrock).
> Here are the results we derived for discrete state and input spaces of size $X = 11^4$ and $U=11^2$, respectively:
>
> | Algorithm | Running time (sec) | Average cost (50 runs) |
> |:----:|:----:|:----:|
> | VI | 9125 | 24.6 |
> | CVI | 65 | 36.6 |
> | CVI-d | 119 | 22.2 |
>
> ## 3. How restrictive is Assumption 3.1-(ii)?
> The reviewer is pointing out the most important limitation of the proposed CVI algorithm.
> The requirement of Assumption 3.1-(ii) can indeed restrict the application of CVI.
> This class of deterministic dynamics includes linear systems $x^+ = Ax+Bu$, which historically has been one of the main focuses in optimal control (the reason being the possibility of linearization around the nominal operating point of the nonlinear system).
> Moreover, for input-affine dynamics of the form $x^+ = f_{\mathrm{s}}(x) + f_{\mathrm{i}}(x) \cdot u$, we can also use CVI, after linearization of only the input dynamics $f_{\mathrm{i}}(x)$ around the nominal operating state of the system.
> Also, for certain electro-mechanical systems, the control action has as an additive impact on the evolution of the state variables (because usually the control action appears as the generalized force in Euler-Lagrange equation).
> This is for example the case for the inverted pendulum (as shown in our numerical example) and the ball-and-beam system.
>
> ## 4. Limitations of CVI and trade-off compared to VI
> As the reviewer also noticed, the better time complexity indeed comes at the expense of restricting the class of problem.
> In particular, there are two main conditions that must be satisfied in order to be able to apply the CVI algorithm:
> 1. the dynamics must be of the form $x^+ = f_{\mathrm{s}}(x)+Bu+w$;
> 2. the cost $C(x,u) = C_{\mathrm{s}}(x)+C_{\mathrm{i}}(u)$ must be separable, which is commonly the case at least in engineering applications.
>
> In this regard, let us clarify that regardless of convexity assumptions, we can still apply the CVI algorithm to find the _convex envelop_ of the value function.
> However, since CVI essentially solves the dual problem, for non-convex problems, it suffers from a non-zero duality gap.
>
> Based on our simulation results, when we compare CVI and VI, we also notice a trade-off between computational complexity (Fig. 1b) and control action quality (Fig. 1c): While CVI has a lower computational cost, VI generates better control actions.
> This is the reason behind introducing CVI-d (with dynamic state dual grid),
> which with a small extra computational burden compared to CVI, achieves almost the same performance as VI when it comes to the quality of control actions.
>
> We thank the reviewer for bringing this issue to our attention. We plan to add this discussion to the final version of the article.

---

> > ### Comment · Reviewer_53cU · 2021-08-23
> > **Feedback**
> >
> > Thank you for the review and clarifying those points, it has been very helpful.

---

### Official Review · Reviewer_mb6w · 2021-07-16

**Rating:** 5
**Confidence:** 2

**Summary:**

This paper proposed a new value iteration algorithm (CVI) for optimal control of stochastic system with continuous state space. The proposed algorithm discretizes the state space and performs the value iteration step in the "conjugate domain" to reduce one-step computational complexity. It was shown the proposed algorithm converges under conditions on the stochastic system, discretization, and etc. The proposed algorithm CVI, its variant CVI-d are compared to classical value iteration (VI) over a noisy inverted pendulum experiment.

**Limitations And Societal Impact:**

See the main comments above.

**Main Review:**

The paper is well written and the language is precise.

The main benefit of the propose algorithm is the reduced per-step complexity (Thm 3.10). However, no guarantees on how many iterations will be needed for the algorithm to stop (see Alg.1 line 4) given the error tolerance parameter e_t in Algo 1. Therefore, Thm 3.10 along with Thm 3.11 cannot really justify the claim of "fast approximate dynamic programming". More on this point in the experiment part below.

I understand, in the inverted pendulum experiment, the CVI algorithm needs less iteration than VI, but we need more evidence to be convinced this is a fast value iteration algorithm. The experiment also brings another question, Fig.1 c shows the resulting greedy policy learned from CVI is worse than classic VI in all state grid settings; it is much worse with coarse grids. The variant of CVI, CIV-d (with dynamic grid in dual space), was able to find better policies, but there is no guarantee for that.

For originality, I am not very familiar with literature in this area (sample based value iteration). It will help a lot if the authors clarify the technical contributions of the current algorithm compared to d-CDP from [18, reference from the paper]. Since the authors claim that the current algorithm extends d-CDP from infinite-horizon, discounted cost to stochastic dynamics.

**Time Spent Reviewing:**

4

---

> ### Author Response · Authors · 2021-08-09
> **Response to Reviewer mb6w**
>
> We appreciate the reviewer’s constructive criticism.
> Here is our response to his/her comments.
>
> ## 1. Rate of convergence and justification of the term "fast"
> Let us first clarify that Theorem 3.9 shows that CVI is linearly convergent with rate $\gamma$.
> This indeed provides us with an upper bound for the number of iterations required for the algorithm to terminate: for a given termination bound $e_{\mathrm{t}}$, the algorithm terminates in $\log\frac{\Vert J_1^{\mathrm{d}} - J_0^{\mathrm{d}} \Vert_{\infty}}{e_{\mathrm{t}} }/\log\frac{1}{\gamma}$ iterations.
> We note that, this convergence rate is the generic case for (convergent) VI algorithms and stems from the fact that the Bellman operator is $\gamma$-contractive; in particular, we do _not_ claim a faster convergence rate compared to VI algorithms.
> However, as the reviewer also clarified, CVI reduces the _per iteration complexity_, which combined with the same convergence rate as VI, leads to a lower total computational cost compared to VI (this effect is shown in Fig. 1b). So, the term ``fast'' refers to this reduction.
>
> We would like to also note that, in our _numerical simulations_, we have also observed a faster convergence rate for CVI, similar to the one reported in Fig. 1a.
> The following table shows similar results for a system of a higher dimension (namely, unstable batch reactor, with $n=4$ states and $m=2$ inputs, borrowed from "Computer aided control system design" by Rosenbrock), for discrete state and input spaces of size $X \times U = 11^4 \times 11^2$:
>
> | Algorithm | Termination iteration | Running time (sec) | Average cost (50 runs) |
> |:----:|:----:|:----:|:----:|
> | VI | 100 | 9124 | 24.6 |
> | CVI | 33 | 65 | 43.5 |
> | CVI-d | 58 | 119 | 22.2 |
>
> Once again, let us clarify that our theoretical results only guarantee linear convergence with rate $\gamma$, but our speculation is that under certain regimes, the approximate discrete conjugation in the last step of CVI also becomes a contraction (as of now, we can only show the last step is non-expansive; see Lemmas 5.1 and 5.2 of the extended manuscript in the supplementary material).
>
> ## 2. The performance of greedy policies with respect to CVI and CVI-d
> The reviewer's assessment is correct.
> Indeed, when we compare CVI and VI, we notice a trade-off between computational complexity (Fig. 1b) and control action quality (Fig. 1c): While CVI has a lower computational cost, VI generates better control actions.
> This is the reason behind introducing CVI-d (with dynamic state dual grid), which with a small extra computational burden compared to CVI, achieves almost the same performance as VI when it comes to the quality of control actions.
> Regarding theoretical guarantees on the performance of greedy policies with respect to output of approximate VI algorithms, we do not provide such guarantees.
> To the best of our knowledge, theoretical guarantees are often of the same type as the one provided in Theorem 3.11, which bound the difference between the output of approximate VI algorithms and the true optimal value functions.
>
> ## 3. Originality of the work compared to [18]
> The technical contribution is three-fold:
> 1. From deterministic to stochastic dynamics -- in particular, the term $e_{\mathrm{e}}$ in our error analysis (Theorem 3.11) correspond to this extension and do not appear in the corresponding result (Theorem 5.3) in [18].
> 2. From analytical to numerical computation of the conjugate of input cost -- in particular, the terms $e_{\mathrm{u}}$ and $e_{\mathrm{v}}$ in our error analysis (Theorem 3.11) correspond to this extension and do not appear in the corresponding result (Theorem 5.3) in [18].
> 3. From finite-horizon to infinite-horizon problems -- in particular, we not only have to consider the convergence of the proposed algorithm (Theorem 3.9),
> but also, in our error analysis, we now have to consider the error of multiple applications of the d-CDP operation, until the termination of the algorithm, with respect to the true optimal value $J_\star$ (Lemmas 5.9 and 5.10 of the extended manuscript in the supplementary material).

---

### Official Review · Reviewer_Hfvo · 2021-07-17

**Rating:** 6
**Confidence:** 4

**Summary:**

This paper studies the problem of dynamic programming with continuous state and action spaces.
The authors first propose the VI which naively discretizes the state and action space and performs the classical value iteration. The authors use this method as a baseline for comparison with CVI developed in this paper.
Regarding the CVI, this algorithm performs VI in the the conjugate domain. In particular, by taking the dual of the objective function, we take the problem to the dual space. By doing so, instead of taking minimum over all the actions, the problem is reduced to a summation. Hence, the complexity of the CVI is better than the VI in the order of the cardinality of the action space.

**Limitations And Societal Impact:**

- Lack of comparison with the related works.

**Main Review:**

- Since the J function can be bounded above by O(1/(1-gamma)), I do not understand the purpose of the Theorem 3.11. Isn't it result in a trivial bound?
- Comparison in the experimental result is not sufficient. For instance, can the authors compare the performance of their algorithm with the ones in [8] and [22]? Specifically section 8.1 in [22] that talks about aggregation techniques to deal with the curse of dimensionality, and references in section 8.1 in the book. Also look at table 4.4 in [22] for a variety of the references to compare with.
- There is no Assumption 3.7-(ii), there is no Fig 1-d.

**Time Spent Reviewing:**

10

---

> ### Author Response · Authors · 2021-08-09
> **Response to Reviewer Hfvo**
>
> We appreciate the reviewer’s constructive criticism.
> We also thank the reviewer for bringing some typos to our attention.
> Here is our response to his/her comments.
>
> ## 1. Practicality of the error bound of Theorem 3.11
> We agree with the  reviewer that the output of the CVI algorithm and the true value function $J_\star$ are bounded above by $\mathcal{O}(1/(1-\gamma))$.
> However, the provided error analysis is particularly used for providing specific guidelines/procedures for construction of the grids $\mathbb{Y}^{\mathrm{g}}$, $\mathbb{V}^{\mathrm{g}}$ and $\mathbb{Z}^{\mathrm{g}}$ in Section 3.4.
> Let us elaborate on this.
> The error bound of Theorem 3.11 most importantly captures the error due to discretization of the primal and dual state and input spaces ($e_\mathrm{d}$).
> In particular, it shows the consistency of the proposed algorithm in the sense that as we use finer discretizations, the output of the algorithm approaches $J_\star$.
> Moreover, through this error analysis, we were able to identify a set of conditions for the grids $\mathbb{Y}^{\mathrm{g}}$, $\mathbb{V}^{\mathrm{g}}$ and $\mathbb{Z}^{\mathrm{g}}$ in order to minimize the discretization error.
> As we mentioned above, these conditions are then translated to specific guidelines/procedures for construction of these grids.
>
> ## 2. Comparison with the related works, e.g., aggregation method
> Let us first clarify that the techniques used in the studies mentioned by the reviewer (for sampled-based VI algorithms) and the technique used in CVI are not necessarily competing. Actually, they can rather be complementary.
> Let us elaborate on this.
> The approximation techniques used in sampled-based VI algorithms for handling continuous state spaces are aimed to address the fact that the Bellman equation is infinite-dimensional (for continuous state spaces) by transforming it to a finite-dimensional problem.
> In general, the time complexity of these methods depend on the product of the cardinalities of state and action spaces, i.e., a complexity of $\mathcal{O}(XU)$.
> The proposed CVI algorithm, on the other hand, focuses on reducing this time complexity to $\mathcal{O}(X+U)$, by avoiding the minimization over input in each iteration.
> Take, for example, the aggregation technique proposed in Section 8.1 of [22] that leads to a piece-wise constant approximation of the value function. Below, we report the result of numerical simulations of a system (namely, unstable batch reactor, with $n=4$ states and $m=2$ inputs, borrowed from "Computer aided control system design" by Rosenbrock) using this type of state space aggregation with and without CVI (aggregation of size $15^4$ for state space and discrete input space of size $15^2$):
>
> | Algorithm | Running time (sec) | Average cost (50 runs) |
> |:----|:----:|:----:|
> | Aggregation | 50284 | 22.3 |
> | Aggregation + CVI | 472 | 41.0 |
> | Aggregation + CVI-d | 507 | 20.3 |
>
> We thank the reviewer for bringing this connection to our attention. We plan to include this discussion in the final version of our paper.

---

> > ### Comment · Reviewer_Hfvo · 2021-08-31
> > **thank you for response**
> >
> > Thank you for your response. Indeed the guideline you provide for a fine discretization can be helpful.
> > Regarding the comparison, can you also include the result with just CVI and CVI-d, without aggregation?

---

> > > ### Author Response · Authors · 2021-09-02
> > > **Response to Reviewer Hfvo**
> > >
> > > We thank the reviewer for his/her positive comments.
> > >
> > > Below, you can find the result of numerical simulations of the same unstable batch reactor using only CVI and CVI-d algorithms (with discrete state and input spaces of sizes $X = 15^4$ and $U=15^2$, respectively, i.e., the same size as in the aggregation technique used above).
> > > In this particular example, we see lower costs when we only use CVI and CVI-d, compared to the case where these algorithms are combined with the aggregation technique described above. However, this need not be the same in general.
> > >
> > > | Algorithm | Discrete states ($X$) | Discrete inputs ($U$) | Running time (sec) | Average cost (50 runs) |
> > > |:----|:----:|:----:|:----:|:----:|
> > > | CVI | $15^4$ | $15^2$ | 316 | 29.0 |
> > > | CVI-d | $15^4$ | $15^2$ | 384 | 17.9 |
> > > | Aggregation + CVI | $15^4$ | $15^2$ | 310 | 41.0 |
> > > | Aggregation + CVI-d | $15^4$ | $15^2$ | 387 | 20.3 |
> > >
> > > NOTE 1: All four algorithms have a time complexity of $\mathcal{O} (X+U)$, leading to almost the same running times reported above.
> > >
> > > NOTE 2: Since the results reported in the preceding table were produced using a different machine, we also reproduced the result of _Aggregation + CVI_ and _Aggregation + CVI-d_ for consistency (the costs are the same since we are using the same set of random initial conditions for the 50 runs).

---

### Official Review · Reviewer_9wHP · 2021-07-17

**Rating:** 7
**Confidence:** 3

**Summary:**

This paper presents a fast method for solving certain classes of optimal control problems, by implementing a value iteration (VI) algorithm that takes advantage of convex duality.
The basic idea is to take advantage of the fact that for two functions $f_1, f_2$, we have $(f_1 \square f_2)^* = f_1^* + f_2^*$, where $f^*$ denotes the Legendre-Fenchel transform of $f$, and $(f_1 \square f_2)(x) = \inf \\{f_1(x_1) + f_2(x_2) : x_1 + x_2 = x \\} $ denotes the infimal convolution of $f_1, f_2$.
Thus, the slow $\inf$ operation can be replaced with a fast addition operation.
The paper provides analyses of the convergence, time complexity, and error of the algorithm.
When compared to a "naive" VI algorithm in which each iteration takes $O(X U)$ (where $X$, $U$ are the grid sizes of the state and input spaces, respectively), the algorithm presented in this paper can achieve $O(X+U)$.
Finally, the paper provides a numerical example implemented in MATLAB.

**Limitations And Societal Impact:**

Limitations

1) The paper clearly states the restriction on the class of problems that the algorithm applies to: principally, the stage cost function $C(x, u)$ must be jointly convex.
The baseline comparison algorithm is a naive version of VI.
A single iteration of VI must solve the Bellman equation
$$
  J(x) = \min_u \\{ C(x, u) + \gamma E\[ J(g(x, u, w) \] \\}
$$
If $X, U$ are the grid sizes of the state space and input space, respectively, then a brute-force implementation will search through every possible $u$ for every possible grid point value of $x$, giving $O(X U)$ time complexity.
However, since it is assumed that $C(x,u)$ is convex, it is not hard to show that the value function $J$ must also be convex, and thus faster optimization algorithms that use gradient information can be used.
Comparing the performance of CVI against a gradient-aware algorithm would provide a more realistic comparison for the speed advantage of CVI.


2) The numerical example provided is a two-dimensional noisy inverted pendulum, which may be too simplistic to provide an indication of the potential strengths and weaknesses of the algorithm.
It would be useful to see how the algorithm handles more complicated problems, and how some practical issues in implementing the algorithm are addressed.
For example, when constructing the approximate Legendre-Fenchel transform $f^*$, the grid space of $f^*$ must be large enough to encompass the full range of slopes of the primal function $f$.
For certain functions, this range may be very large, extending to infinity (e.g. $\log(x)$ or $x^{-\alpha}$).
Especially in higher dimensions, it is non-trivial to construct an approximation for $f^*$ that encompasses such a large range while maintaining accuracy.

**Main Review:**

This paper presents an fast algorithm that is probably novel to most researchers and practitioners who need to solve optimal control problems.
The ideas first appeared in Bellman & Karush (1962), and the algorithm was most recently presented and analyzed in Kolarijani and Esfahani (2020).
The latter paper applied the algorithm to finite-horizon optimal control problems with deterministic input-affine dynamics, and provided detailed analyses of the algorithm's time complexity and error properties.
This current paper goes beyond that paper in applying the algorithm to discounted, infinite-horizon problems with stochastic dynamics.
The marginal contribution of this paper is incremental; however, the novelty and performance of the algorithm may be of interest to researchers who need to solve problems that fall within the class handled by this algorithm.

The paper is well-written, clear and concise. The mathematical details seem correct (though I did not check them closely). I have addressed some limitations of the paper in the "Limitations" section below.

Overall, the paper is interesting and presents a novel algorithm with the potential for large performance gains in solving optimal control problems. The applicability of the algorithm to more complicated, real-world problems could be an interesting and useful research question.

Bellman and Karush (1962). Mathematical programming and the maximum transform. Journal
of the Society for Industrial and Applied Mathematics, 10(3):550-567.

Kolarijani and Esfahani (2020). Fast approximate dynamic programming for input-affine
409 dynamics. preprint arXiv:2008.10362, 2020.

**Time Spent Reviewing:**

6

---

> ### Author Response · Authors · 2021-08-09
> **Response to Reviewer 9wHP**
>
> We appreciate the reviewer’s constructive criticism. Here is our response to his/her comments.
>
> ## 1. Application of gradient-aware algorithms
> We agree with the reviewer that CVI may not lead to a significant complexity reduction compared to a VI that utilizes gradient-aware algorithms.
> However, we note that application of gradient-based minimization over $u$ in sampled-based VI algorithms usually involves solving a difficult non-convex problem.
> Moreover, this minimization problem again must be solved for each discrete (sample) point $x \in \mathbb{X}^{\mathrm{d}}$ in each iteration, while application of CVI avoids solving the minimization over $u$ in each iteration.
>
> Let us further clarify these two points.
> For simplicity, let us focus on deterministic dynamics.
> As the reviewer suggests, the idea is to implement the following iteration,
> $$J_{k+1}^{\mathrm{d}}(x) = \min_u \{ C(x,u)+\gamma \widetilde{J^{\mathrm{d}}_k}(f(x,u)) \}, \quad x \in \mathbb{X}^{\mathrm{d}},$$
> where $\widetilde{J^{\mathrm{d}}_k} : \mathbb{X} \rightarrow \mathbb{R}$ is a differentiable approximation/extension of the output of the previous iteration $J^{\mathrm{d}}_k: \mathbb{X}^{\mathrm{d}} \rightarrow \mathbb{R} $ .
> In this regard, recall that the output of each iteration is the sampled function $J^{\mathrm{d}}_k$,
> which is used to compute (the parameters of) the approximator $\widetilde{J^{\mathrm{d}}_k}$ via some form of regression (we have to fit $\widetilde{J^{\mathrm{d}}_k}$ to $J^{\mathrm{d}}_k$).
> Then, even if the true value function $J$ is convex (for instance, since the dynamics is linear, i.e., $f(x,u)=Ax+Bu$), for the gradient-based algorithm to converge to the global minimum, we require the extension $\widetilde{J^{\mathrm{d}}_k}(f(x,u))$ to be convex in $u$.
> The is almost always not the case (e.g., take kernel-based approximations and neural networks).
> This is why in MDP and RL literature, it is actually quite common to consider a finite action space in the first place.
>
> Nevertheless, let $M$ denote the complexity of solving the minimization over $u$ using a gradient-based algorithm.
> Note that $M$ in particular contains the complexity of each evaluation of the gradient of the appoximator $\widetilde{J^{\mathrm{d}}_k}$  (this depends on the number of the parameters used in this approximator, which is expected to increase exponentially with dimension of the state space).
> Then, the iterative algorithm described above requires $\mathcal{O}(XM)$ operations per iteration for finding , where $X$ is the size of sampled state space.
> Moreover, there is the computational cost of the regression within each iteration, which itself is of $\mathcal{O}(XE)$, where $E$ denotes the complexity of each evaluation of the approximator $\widetilde{J^{\mathrm{d}}_k}$  (once again, this depends on the number of the parameters used in this approximator, which is expected to increase exponentially with dimension of the state space).
> On the other hand, using the same sampled state space $\mathbb{X}^{\mathrm{d}}$, CVI has a one-time compilation complexity of $\mathcal{O}(X+U)$ (which reduces to $\mathcal{O}(X)$ if the conjugate of input cost is analytically available)
> and a per iteration complexity of  $\widetilde{\mathcal{O}}(X)$; CVI, in worst case, avoids solving the minimization problem in each iteration.
> So, the per iteration complexity of CVI is most probably better than that of a gradient-based VI (of course, it generally depends on the constant factors and the particular approximation and regression techniques used in VI).
> In this regard, let us also note that in our simulations, to make sure that we are providing a fair comparison, we particularly chose nearest neighbour and multi-linear extensions so that there is no need for regression in VI.
>
> ## 2.1. Simplicity of the numerical example
> Following the reviewer's suggestion, we implemented and plan to include the numerical results for a system of a higher dimension (namely, unstable batch reactor, with $n=4$ states and $m=2$ inputs, borrowed from "Computer aided control system design" by Rosenbrock).
> Here are the results we derived for discrete state and input spaces of size $X = 11^4$ and $U=11^2$, respectively:
>
> | Algorithm | Running time (sec) | Average cost (50 runs) |
> |:----:|:----:|:----:|
> | VI | 9125 | 24.6 |
> | CVI | 65 | 36.6 |
> | CVI-d | 119 | 22.2 |
>
> ## 2.2. The dual grid for non-Lipschitz functions
> Let us first clarify that because of Lipschtiz-continuity of the cost and dynamics (Assumption 3.1), the value functions is also Lipschitz-continuous.
> Nevertheless, the reviewer is raising an important issue: since the size $Y$ of the dual grid  is controlled by the size $X$ of the discrete state space (Assumption 3.6), as the range of slopes of the value function $J_\star$ increases, the error due to discretization of the dual state space increases (same concerns hold for the construction of the grid $\mathbb{V}^{\mathrm{g}}$).
> We tried to partially address this issue by proposing the dynamic approach for construction of the state dual grid $\mathbb{Y}^{\mathrm{g}}$ by focusing on the range of slopes of $J^{\mathrm{d}}_k$  in each iteration in order to minimize the discretization error of that same iteration $k$.
> However, when the cost function has a large Lipschitz constant, as noticed by the reviewer, even CVI-d can fail to provide a good approximation.
> Below we report the result of the numerical simulation of the same unstable batch reactor with the following cost (as opposed to quadratic cost considered for the results presented earlier)
>
> $$C(x,u) = -\\frac{4}{1+\\epsilon} + \\sum_{i} \frac{1}{1+\\epsilon-|x_i|} -\\frac{2}{2+\epsilon} + \sum_{j} \\frac{1}{2+\epsilon-|u_j|},\\quad |x_i| \\leq 1,\ |u_j| \\leq 2. $$
>
> Clearly, as $\epsilon \rightarrow 0$, we increase the range of slopes of the cost function.
> The following table shows the results of our simulations for $\epsilon = 0.01$:
>
> | Algorithm | Running time (sec) | Average cost (50 runs) |
> |:----:|:----:|:----:|
> | VI | 7669 | 33.9 |
> | CVI | 55 | 73.5 |
> | CVI-d | 90 | 74.0 |
>
> As can be seen, the quality of the greedy action generated by CVI-d also deteriorates in this case.
> We thank the reviewer for bringing this issue to our attention. We plan to add this discussion in the final version of the paper.

---

> > ### Comment · Reviewer_9wHP · 2021-08-21
> > **Comment on Response to Reviewer 9wHP**
> >
> > Thank you for the response, this mostly addresses the issues raised in my review.
> >
> > With regard to point 1 (Application of gradient-aware algorithms), I agree that gradient-aware algorithms only make sense when the function approximation preserves convexity of the underlying function, and this is not common in the MDP and RL literature.
> > I would point out, however, that in fields such as economics, where it is common to assume convexity (or concavity) of the cost and value functions, shape-preserving function approximations (such as linear interpolation) become more valuable and are frequently used, enabling the use of gradient-aware algorithms.
> >
> > After reading the other reviews and authors' responses, I feel that the authors have generally responded to the specific issues raised in the reviews.
> > In general, I think there would be more interest if the authors could suggest some more potential applications in the fields relevant to the audience of this conference, that might benefit from the CVI algorithm.

---

> > > ### Author Response · Authors · 2021-08-25
> > > **Response to Reviewer 9wHP**
> > >
> > > We thank the reviewer for his/her positive comments.
> > >
> > > Regarding point 1, thanks to the reviewer's comment, we have looked at the possibility of using a gradient-based algorithm for minimization over $u$ within the CVI algorithm.
> > > While such a possibility indeed exists,
> > > it unfortunately does not allow us to benefit from the computational advantage of CVI.
> > > In what follows we elaborate on this point.
> > > As the reviewer suggested, if the true value function is convex and its approximation is also convex in $u$, one can employ a gradient-based algorithm for minimization over $u$.
> > > Now, the very important observation is that CVI actually uses a convex approximation of the value function.
> > > In particular, one can show that, in each iteration $k \geq 0$, CVI solves (for deterministic dynamics)
> > > $$ J_{k+1}^{\\mathrm{d}}(x) =  C_{\\mathrm{s}}(x) +  \\min_u \\left\\{ C_{\\mathrm{i}}(u)+\\gamma \\cdot \\max_{y \\in \\mathbb{Y}^{\\mathrm{g}}} \\left[ \\left\\langle y, f_{\\mathrm{s}}(x) + Bu \\right\\rangle -  J_k^{\\mathrm{d}*\\mathrm{d}}(y) \\right] \\right\\}, \\quad x \\in \\mathbb{X}^{\\mathrm{d}},$$
> > > where $J_k^{\\mathrm{d}*\\mathrm{d}}(y) = \\max_{x \\in \\mathbb{X}^{\\mathrm{d}}} \\left\\{ \\left\\langle x, y \\right\\rangle -  J_k^{\\mathrm{d}}(x) \\right\\}$ for $y \in \mathbb{Y}^{\mathrm{g}}$, is the discrete conjugate of the output of the previous iteration.
> > > Then, it is not hard to see that a subgradient of the objective of the minimization with respect to $u$ can be computed using $\mathcal{O} (Y)$ operations:
> > > for a given $u$, assuming we have access to the subdifferential $\partial C_{\mathrm{i}} (u)$, the subdifferential of the objective function is $\partial C_{\mathrm{i}} (u) + \gamma \cdot B^{\top} y_u$, where
> > > $$y_u = \\mathrm{arg} \\max_{y \\in \\mathbb{Y}^{\\mathrm{g}}} \\left[ \\left\\langle y, f_{\\mathrm{s}}(x) + Bu \\right\\rangle -  J_k^{\\mathrm{d}\*\\mathrm{d}}(y) \\right].$$
> > > This, in turn, leads to a time complexity of $\mathcal{O} (XY)$ for computing $J_{k+1}^{\\mathrm{d}}: \\mathbb{X} \\rightarrow \\mathbb{R}$.
> > > The complexity of the proposed CVI, on the other hand, is of $\tilde{\mathcal{O}} (X+Y)$, assuming the grid $\mathbb{Z}^{\mathrm{g}}$ is of the same size as $\mathbb{Y}^{\mathrm{g}}$ and we have access to $C^{\*}_{\mathrm{i}}$.
> > > We again thank the reviewer for bringing this connection to our attention. We will include this discussion in the final version of paper.
> > >
> > > Regarding potential application of the proposed CVI algorithm, we view the relevance of our paper to the  NeurIPS audience in two ways:
> > > First, the proposed CVI algorithm addresses the optimal control problem of systems with _continuous_ state spaces, and hence it can be particularly of interest for applications in engineering (control and communication), operations research (portfolio optimization), and economics.
> > > Second, from an algorithmic point of view, the idea for complexity reduction introduced in this
> > > study can be potentially combined with and further improve existing approximate VI algorithms that solely focus on transforming infinite-dimensional optimizations in MDP/RL problems into computationally tractable ones.
> > > This complexity reduction is indeed achieved by employing the LLT algorithm and avoiding the minimization within each iteration.

---

### Official Review · Reviewer_cP9r · 2021-07-18

**Rating:** 6
**Confidence:** 4

**Summary:**

The authors consider value iteration [TJ](x) = min_u [c(x,u) + gamma E J(x’)] for problems where the transition function and cost functions are additively separable: c(x,u)=c1(x)+c2(u) and x’=f1(x)+f2(u) + w (Assumptions 3.1). Furthermore, f1(x) is Lipschitz continuous, f2(u) is linear, c1(x) is convex and Lipschitz continuous, and c2(u) is linear in u. The disturbance w has a finite support (Assumption 2.1). The state space for x and decision space for u are continuous. Thus, the authors consider problems such that [TJ](x) = c1(x) + min_u [c2(u) + gamma sum_i  p_i J(f1(x)+Bu+w_i)]. A problem that fits this description is the stabilization of an inverted pendulum with discretized disturbances, studied in Section 4. With this choice of assumptions, and assuming the value function is convex (which holds true if f1(x) is linear or linearized, that is, if x’=Ax+Bu+w), convex duality can be applied to the minimization problem over (u,x”) given x and subject to x”=Ax+Bu, which gives another representation of the optimization problem, based on the convex conjugate of c1 and the convex conjugate of e(x”)=gamma E J(x”+w).

The authors propose and study an algorithm that relies on computing these convex conjugate functions numerically in order to provide an alternative way to the computation of the value function J. They establish error bounds (Theorem 3.13) in terms of theoretical quantities of the problem data.

**Limitations And Societal Impact:**

The contributions are essentially mathematical. The work does not appear to have potential negative societal impact.

**Main Review:**

The results appear to be new. While dualization is a classical concept in optimization, and there is preliminary work on applying these ideas to deterministic systems, the extension to stochastic systems and the results on error bounds appear to be new.
The work is technically sound given the assumptions. Matlab codes are provided. In the opinion of this reviewer, the limitations of the work are
(i) the scope of problems that can be treated by the proposed approach, compared to classical VI. For instance, the authors state that the curse of dimensionality from classical VI has exponential rate m+n, while their method has exponential rate max(m,n). But classical VI does not rely on a convex value function assumption that restrict the admissible class of transition and cost functions. The test problem presented in the numerical section does not itself satisfy all the assumptions assumed by the theory (as pointed out by line 463 of the extended manuscript).
(ii) the practicality of the bounds, compared to classical VI. The error bounds of the proposed approach do not seem to easily lead to a stopping criterion as they rely non-universal constants and quantities that seem difficult to evaluate or bound.
(iii) it is not clear if the assumption that the transition function is linear to ensure convexity of the value function and fully justify a dualization approach is needed for the approach to be sound. The numerical examples provided actually use a system where control techniques using a linearization of the transition function are known to work well.
(iv) the fundamental reason that would explain that solving in the dual space is easier than solving in the primal space is not fully apparent. The test problem has a unidimensional decision variable, so it is difficult to draw strong conclusions on scaling from there.
The submission is clearly written. The results are significant, although they seem to be most relevant for certain classes of control problems that are already well solved (it is hard from the current manuscript to identify problems that would significantly gain from the new approach).


**Time Spent Reviewing:**

4

---

> ### Author Response · Authors · 2021-08-09
> **Response to Reviewer cP9r**
>
> We appreciate the reviewer’s constructive criticism. Here is our response to his/her comments.
>
> ## (i.1) Required assumptions to implement CVI
> The better time complexity of CVI indeed comes for a particular class of problems.
> However, let us clarify that there are two necessary conditions for implementation of CVI algorithm:
> 1. the dynamics must be of the form $x^+ = f_{\mathrm{s}}(x)+Bu+w$;
> 2. the cost $C(x,u) = C_{\mathrm{s}}(x)+C_{\mathrm{i}}(u)$ must be separable, which is commonly the case at least in engineering applications.
>
> In particular, regardless of convexity assumptions, we can still apply the CVI algorithm to find the _convex envelop_ of the value function.
> These convexity assumptions are used to facilitate our error analysis.
> In other words, since CVI essentially solves the dual problem, it suffers from a non-zero duality gap for non-convex problems.
>
> ## (i.2) Relaxation of feasibility assumption
> Let us clarify that in our analysis of CVI algorithm in Section 3.2 the standing assumption is the (relaxed) feasibility Assumption 3.5 for all discrete $x \in \mathbb{X}^{\mathrm{d}}$, rather than the (original) feasibility Assumption 3.1-(iii) for all continuous $x \in \mathbb{X}$.
> While the inverted pendulum example does not satisfy Assumption 3.1-(iii),  it does satisfy Assumption 3.5.
>
> ## (ii) Practicality of the error bound of Theorem 3.11
> We agree with the reviewer that the error bound of Theorem 3.11 does not give explicit conditions on the algorithm's parameters for a prescribed bound on the error (due to the fact that some of the constants depend on the true optimal value function to which we do not have access).
> However, the provided error analysis is particularly used for providing specific guidelines/procedures for construction of the grids $\mathbb{Y}^{\mathrm{g}}$, $\mathbb{V}^{\mathrm{g}}$ and $\mathbb{Z}^{\mathrm{g}}$ in Section 3.4.
> Let us elaborate on this.
> The error bound of Theorem 3.11 most importantly captures the error due to discretization of the primal and dual state and input spaces ($e_{\mathrm{d}}$).
> In particular, it shows the consistency of the proposed algorithm in the sense that as we use finer discretizations, the output of the algorithm approaches $J_\star$.
> Moreover, through this error analysis, we were able to identify a set of conditions for the grids $\mathbb{Y}^{\mathrm{g}}$, $\mathbb{V}^{\mathrm{g}}$ and $\mathbb{Z}^{\mathrm{g}}$ in order to minimize the discretization error.
> As we mentioned above, these conditions are then translated to specific guidelines/procedures for construction of these grids.
>
> ## (iii) Linear state dynamics and preservation of convexity
> Indeed, for the duality gap to be zero, we require the value function $J_k$ to remain convex in each iteration $k$.
> This, in turn, requires the composition $J_k(f(x,u))$ to be jointly convex in $x$ and $u$, given that $J_k$ is convex.
> Linearity of the state dynamics $f_{\mathrm{s}}(x)$ is a sufficient condition as the reviewer also clarified, however, it is not necessary.
> For example, the dynamics of the inverted pendulum in the numerical example of Section 4 is not linear, however, within the considered state space, $J(f(x,u))$ is convex if $J$ is convex.
>
> ## (iv.1) When it is beneficial to use CVI?
> The proposed CVI algorithm can be employed with a time complexity of $\mathcal{O}(X+U)$, compared to $\mathcal{O}(XU)$ complexity of (standard) VI.
> This is because this class of problems allow us to use the operational duality of addition and infimal convolution with respect to the conjugate transform: $(f_1 \diamond f_2)^* = f_1^* + f_2^*$, where $ f_1 \diamond f_2 (x) = \inf \\{ f_1(x_1) + f_2(x_2): x_1+x_2 = x \\}$.
> In particular, we use the preceding operational duality in order to transform the minimmization over input in the DP operation into a simple addition.
> Therefore, one can particularly enjoy this reduction when the system has a multi-dimensional action space.
> Nevertheless, in our numerical example, we intentionally chose a system with a two-dimensional state space and a one-dimensional action space to show that even in such a scenario, application of CVI can lead to a huge reduction in the time requirement compared to VI.
> As the reviewer also mentioned in the summary, the reason is that the size of discrete state and input spaces increase exponentially with their dimensions,
> hence the reduction in the exponential growth rate from $n+m$ to $\max (n,m)$ already has a significant effect in our numerical example with $n/m=2$.
>
> ## (iv.2) Simplicity of the numerical simulations
> Following reviewer's suggestion, we implemented and plan to include the numerical results for a system of a higher dimension (namely, unstable batch reactor, with $n=4$ states and $m=2$ inputs, borrowed from "Computer aided control system design'' by Rosenbrock).
> Here are the results we derived for the discrete state and input spaces of size $X = 11^4$ and $U=11^2$, respectively:
>
> | Algorithm | Running time (sec) | Average cost (50 runs) |
> |:----:|:----:|:----:|
> | VI | 9125 | 24.6 |
> | CVI | 65 | 36.6 |
> | CVI-d | 119 | 22.2 |

---

> > ### Comment · Reviewer_cP9r · 2021-09-01
> > **Comment on Response to Reviewer cP9r**
> >
> > Thank you to the authors for their response. This confirms my overall positive assessment for acceptance.
> >
> > About response to (iii) on the inverted pendulum, "J(f(x,u)) is convex if J is convex":  the additional assumption stated (J convex) is actually what needs to be proven. Furthermore, in the model where the dynamics are linearized, the value function being considered is the approximate value function associated with the linearized dynamics. It is likely that the approximate value function is convex. It is less evident that the exact value function associated with the exact nonlinear dynamics is convex...

---

> > > ### Author Response · Authors · 2021-09-06
> > > **Response to Reviewer cP9r**
> > >
> > > We thank the reviewer for his/her encouraging words.
> > >
> > > We agree with the reviewer.
> > > As the reviewer also noticed, the only generic case, that we are also aware of, in which the convexity is preserved, is for linear dynamics $x^+ = Ax+Bu+w$, with convex stage cost $C$ (jointly in $x$ and $u$) and convex state and input constraints $x \in \mathbb{X}$ and $u \in \mathbb{U}$; see, e.g., _Linear Convex Stochastic Control Problems Over an Infinite Horizon_ by D. Bertsekas. Indeed, showing that $J_k$'s  stay convex over consecutive iterations for nonlinear dynamics requires a treatment specific to the case at hand (it is not possible for a generic nonlinear dynamics). We will further clarify this point in the final version of the paper.

---

### Decision · Program_Chairs · 2021-09-27

**Decision:**

Accept (Poster)

**Comment:**

The paper suggests a fast method for solving a class of optimal control problems by introducing a variant of the Value Iteration (VI) algorithm that benefits from the convex conjugate transform. This improves the computational complexity of solving each step of VI from O(|X||U|) to O(|X| + |U|).

Most reviewers are positive about this work. One of the reviewers (mb6w) is on the negative side, but I believe their concerns and questions are adequately answered (the reviewer did not acknowledge the authors' response). Therefore, I would recommend acceptance of this paper.

I would encourage the authors to incorporate reviewers' comments and suggestions in order to improve their paper. Some of improvements can be along the followings:

- Emphasizing more how this work can be relevant to NeurIPS community, as this paper is different from a typical paper on MDP or RL appearing at NeurIPS (no focus on learning; focusing on a special class of problems, etc.)
- Clarifying and emphasizing the technical contributions of this work compared to [18].
- Including larger-scale experiments.

I would also note that the acronym of CVI has already been used in the context of Value Iteration:
*Conservative Value Iteration* (Kozuno, Uchibe, Donya, "Theoretical Analysis of Efficiency and Robustness of Softmax and Gap-Increasing Operators in Reinforcement Learning," AISTATS, 2019) and *Characteristic Value Iteration* (Farahmand, "Value Function in Frequency Domain and the Characteristic Value Iteration Algorithm," NeurIPS, 2019), so the authors may want to use another acronym to reduce possible confusion. There is no need to cite these work, as they are not related enough.